# FastCTM (v1.0): Atmospheric chemical transport modelling with a principle-informed neural network for air quality simulations

Baolei Lyu[1,2,3], Ran Huang[4,5], Xinlu Wang[4], Weiguo Wang[6], Yongtao Hu[7]

[1] Huayun Sounding Meteorological Technology Co. Ltd., Beijing 102299, China

[2] Key Laboratory of Intelligent Meteorological Observation Technology, Beijing 100081, China

[3] China Meteorological Administration Xiong'an Atmospheric Boundary Layer Key Laboratory, Xiong'an, 071000, China

[4] Hangzhou AiMa Technologies, Hangzhou, Zhejiang 311121, P. R. China

[5] Nanjing AiMa Environmental, Nanjing, Jiangsu 210000, P. R. China

[6] SAIC, at Environment Modelling Center, NOAA/National Centers for Environmental Prediction, College Park, Maryland 20740, United States

[7] School of Civil and Environmental Engineering, Georgia Institute of Technology, Atlanta, Georgia 30332, United States

*Correspondence to*: Baolei Lyu (baoleilv@foxmail.com), Ran Huang (ranhuang2019@163.com)

**Abstract.** Chemical-transport models (CTMs) are indispensable for air-quality assessment and policy development, yet their operational use is hampered by high computational cost. We present FastCTM, a physics-informed neural-network emulator that rapidly predicts hourly concentrations of ten key pollutant variables: major $PM_{2.5}$ species ($SO_4^{2-}$, $NO_3^-$, $NH_4^+$, organic matter, elemental carbon, crustal material), coarse $PM_{10}$, $SO_2$, $NO_2$, CO, and $O_3$. FastCTM embeds five process-specific neural modules—primary emissions, horizontal transport, turbulent diffusion, chemical reactions and deposition within a unified framework. Given 1-hour initial condition data, FastCTM can simulate future 24-hour concentrations for ten air pollutants using corresponding meteorological fields and emissions as input. Trained on 2018–2022 WRF-CMAQ forecasts over China and evaluated on 2023 data, FastCTM reproduces CMAQ with mean RMSE ($\mu g\ m^{-3}$) of 9.1, 11.9, 4.4, 4.0, 48.9, 10.9 and $R^2$ of 0.80, 0.81, 0.80, 0.83, 0.90, 0.70 for $PM_{2.5}$, $PM_{10}$, $SO_2$, $NO_2$, CO and $O_3$, respectively. Sensitivity tests confirm physically plausible responses to temperature, wind speed, boundary-layer height and precursor emissions. The modular architecture enables quantitative process analysis, offering CTM-like insight at GPU-accelerated speeds. In a nutshell, FastCTM provides a computationally efficient solution for air-quality simulations, sensitivity analysis, and process attribution with high accuracy and physical consistency.

## 1 Introduction

Effective air quality management requires accurate characterization of current and future pollution conditions to implement targeted emission control measures (Wang et al., 2010; Council, 2004). Driven by this demand, deterministic air quality numerical models have been developed to simulate the spatiotemporal variability and evolution of ambient air pollutants in the atmosphere (Hakami et al., 2003; Eder et al., 2006). In these models, such as the Community Multiscale Air Quality (CMAQ) model, atmospheric physical and chemical processes (e.g., emissions, chemical reactions, horizontal advection, and diffusion) are mathematically represented by partial differential equations. The air pollutant and species concentrations can then be calculated by solving these complex equations using numerical methods (Byun and Schere, 2006), which is often time-consuming (Leal et al., 2017) and requires substantial computational resources such as high-performance computing (Efstathiou et al., 2024).

Deep learning offers promising alternatives for developing rapid, data-driven CTMs by leveraging the capacity of neural networks to encode complex spatiotemporal patterns from large datasets (Lecun et al., 2015; He et al., 2016; Liao et al., 2020). These deep learning-based CTM models are expected to provide accurate simulations that are comparable to the current deterministic numerical CTMs while offering much higher computational efficiency and enhanced learning capabilities. However, progress has been hindered by challenges in designing neural architectures that simultaneously achieve high accuracy, interpretability, and long-term simulation stability and fidelity (Reichstein et al., 2019; Irrgang et al., 2021). In constructing deep learning-based CTM models, air quality modeling is often formulated as a sequence-to-sequence prediction problem (Shi et al., 2015; Zhang et al., 2024) to capture the spatiotemporal correlations among multiple variables. Consequently, previous studies have mainly focused on refining neural network's representation capabilities by proposing new neural-network structures to improve error back-propagation efficiencies and model encoding capabilities (Wang et al., 2018; Huang et al., 2021; Mao et al., 2021). For example, Xing et al. (2022) developed a deep learning-based module named deepCTM that mimics atmospheric photochemical modeling to simulate ozone concentrations. However, these deep learning-based CTMs are often structured as uninterpretable black-box models that generate simulations reflecting the cumulative effect of all physical and chemical processes. Such black-box models hinder error attribution, inspection of internal processes, and knowledge discovery (Reichstein et al., 2019).

Quantifying individual atmospheric processes enables a mechanistic interpretation of model predictions and identification of error sources (Liu et al., 2010). Motivated by this need, recent studies have developed models that learn specific atmospheric processes, such as chemical reactions and deposition, within CTM frameworks. Kelp et al. (2022) developed a neural network chemical solver for stable long-term global simulations of atmospheric chemistry, trained from the GEOS-Chem model. Xia et al. (2024) simulated 74 chemical species and 229 reactions following the SAPRC-99 mechanism using an artificial intelligence photochemistry (AIPC) scheme, achieving approximately 8-fold speed-up. Sturm and Wexler (2020) developed a mass- and energy-conserving framework for using machine learning to accelerate computations, demonstrating successful application in a photochemistry example. For the deposition process, Silva et al. (2019) proposed a deep learning parameterization for ozone dry deposition velocities that provided accurate predictions on independent new datasets, revealing the potential of neural networks to capture complex spatio-temporal latent processes. Liu et al. (2025) proposed a Neural Network Emulator, named ChemNNE, for rapid chemical concentration modelling, which achieved strong performance in both accuracy and efficiency. Although these successes, a gap remains in coupling these NN operators into a complete deep-learning CTM.

The main objective of our study is to develop and validate a principles-guided, neural network-based FastCTM, capable of simulating spatial-temporal fields of hourly concentrations of 10 criteria pollutants, including major species of $PM_{2.5}$ ($SO_4^{2-}$, $NO_3^-$, $NH_4^+$, organic matters and other inorganic components, coarse part in $PM_{10}$, CO, $NO_2$, $SO_2$, and $O_3$. FastCTM models individual atmospheric process: transport, diffusion, deposition, chemical reactions, and emissions. FastCTM is capable of performing analysis of internal chemical and physical processes, offering benefits like high computational speed, efficient data assimilation, and rapid model updates.

## 2 Data and Methods

### 2.1 Parent Model Simulations and Datasets

In this study, the FastCTM model was designed to replicate the parent model CMAQ, trained by learning CMAQ's

underlying physical and chemical processes among multiple air pollutants, including the complicated chemical reaction, transport, diffusion, and deposition. CMAQ has a process analysis (PA) tool to separate out and quantify the contributions of individual physical and chemical processes to the changes in the predicted concentrations of a pollutant, which provides the opportunity to conduct a sensitivity analysis by comparing process contributions between CMAQ and FastCTM.

Weather and air quality simulations from 2018 to 2023 were conducted using a WRF-CMAQ modeling system consisting of three major components: (1) the meteorology component, the Weather Research and Forecasting model, WRF v3.4.1 (Michalakes et al., 2005; Skamarock et al., 2008), which provides meteorological fields; provides meteorological fields, (2) the emission component, which supplies gridded estimates of hourly emission rates for primary pollutants matched to model species, and (3) the CTM component, CMAQ v5.0.2 (Byun and Schere, 2006), which solves the governing physical and chemical equations to obtain 3-D pollutant concentration fields. WRF-CMAQ simulations are not two-way coupled, so weather and chemistry do not influence each other. We used hourly average concentrations of dominant $PM_{2.5}$ components of sulfate ($SO_4$), nitrate ($NO_3$), ammonium ($NH_4$), organic carbon (OC), and other components (EC and soil, etc.), and CO, $SO_2$, $NO_2$, and $O_3$ in the surface layer. The 10 species were selected based on their direct relevance to regulatory standards (e.g., $PM_{2.5}$, $PM_{10}$, $O_3$, $NO_2$, $SO_2$, and CO) and their dominance in driving health and environmental impacts in urban and industrial regions.

Meteorological variables used in this study include relative humidity (RH), air temperature (T), wind components (U, V) at surface 10 meters height, precipitation (RN), cloud fraction (CFRAC), and planetary boundary layer height (PBLH). Wind speed (WS) was calculated from U and V. The data covered the whole of China at a horizontal resolution of 12km with 372×426 grid cells. The simulation data from 2018-2022 are used as the training dataset, while the remaining simulation data in 2023 is used for independent evaluation. The surface topographic data (HGT, Figure S1 in the supplementary material, obtained from https://lta.cr.usgs.gov/GTOPO30) and land cover data (Zhang et al., 2020) of urban and tree fraction (LULC) are also used to reflect the effects of land surface conditions in this study.

The original primary emissions used in the aforementioned WRF-CMAQ modelling system are used as input to the FastCTM. The large amount of emission data is grouped according to the simulated 10 pollutant variables. Specifically, the primary $PM_{2.5}$ emissions of $SO_4$, $NO_3$, $NH_4$, OC, and other components, and gaseous emissions including sulfur oxide ($SO_2$), nitrogen oxides ($NO_x$, including HONO, NO, and $NO_2$), ammonia ($NH_3$), volatile organic species (VOCs, including isoprene (ISOP), terpene (TERP), and other species of VOC) are used in the FastCTM. Annual average emissions of NOx, $SO_2$, and VOC are respectively depicted in Figure S2-4 in the supplementary material.

**2.2 FastCTM Model Formulations**

**2.2.1 FastCTM Model Framework**

The deterministic CTM models simulate emissions, transport, deposition, diffusion, and chemical transformations of gases and particles in the troposphere through numerically solving the governing equations as follows,

$$\frac{\partial C_i}{\partial t} = -\nabla \cdot (\vec{u} C_i) + \nabla(K \nabla C_i) + R_i + E_i + D_i \ (1)$$

where $C_i$ is the concentration of species $i$, $u$ is the air fluid velocity, $K$ is the eddy diffusivity tensor, $R_i$ is the net rate of chemical generation of species $i$, $E_i$ is the rate of direct addition of the species from primary emissions, and $D_i$ is the deposition rate caused by both dry and wet depositions. A detailed description of CMAQ principles is available elsewhere (Byun and Schere, 2006; Appel et al., 2017). Inspired by numerical CTMs principles and equations, the guiding framework of FastCTM was also structured in a similar formulation to represent the dominant processes in order to simulate air

pollutant spatiotemporal variations.
In the context of deep learning, hourly air quality simulation is a spatiotemporal sequence-to-sequence learning problem
aimed at predicting the most probable future sequence of length K, given a previous sequence of length J, as shown in Eq.2,
$$\hat{Y}_{t+1},\dots,\hat{Y}_{t+K} = arg\max p\left([Y_{t-J+1},\dots,Y_t],[X_{t-J+1},\dots,X_t,X_{t+1},\dots,X_{t+K}]\right) \quad (2)$$
Where the *arg* max (short for "argument of the maximum") function is used to find the *p* class with the highest predicted
probability. The $X_t \in \boldsymbol{R}^{M \times N \times V_X}$ is the data of $V_X$ input variables at the spatial grid of $M \times N$ at time *t*. The $Y_t \in \boldsymbol{R}^{M \times N \times V_Y}$
is the data of $V_Y$ predictive variables at time *t*. Specifically, the FastCTM simulates future *K*-hour air pollutant
concentrations, given *J*-hour air pollutant concentrations $[Y_{t-J+1},\dots,Y_t]$ as initial fields and (*K+J*)-hour meteorological and
emission conditions $[X_{t-J+1},\dots,X_t,X_{t+1},\dots,X_{t+K}]$. Previous studies generally used multiple-step input data with *J*>1 to
ensure sufficient spatial-temporal correlations contained in the training data (Sum et al., 2022; Xing et al., 2022). Instead,
we use a 1-hour initial pollutant concentration (*J*=1) to simulate 24-hour air quality pollutants (*K*=24), to ensure FastCTM
is dedicated to learning air quality changes between two neighboring hours as shown in Figure 1a. In other words, at time
$t = 0$, FastCTM predicted *K*-hour air pollutant concentrations of $C_{t=0}, C_{t=1}, \dots, C_{t=K-1}$, given the input air pollutant
concentration in the previous hour $C_{t=-1}$ and corresponding meteorological data and emissions at time $t = 0,1,\dots,K$-1. The
unit of concentrations is μg/m$^3$ for all pollutants.

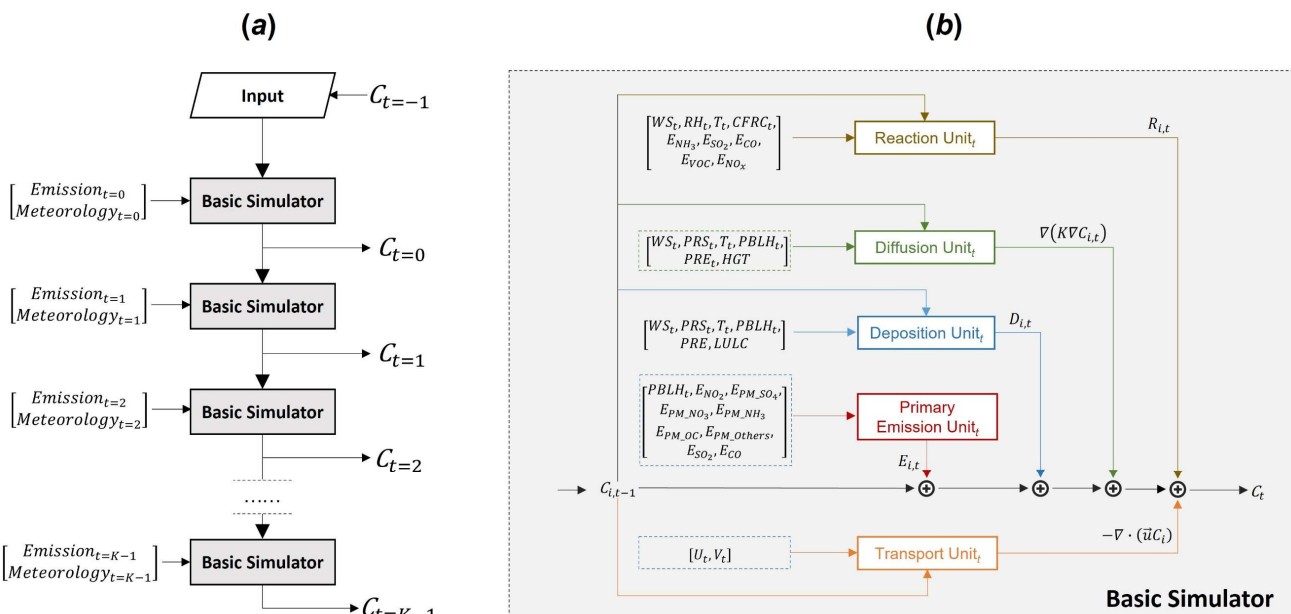

**Figure 1: (a) General model workflow, and (b) the basic simulator module structure at the time step *t* of the deep learning**
**simulation model FastCTM, designed according to Eq.1. Arrows and boxes with different colours represent calculation modules**
**of different atmospheric physical and chemical processes.**
The FastCTM model uses the basic simulator module (Figure 1a) recursively for hourly simulations, using output air
pollutant concentrations from one step as input to the next step basic simulator. In contrast to directly learning
spatiotemporal correlations of predictand itself as in most previous studies (Wang et al., 2018; Shi et al., 2017), the basic
simulator (Figure 1b) is formulated following the atmospheric physical and chemical equations and constraints shown in
Eq.1, and is composed of five modules to respectively represent the physics-chemical processes to improve the model
performance. The modules for each of the five processes in the basic simulator are described in the following section. The
time step used in FastCTM was 60 seconds.

**2.2.2 Primary Emissions Module**

Primary pollutants are assumed to be directly emitted into the atmosphere and instantly well-mixed within the PBL. Therefore, hourly enhancement of air-pollutant concentrations caused by primary emissions could be described in the following Eq.3.

$$E_{m,n,i,t} = \frac{1000 \times PE_{m,n,i,t}}{PBLH \times \mathrm{d}x \times \mathrm{d}y} \quad (3)$$

Where $E_{m,n,k,t}$ refers to the concentration changes contributed by primary emissions at spatial coordinate $(m, n)$ for species $i$ at time $t$. The $PE_{m,n,i,t}$ is the corresponding total primary emissions within the grid cell per second, which has a unit of g/s. Considering that the cell size in the FastCTM is 12km by 12km, we have $\mathrm{d}x = 12000$ and $\mathrm{d}y = 12000$ in this study. The boundary layer height PBLH ,i s also in the unit of meters(m). Therefore, the resulting air pollutant concentration increases by primary emission $E_{m,n,i,t}$ has a unit of $\mu g/m^3$.

**2.2.3 Horizontal Transport Module**

In the FastCTM, horizontal transport usually has a significant influence on air quality variations (Lang, 2013). In CMAQ, the regional transport is generally represented by the divergence of the product of wind field and air pollutant species as in Eq.1, inferred from continuity equations and convection equations (Michalakes et al., 2001; Byun and Schere, 2006). By decomposing the air mass movement into two orthogonal directions of east-west ($x$) and north-south ($y$), they could be rewritten in the form shown in Eq. 4,

$$\nabla \cdot (\vec{u} C_i) = \frac{\partial (C_i U)}{\partial x} + \frac{\partial (C_i V)}{\partial y} \quad (4)$$

Where the wind field is represented as $\vec{u}$, which is then decomposed into $U$ and $V$, respectively, in the $x$ and $y$ directions. In the deep learning framework, the partial equation in Eq. 4 could be rewritten in a discrete form as convolution operations and inner product calculations as shown in Eq. 5 with a finite difference method. The convolutional kernels of $W_x$ and $W_y$ were defined in an upwind scheme as shown in Eq. 6 and Eq. 7. With the scheme, this transport module itself is mass-conserved, even though FastCTM is not mass-conserved as a whole.

$$\nabla \cdot (\vec{u} C_i) = \frac{W_x * (C_i \times U)}{\mathrm{d}x} + \frac{W_y * (C_i \times V)}{\mathrm{d}y} \quad (5)$$

$$W_x = \begin{cases} [-1 \quad 1 \quad 0] & if \ U < 0 \\ [0 \quad -1 \quad 1] & if \ U \geq 0 \end{cases} \quad (6)$$

$$W_y = \begin{cases} \begin{bmatrix} 0 \\ 1 \\ -1 \end{bmatrix} & if \ V < 0 \\ \begin{bmatrix} 1 \\ -1 \\ 0 \end{bmatrix} & if \ V \geq 0 \end{cases} \quad (7)$$

**2.2.4 Diffusion Module**

Diffusion involves the physical and chemical processes that disperse pollutants in the atmosphere. It is influenced by meteorological conditions, i.e. atmospheric stability and humidity, and surface features, i.e., land terrains and vegetation (Jiang et al., 2021). The turbulence diffusion process $\nabla(K \nabla C_i)$ in Eq.1 helps the spread of pollutants in the atmosphere. It is expressed as the second-order deviation of species concentrations as shown in Eq. 8. They could also be discretized to

convolutional operations with the finite difference method as shown in Eq. 9, just like that in the horizontal transport
process module.

$$\nabla(K\nabla C_i) = \frac{\partial}{\partial x}\left(K\frac{\partial C_i}{\partial x}\right) + \frac{\partial}{\partial y}\left(K\frac{\partial C_i}{\partial y}\right) (8)$$

$$\nabla(K\nabla C_i) = \frac{W_x*(K\times W_x*C_i)}{dx\times dx} + \frac{W_y*(K\times W_y*C_i)}{dy\times dy} (9)$$

$$K = Encoder_K([T, RH, PRS, PBLH]) (10)$$

The turbulent diffusivity $K$ is closely related to the meteorological conditions of the atmosphere and is simulated with an
encoder module $Encoder_K$ (Eq. 10). The input variables of the $Encoder_K$ include temperature $T$, humidity $RH$,
pressure $PRS$, and boundary layer height $PBLH$. The $Encoder_K$ is determined to be a grid-to-grid regression model based
on the Unet++ model with a nested structure (Zhou et al., 2018; Ronneberger et al., 2015). The $Encoder_K$ model consists
of 5 layers with each layer respectively composed of 16, 32, 64, 128 and 256 filters.

**2.2.5 Chemical Reaction Module**

Reduced-form models like InMAP (Tessum et al., 2017) and EASIUR (Gentry et al., 2023) focus on annual-average
exposure, while FastCTM provides hourly-resolved simulations critical for real-time management. FastCTM quantifies
hourly contributions from individual processes (transport, chemistry, emissions) via its modular design, rather than
aggregating source impacts in reduced-form models (e.g., EASIUR's source-receptor matrices). Furthermore, FastCTM
explicitly couples meteorology (PBLH, T, RH) with chemistry, whereas InMAP/APEEP (Muller and Mendelsohn, 2006)
assume static meteorology, which limits their utility in capturing diurnal or synoptic-scale variations. Specifically, the air
pollutant concentration changes caused by chemical reactions are represented in the following Eq. 11. In the equation, the
rate of chemical reaction of species $i$ is expressed as the product of a rate constant $k$ and a term that is dependent on the
concentrations of its reactants $j$ (Carter, 1990; Carter and Atkinson, 1996).

$$R_{m,n,i,t} = k_{m,n,i,t} \times f\left(C_{m,n,j,t}\right) (11)$$

$$k_i = Encoder_k([T, RH, PRS, WS, PRE, CFRAC]) (12)$$

The reaction kinetics constant $k$ is generally temperature-dependent. They could also be related to atmospheric pressures
and moisture humidity in some reaction processes. Therefore, the reaction rate constant $k$ is simulated using a spatial
encoder function $Encoder$ as shown in Eq. 12, which has the same structure as that of diffusion encoder modules (Eq. 10).
There are 6 input variables of the $Encoder_k$ including $T, RH, PRS, WS, RN$ and $CFRAC$. The concentration processor $f$ is
designed as a simple multi-layer convolutional network with a kernel size of 1 to represent high-order and complex relations
among different reactants.

**2.2.6 Deposition Module**

Air pollutant deposition refers to the process by which atmospheric pollutants are transferred to Earth's surfaces (land,
water, vegetation) or removed from the air. This phenomenon plays a critical role in environmental pollution dynamics and
ecosystem impacts. The deposition was closely influenced by meteorological conditions and surface characteristics (Janhäll,
2015). For example, high wind disperses pollutants, while turbulence enhances dry deposition. Forests and crops act as
sinks due to large surface areas for adsorption. Air quality changes due to the deposition process are expressed linearly as
the product of the deposition rate $d$ and the corresponding air pollutants concentrations $C$, as shown in Eq. 13. The constant
$d$ is closely related to the current and previous meteorological conditions, terrains, and underlying land cover types.
Therefore, they are all simulated with an *Encoder* module as shown in Eq. 14.

$$D_{m,n,i,t} = d_{m,n,i,t} \times C_{m,n,i,t} \quad (13)$$

$$d = Encoder_d([WS, RH, RN, HGT, LULC]) \quad (14)$$

The model structure and parameter configurations are also the same as that of $Encoder_K$ and $Encoder_k$. The input data
variables of $Encoder_d$ include WS, RH, RN, HGT and LULC.

## 2.3 Model Training

The FastCTM was programmed with Python 3 on the deep learning framework TensorFlow (Abadi et al., 2016). The model
was trained with the WRF-CMAQ operational forecast data in China for 2018-2022. Considering that on each day we had
120-hour forecasts with a spatial coverage of 426×372 grid cells (each with a size of 12×12km$^2$) for 9 meteorological
variables and $I$=10 air pollutant variables, the total training dataset has a size of $\boldsymbol{TD} = \boldsymbol{R}^{1826,120,426,372,19}$, where 1826
represents the total counting days from 2018 to 2022. Since the model was set to predict 24-hour PM$_{2.5}$ concentrations from
1-hour input data, the total input sequence length was 25 hours in each training step. Besides, the size $M \times N$ of the input
data $X_t$ to FastCTM was decided to be 150×150, equal to an area of 1,800×1,800 km$^2$ in 12-km resolution. Therefore, the
input batch data for FastCTM in each step should be the size of $\boldsymbol{BD} = \boldsymbol{R}^{b,25,150,150,19}$, where $b$ is the batch size (determined
as 1 in this study). The input data $\boldsymbol{BD}$ are randomly sliced from the whole training dataset $\boldsymbol{TD}$ in each training iteration,
indicating each $\boldsymbol{BD}$ represents different spatial and temporal coverages. The random sampling tactics help the model learn
inherent physical and chemical principles rather than just statistical spatiotemporal autocorrelations using data in a constant
spatial area (Xing et al., 2022). Besides, the spatio-temporal random samples contain varied emissions, which would
improve FastCTM adaptation to changing emission levels.
Even though five modules are defined in FastCTM, individual processes are not trained separately. The model was trained
as a whole with hour-to-hour air pollutant concentrations, while each process could learn its parameters under the
constraints of its dedicated formulation. Specifically, FastCTM was tuned to minimize the loss function $\mathcal{L}$, which was
determined to be L2 loss (Bühlmann and Yu, 2003) of the regularized mean squared error (MSE) as shown in Eq. 15. The
model was optimized using the Adam optimizer (Kingma and Ba, 2014).

$$\mathcal{L} = \frac{1}{J \times N \times M \times I} \sum_{t=1}^{J} \sum_{m=1}^{M} \sum_{n=1}^{N} \sum_{i=1}^{I} \left( C_{m,n,i,t} - \tilde{C}_{m,n,i,t} \right)^2 \quad (15)$$

The learning rate was set to be 0.001, and the batch size to be 1. The FastCTM model was trained on one entry-level
professional acceleration card of NVIDIA A40 with a running time of 10 hours for every 10000 iterations. A total of 300,000
iterations were performed until the remaining model loss stabilized.

## 2.4 Model Evaluation

FastCTM was assessed against CMAQ simulations using the same input emission data and meteorological fields. Starting
from 0:00 local time on each day, the CMAQ model simulated 120-hour forecasts in one cycle. There are 139 cycles in the
evaluation year of 2023 due to data unavailability in the remaining days. The FastCTM model generated 119-hour forecasts
using a 1-hour initial input condition. The 119-hour forecasts in the leading hours from 2 to 120 by the two models were
compared regarding to the corresponding leading time. For example, when we had a 120-hour forecast starting at 0:00 on
January 1, 2023 at Beijing Local Time (BLT), the data of 0:00 on January 1, 2023 were fed into FastCTM to get the 119-
hour forecasts until 23:00 on January 5. The 10 species forecasts by FastCTM were compared against the CMAQ forecasts
at each corresponding hour. The metrics of root mean square error (RMSE) and coefficient of determination (R$^2$) were

calculated daily in each of 119 leading hours on the difference in each of the 158,742 grid cells between CMAQ and FastCTM. Therefore, metrics of $R^2$ and RMSE were obtained on each lead hour on each day of the independent test year of 2023. The statistical values on each day are then averaged for the same leading hour for comparison.

The FastCTM was also assessed in terms of sensitivity analysis to emission inputs and meteorological fields. For meteorological variables, responses of six criteria pollutant concentrations to T, WS, and PBLH were calculated. For emissions, responses to paired variables of $SO_2$/$NH_4$ and $NO_x$/VOC were calculated. Besides, FastCTM's capability to simulate responses to emission changes was also evaluated by comparing with CMAQ simulations in 11 emission-intervention scenarios. Finally, the contributions of five internal processes of transport, diffusion, emission, reaction, and deposition were also analyzed and discussed for an example pollution episode.

## 3 Results

### 3.1 Forecast Performance by FastCTM

FastCTM has exhibited strong, stable performance in reproducing CMAQ forecasts over the 119-hour forecast period evaluated for 2023 (Figure 2). The average RMSE values for six criteria pollutants of $PM_{2.5}$, $PM_{10}$, $SO_2$, $NO_2$, CO, and $O_3$ are, respectively 9.1, 11.9, 4.4, 4.0, 48.9 and 10.9 μg/m$^3$. For $R^2$ values, they are 0.8, 0.81, 0.8, 0.83, 0.9 and 0.7. As for $PM_{2.5}$ components, RMSE values are 1.68, 2.68, 1.52, 1.98 and 4.25 μg/m$^3$, respectively for $SO_4^{2-}$, $NO_3^-$, $NH_4^+$, organic matters and other inorganic components, while the $R^2$ values are 0.72, 0.6, 0.3, 0.83 and 0.68. Compared to the ~5ppb (~10.5 μg/m$^3$) in the previous study by Xing et al. (2022), the FastCTM model has similar RMSE values in forecasting $O_3$. To test the influences of initial conditions on FastCTM long-term simulations, FastCTM forecasts using zero values as input air quality data were almost the same as those using ordinary input in the long leading hours. Results indicating that FastCTM simulations in long leading hours are not affected by initial conditions (Figure S5 in the SI), just like deterministic CTMs (such as CMAQ). In other words, the insensitivities of FastCTM to initial conditions indicate that it has well learned and encoded the most physical and chemical principles in CMAQ CTM, rather than just spatio-temporal correlations among air quality sequences.

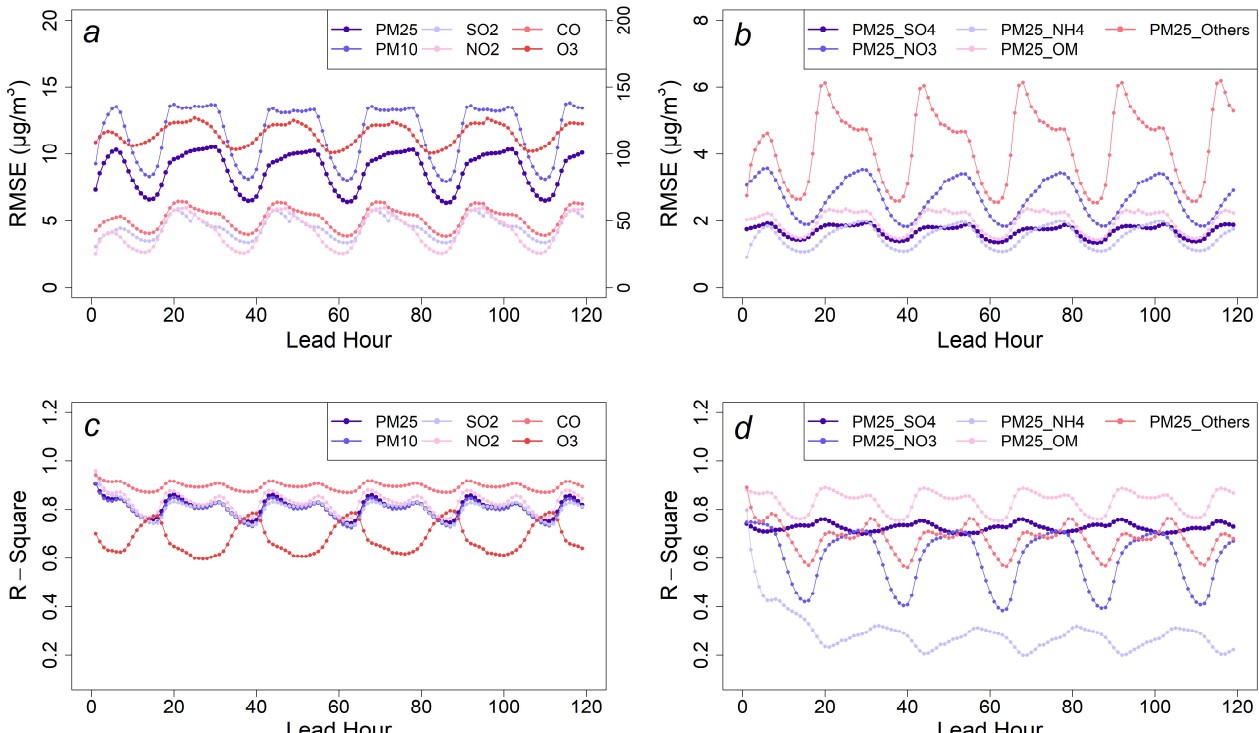

**Figure 2: The evaluation performances of FastCTM forecasts against CMAQ forecasts in 2023. Panel (a) and (b) respectively show RMSE values of criteria pollutants and the PM$_{2.5}$ components. Panel (c) and (d) show R$^2$ values. It should be noted that the RMSE value of CO corresponds to the right axis in panel (a).**

Hourly RMSE values show clear diurnal variation with higher RMSE values in the nighttime than that in the daytime, which could be due to higher hourly concentrations of air pollutants in the nighttime, except for O$_3$ (Figure S6 of SI). Consistency between CMAQ and FastCTM, as characterized by R$^2$, is lower in the daytime. Since the FastCTM is a 2-D model only considering atmospheric processes within the boundary layer, lower consistency with the CMAQ model during daytime, possibly due to more vigorous vertical mixing. Strong vertical mixing of air pollutants to the height above PBLH has been found (Li et al., 2017; Tang et al., 2016), which may not be fully represented in FastCTM. It is important to note that the relatively low R$^2$ values observed for NH$_4^+$. While CMAQ explicitly resolves NH$_4^+$ formation reactions, FastCTM does not explicitly encode these pathways. Instead, the neural network implicitly learns relationships between NH$_4^+$ and precursor emissions (NH3, NOx, SO2) and meteorological variables (e.g., temperature, humidity). This simplification omits acid-base equilibria and aerosol thermodynamics, which are critical for partitioning NH$_4^+$ between gas and particle phases. The low R$^2$ for NH$_4^+$ primarily reflects FastCTM's simplified chemical mechanism in this part, which could be improved by adding related species in the simulation.

The spatial distributions of the mean absolute error (MAE) and the normalized mean absolute error (NMAE) are presented in Figure 3. For all six pollutants under consideration, MAE values tend to be higher in polluted areas. In polluted environments, there are often multiple sources of emissions, complex chemical reactions, and variable meteorological conditions that can lead to greater discrepancies between the predicted concentrations of the two models. Conversely, the NMAE values exhibit an opposite trend, being lower in polluted areas. In these regions, the NMAE values typically hover around 0.2, in contrast to the relatively higher values of approximately 1 in cleaner areas. The NMAE is a normalized metric that takes into account the magnitude of the actual pollutant concentrations. A lower NMAE in areas with high pollution levels suggests that the FastCTM model is effectively capturing the overall magnitude and trends relative to the

reference CMAQ simulation. The Air quality forecasts starting from 00:00 a.m. on March 4th, 2023 (Figure S7 in the SI) demonstrate FastCTM's strong capability in modelling the complex spatio-temporal changes in a large spatial domain and over a relatively long period and a large area.

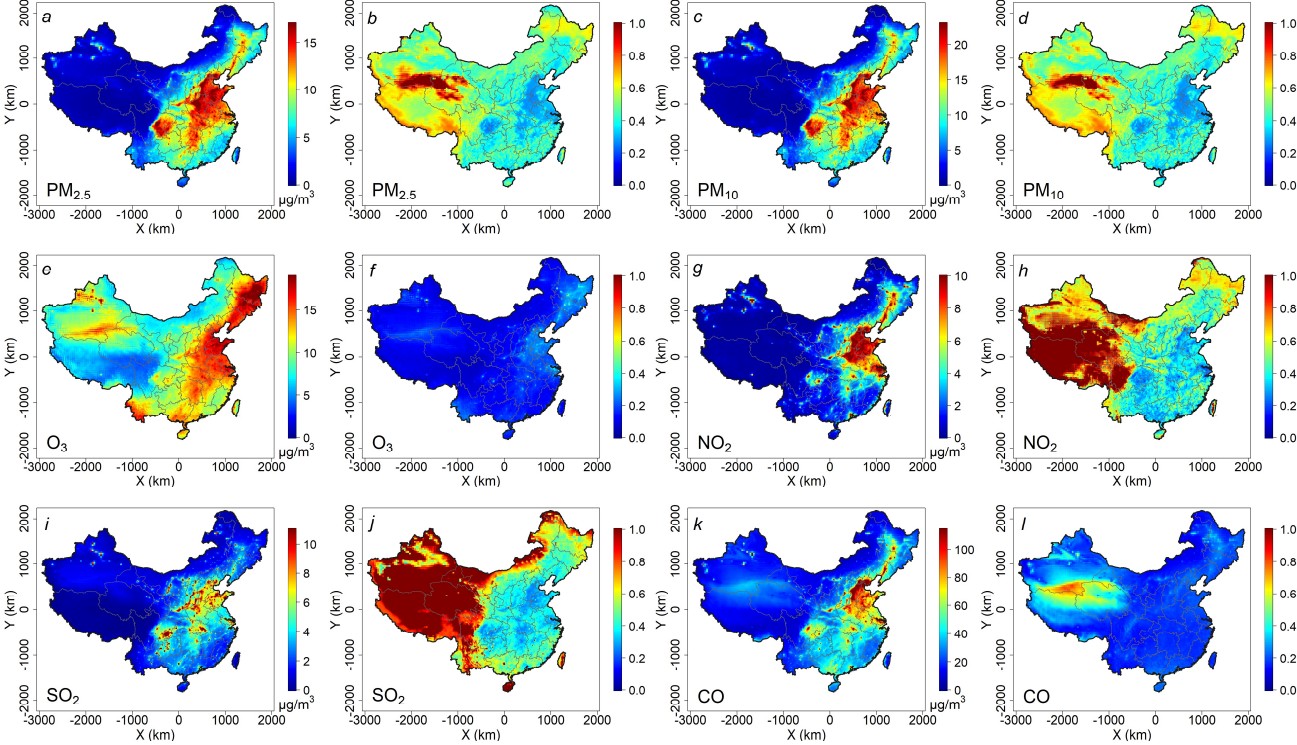

**Figure 3: Spatial distribution of mean absolute error (panels a, c, e, g, i, and k) and normalized mean absolute error for the six criteria pollutants (panels b, d, f, h, j, and l) of FastCTM compared with CMAQ in 2023.**

Defining the warm season as the months from April to September and the winter and cold season as the remaining months, the FastCTM model exhibited comparable performances. As shown in Figure 4 (with detailed information in Figure S8 in the SI), the coefficient of determination $R^2$ values for the six criteria pollutants were 0.82, 0.8, 0.8, 0.82, 0.91, and 0.7 in the warm season, and 0.8, 0.79, 0.78, 0.83, 0.88, and 0.68 in the cold season, respectively. To assess the performance variations of FastCTM across different spatial locations, comparative evaluations were carried out in urban and rural areas as well as in inland and coastal regions. Generally, FastCTM demonstrated slightly higher accuracies in rural areas compared to urban areas (as presented in Figure S9 in the SI). This outcome is reasonable given the more intricate emission and chemical processes prevalent in urban settings (Guo et al., 2014). Similarly, FastCTM exhibited comparable performances in inland areas to those in coastal areas, except for $PM_{2.5}$ and $PM_{10}$ (Figure S10 in the SI).

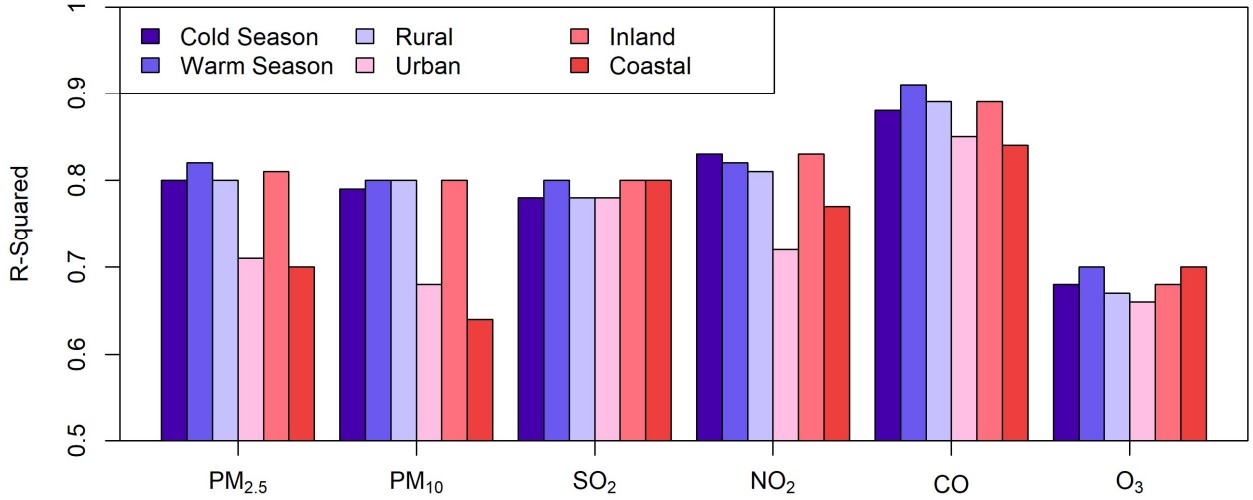

304

**Figure 4: The mean evaluation $R^2$ values for all 119 leading hours of FastCTM forecasts in warm/cold seasons, rural/urban areas, and coastal/inland areas.**

To validate the FastCTM model, three land use regression (LUR) models were constructed, namely the linear regression model, the random forest model (with the number of trees set at 500), and the XGBoost model (with the booster specified as gbtree). These LUR models were developed using the same input meteorological data, emissions, and geophysical variables as FastCTM to ensure fair comparison. When compared with the FastCTM model, the performance of the LUR models was found to be significantly inferior, as demonstrated in the Table. 1 and Figure S10 – S12 in the SI. For example, $R^2$ values for FastCTM range from 0.68-0.90, whereas the LUR models only achieve 0.06-0.33. This outcome is anticipated when we consider the complex nature of air quality dynamics in predicting future air quality. Air quality is not a static entity, but it varies both spatially and temporally, determined by the joint effects of local emissions, meteorological conditions, and surface features, etc. For instance, the transport of air pollution is a highly dynamic process that hinges on wind fields and air pollution concentrations in a reciprocal manner. The wind direction and speed dictate the trajectory along which pollutants travel, while the existing pollutant concentrations in different regions influence the overall dispersion and mixing patterns. LUR models, which on the other hand predominantly rely on local input data (Wong et al., 2021; Cheng et al., 2021), struggle to capture these intricate, non-local interactions. They cannot account for the far-reaching effects, such as wind-driven pollutant transport and the temporally accumulated changes in air quality over larger geographical areas. As far as we know, LUR models have been mostly applied in predicting air pollution fields in retrieval given corresponding air quality observations as training and constrained input data. They have been seldom used in air quality forecasts and simulations, as we have demonstrated with the FastCTM model.

**Table 1**. Performance metrics of LUR models and FastCTM compared against CMAQ

| Variable | Model | RMSE | $R^2$ | NMB |
|---|---|---|---|---|
| **PM$_{2.5}$** | FastCTM | 8.78 | 0.81 | -0.15 |
| | Liner Model | 35.05 | 0.09 | -0.24 |
| | Random Forest | 33.08 | 0.19 | -0.25 |
| | XGBoost | 33.02 | 0.14 | -0.12 |

| | | | | |
|---|---|---|---|---|
| **PM$_{10}$** | FastCTM | 11.58 | 0.80 | -0.17 |
| | Liner Model | 44.66 | 0.10 | -0.23 |
| | Random Forest | 45.07 | 0.19 | -0.33 |
| | XGBoost | 44.53 | 0.15 | -0.21 |
| **SO$_2$** | FastCTM | 4.51 | 0.80 | 0.09 |
| | Liner Model | 39.42 | 0.14 | -1.18 |
| | Random Forest | 25.74 | 0.33 | -0.65 |
| | XGBoost | 25.57 | 0.26 | -0.60 |
| **NO$_2$** | FastCTM | 4.24 | 0.83 | 0.04 |
| | Liner Model | 21.42 | 0.27 | -0.30 |
| | Random Forest | 25.13 | 0.16 | -0.58 |
| | XGBoost | 23.88 | 0.15 | -0.43 |
| **CO** | FastCTM | 51.84 | 0.90 | 0.01 |
| | Liner Model | 427.67 | 0.03 | 6.38 |
| | Random Forest | 83.25 | 0.08 | 1.32 |
| | XGBoost | 70.06 | 0.06 | 1.10 |
| **O$_3$** | FastCTM | 11.46 | 0.68 | 0.02 |
| | Liner Model | 357.97 | 0.09 | -0.46 |
| | Random Forest | 285.16 | 0.19 | -0.21 |
| | XGBoost | 291.58 | 0.15 | -0.22 |

Annually, the daily air quality typically exhibits similar fluctuations to those in other years, which can be primarily
attributed to the cyclical nature of meteorological conditions and pollutant emission patterns. The FastCTM model was
trained using a comprehensive dataset spanning five years, from 2018 to 2022. In light of this, it was crucial to rule out the
possibility that the model was merely reproducing historical averages during the test year of 2023. To this end, the daily
national average concentrations of PM$_{2.5}$ and O$_3$ in 2023, as predicted by FastCTM, were meticulously compared with
those simulated by CMAQ in the same test year, as well as with the CMAQ forecasts from the training years of 2018-2022.
As illustrated in Figure 5, the predictions made by FastCTM in 2023 align more closely with the actual CMAQ forecasts
for that year, with $R^2$ = 0.94 and 0.72, respectively, for PM$_{2.5}$ and O$_3$, rather than with the forecasts generated from the
training data of 2018-2022, with $R^2$=0.54 and 0.59. The NMB was also lower between FastCTM and CMAQ for the same
year, 2023. These results not only validate the adaptive learning capabilities of the FastCTM model but also indicate that
the model is not using a simplistic approach of averaging concentrations from the previous five years based on time of day.
Hourly time series plots of air pollutant concentrations (Figure S6 in the SI) further demonstrate that FastCTM appears to
incorporate real-time meteorological feedback, adjust for shifts in emission patterns, and leverage its learned relationships
to provide more accurate and contemporaneous predictions.

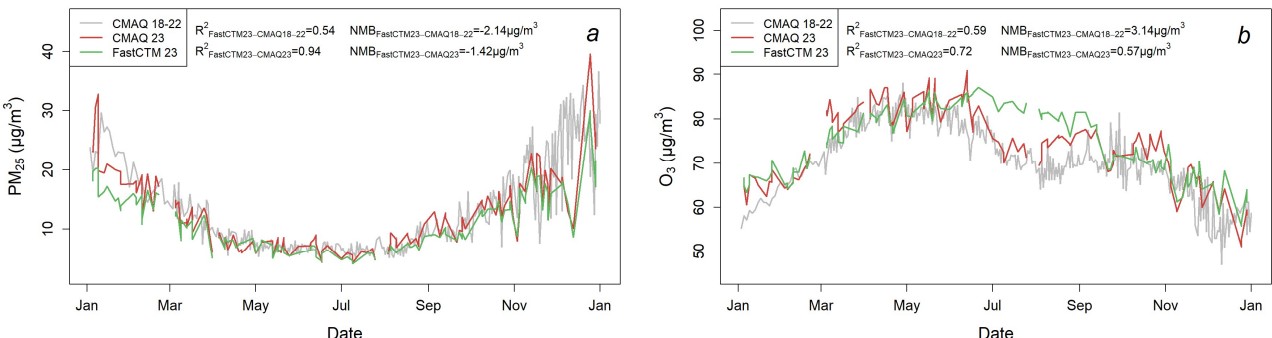

**Figure 5: The daily FastCTM forecasts compared with CMAQ forecasts, respectively, in the training period of 2018-2022 and the evaluation period of 2023 for (a) PM₂.₅ and (b) O₃. The gaps for FastCTM and CMAQ in 2023 are due to data unavailability these days.**

**3.2 Sensitivity Analysis with FastCTM**

The FastCTM model was trained with 5-year meteorological and air quality simulations by WRF-CMAQ. These simulations used an emission inventory that was identical for every year. In this condition, the FastCTM model has learned the relationships between the air quality and varied meteorology with fixed emissions input. Considering that the FastCTM model has exhibited high accuracy in an independent evaluation year 2023, when new meteorological fields are fed into FastCTM, the deep learning model should be able to simulate responses of air pollutant concentrations to meteorological variables. However, for the response of air pollutant concentrations to emissions, the training data do not contain relationships between inter-annual varied emissions and air quality under the condition of the same annual meteorological fields. Therefore, it is less expected for FastCTM to simulate reliable and correct response relationships between emissions and air quality. To validate these analyses, we calculated the sensitivities of simulated air pollutant concentrations to changes in meteorological variables and emissions.

**3.2.1 Response of Air Pollutant Concentration to Meteorology**

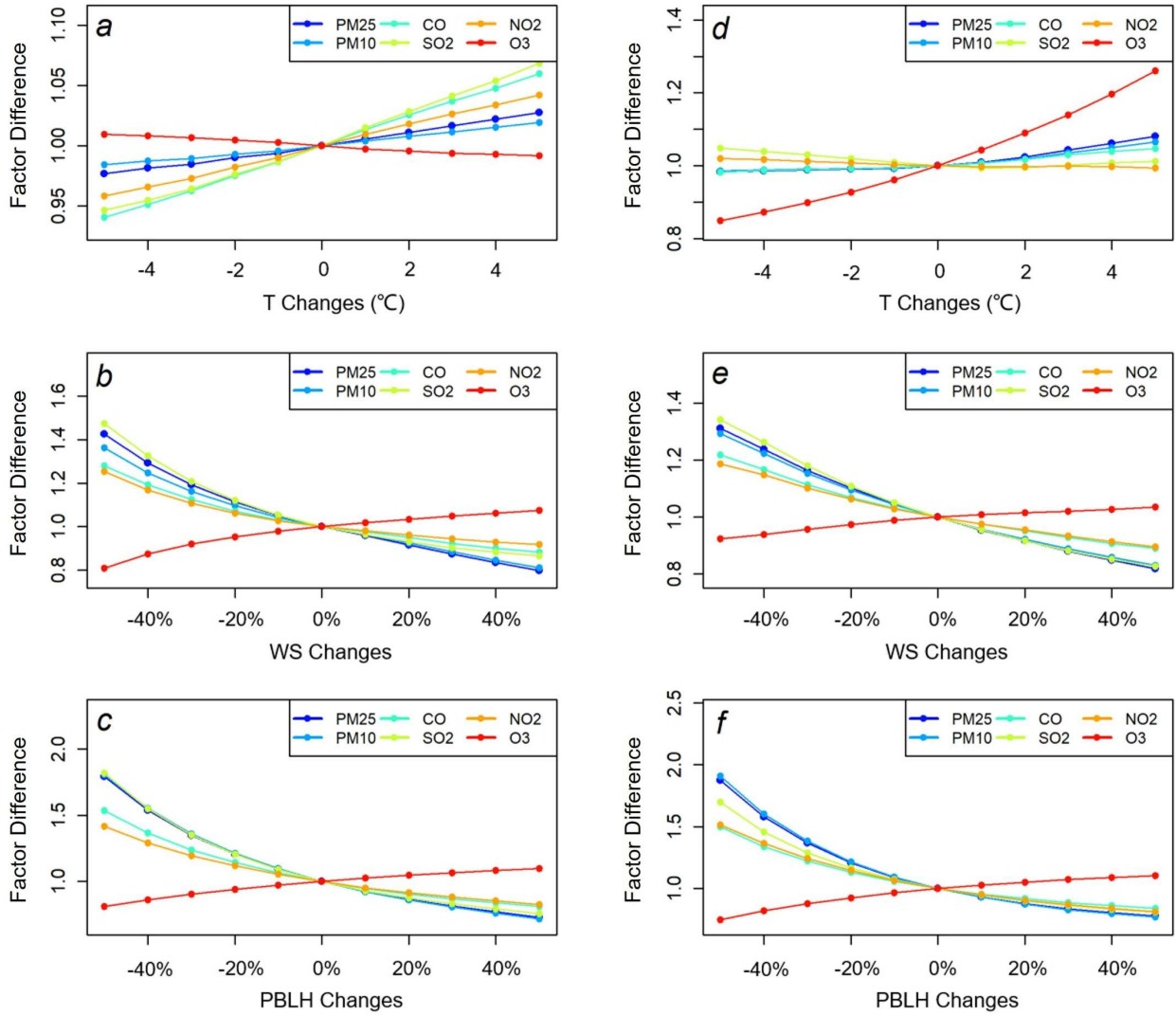

**Figure 6: The FastCTM predicted air pollutant percentage changes in response to changes of T, WS, and PBLH in Beijing on**
**January 2nd (a-c respectively in the left column) and August 1st (d-f respectively in the right column), 2023. The air pollutant**
**concentrations are relative to those at the baseline meteorological conditions.**
The responses of six criteria pollutants to meteorological changes simulated by FastCTM are evaluated as exhibited in
Figure 6. For ground-level temperature (T) elicited a distinct response in $O_3$ concentrations compared to the other five
criteria pollutants. $O_3$ concentrations have slight negative responses to T in January, as shown in Figure 6a, which is
probably because higher temperatures increase $NO_x$ emissions, enhancing dilution. $O_3$ concentrations had the strongest
positive responses in August among six pollutants, which is consistent with previous observation-based studies (Flaum et
al., 1996). The $O_3$ had larger sensitivities when the air temperature was higher. The gaseous pollutants of CO, $NO_2$, and
$SO_2$ show the strongest positive response to temperature, which could be caused by the shift of chemical equilibrium
towards the higher release of these gaseous pollutants (Bassett and Seinfeld, 1983; Cox, 1982). The particulate matter
pollutants, especially $PM_{10}$, have the weakest responses among six pollutants. Considering that there are dominating
proportions of chemically inert species in particulates, the weak responses of PM$_{2.5}$ and PM$_{10}$ are expected.
For the wind speed and PBLH, the responses of pollutants have similar patterns for the same pollutant. First, O$_3$
concentrations exhibited patterns opposite to other pollutants both in January and August. Higher wind speed would
increase the dispersion and transport of air pollutants (Feng et al., 2015; Lv et al., 2017), resulting in lower pollution levels,
so concentrations decrease as wind speed increases, except for O$_3$. The contradictory response of ozone and particulate
matter concentrations to PBLH is consistent with the analysis results of multiple-year observations (Liu and Tang, 2024).
Theoretically, the air pollutant concentrations should exhibit an inverse relationship between air pollution concentrations
and PBLH. The actual air pollutant concentration changes simulated by FastCTM generally fit the theory that there are
negative nonlinear effects with increasing PBLH. Meanwhile, the sensitivity is stronger when the PBLH is lower (Figures
6e and 6f), which is consistent with previous observation-based analysis (Wang et al., 2019; Su et al., 2020). The totally
different relationship of O$_3$ to wind speed and PBLH compared to other pollutants could be due to its high dependence on
chemical precursors, such as NO$_x$ and VOC. Concentrations of these precursors could have an inverse relationship with O$_3$
at specific locations. The FastCTM model itself is trained with multi-year CMAQ simulations, indicating that it is
preconditioned on varied meteorological fields with the same atmospheric physical and chemical rules. Therefore, the
sensitivity of air quality simulations to meteorological variations could be well learned, especially with the discipline-based
model FastCTM.
**3.2.2 Response of Air Pollutant Concentration to Emission**
The sensitivity analysis with a "brute force" method can be carried out with the FastCTM model quickly due to its high
computational efficiency on GPU. The responses of PM$_{2.5}$ concentrations to doubled emissions of SO$_2$, NOx were explored
in the winter month of January 2023 (Figure 7). For doubled NOx, the PM$_{2.5}$ concentrations exhibited positive responses
in most areas of China as shown in Figure 7a. The largest increases occurred in North China, Heilongjiang province in
Northeast China, the Yangtze River Delta, and Sichuan province. In these places, the NOx emissions are relatively large.
For doubled SO$_2$, PM$_{2.5}$ concentrations increased in almost all of China as shown in Figure 7b. The response was larger in
North China, Northeast China and the Sichuan basin. The PM$_{2.5}$ responses simulated by FastCTM were generally consistent
with previous studies (Li et al., 2022).

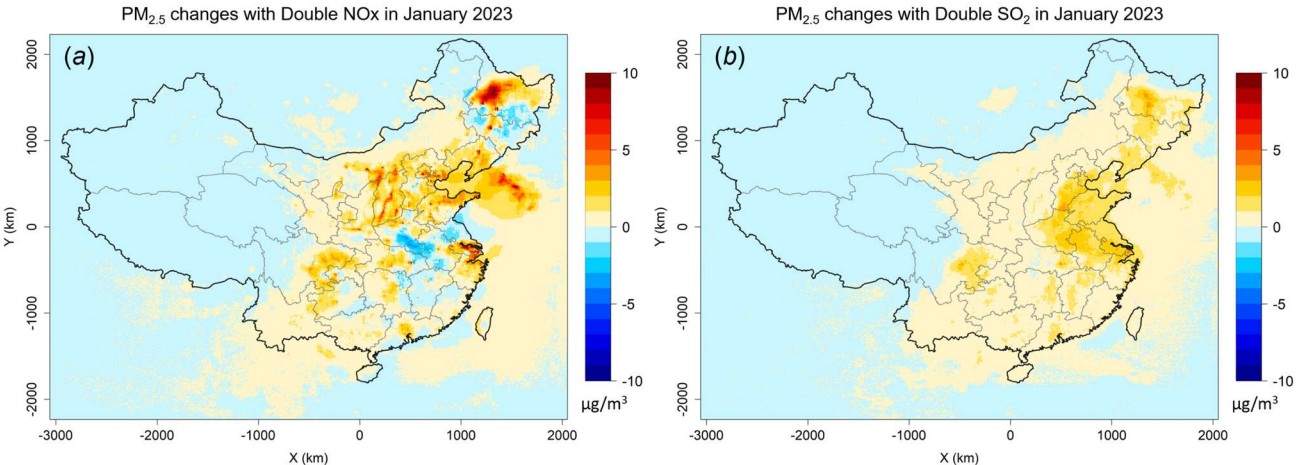


**Figure 7: Average predictions of PM$_{2.5}$ concentrations in 5 lead-days with doubled emissions in January 2023. Panel (*a*) refers**
**to predictions with doubled NOx, and panel (*b*) refers to double SO$_2$.**
As for ozone, its responses to doubled NOx and VOC are explored as shown in Figure 8. For NOx emission, decreases in
O₃ concentrations in polluted regions like North China, the Yangtze River Delta, and other highly industrial regions are
well captured by FastCTM. The response is reasonable considering that these regions are generally abundant with NOx
emissions and at VOC-limited conditions. Doubling VOC emissions leads to a significant decrease in O₃ (Figure S14 in
the supplementary material), since increased VOC would consume more O₃ in these regions. The spatial patterns of O₃
responses to NOx and VOC are similar to a previous deep learning study trained by emission-controlled simulation data
(Xing et al., 2022). However, due to the complex speciation of VOC emissions that is simplified in the FastCTM,
uncertainties for responses of O₃ to VOC should be noted.

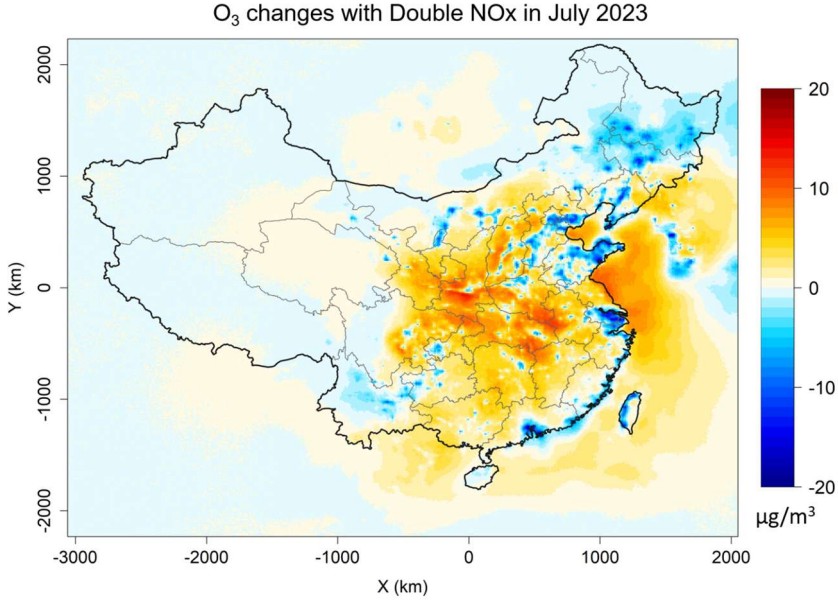


**Figure 8: Average predictions of hourly O₃ concentrations in 5 lead-days with doubled NOx emissions in July 2023.**
The sensitivities of FastCTM simulations to emission interventions were contrasted with those of CMAQ. Specifically,
CMAQ was employed to simulate 11 emission scenarios over the two-month periods of January and July 2019 in Southwest
China (Huang et al., 2022). The alterations in emissions relative to the base case are presented in Table 1. Among these
scenarios, 10 involved reduced emissions of major species, with only the no-control scenario exhibiting increased emissions.
Utilizing the identical emissions and meteorological data, FastCTM also conducted simulations, which were then compared
to those of CMAQ. For the 11 scenarios in question, the changes in air pollutant concentrations relative to the base case at
the locations of 139 national air quality monitoring stations (Figure S15 in the SI) were extracted and compared in the
winter month of January 2019 (Figure 9a) and in the summer month of July 2019 (Figure 9b). The results indicated that,
overall, the FastCTM simulations due to emissions changes were in good agreement with those of CMAQ, as reflected in
two aspects. The correlation coefficient R values are around 0.9 for SO₂, NO₂, and O₃ in both summer and winter months.
For PM₂.₅ and PM₁₀, FastCTM exhibited higher consistency with CMAQ in July than in January, with R values around 0.6
for most cases. For CO, FastCTM has much better performance in January than in July, with R values of approximately
0.8 and 0.2. Considering that CO concentration changes are mostly due to physical dispersion and transport, the decreased
performance is probably due to increased vertical mixing in summer, which is not fully represented in the 2D scheme of
FastCTM. Specifically, in January 2019, except for NO₂, FastCTM responded to emission changes with an interquartile
range (IQR, 25% - 75% percentile) similar to that of CMAQ (Figure S16). In July 2019, as depicted in Figure S17, all the
criteria pollutants except CO demonstrated a comparable degree of response to emission reductions.
**Table 2**. The emission change details of the emission scenarios

| Scenario | abbreviation | Sector | NO$_x$ | VOCs | SO$_2$ | CO | PM$_{2.5}$ | PMC |
|---|---|---|---|---|---|---|---|---|
| **nocontrol** | **NCtrl** | Industrial | 30% | 30% | 30% | 30% | 30% | 30% |
| | | Traffic | 20% | 20% | 20% | 20% | 20% | 20% |
| **medianX** | **MedX** | Industrial | -36% | -35% | -48% | -23% | -9% | -9% |
| | | Traffic | -40% | -10% | 0 | -26% | -10% | -10% |
| **medianY** | **MedY** | Industrial | -26% | -20% | -38% | -13% | -4% | -4% |
| | | Traffic | -30% | 0% | 0 | -16% | -5% | -5% |
| **medianZ** | **MedZ** | Industrial | -36% | -10% | -48% | -23% | -9% | -9% |
| | | Traffic | -40% | 0% | 0 | -26% | -10% | -10% |
| **median-3** | **Med-3** | Industrial | -10% | -10% | -18% | 0 | 0 | 0 |
| | | Traffic | -10% | 0% | 0 | 0 | 0 | 0 |
| **median-2** | **Med-2** | Industrial | -16% | -20% | -28% | -3% | 0 | 0 |
| | | Traffic | -20% | 0% | 0 | -6% | 0 | 0 |
| **median-1** | **Med-1** | Industrial | -26% | -35% | -38% | -13% | -4% | -4% |
| | | Traffic | -30% | -10% | 0 | -16% | -5% | -5% |
| **median0** | **Med0** | Industrial | -36% | -50% | -48% | -23% | -9% | -9% |
| | | Traffic | -40% | -20% | 0 | -26% | -10% | -10% |
| **median+1** | **Med+1** | Industrial | -46% | -65% | -58% | -33% | -19% | -19% |
| | | Traffic | -50% | -30% | 0 | -36% | -20% | -20% |
| **median2030** | **Med30** | Industrial | -55% | -70% | -80% | -40% | -40% | -40% |
| | | Traffic | -60% | -40% | 0 | -40% | -40% | -40% |
| **median2035** | **Med35** | Industrial | -80% | -80% | -90% | -60% | -50% | -50% |
| | | Traffic | -80% | -60% | 0 | -60% | -50% | -50% |



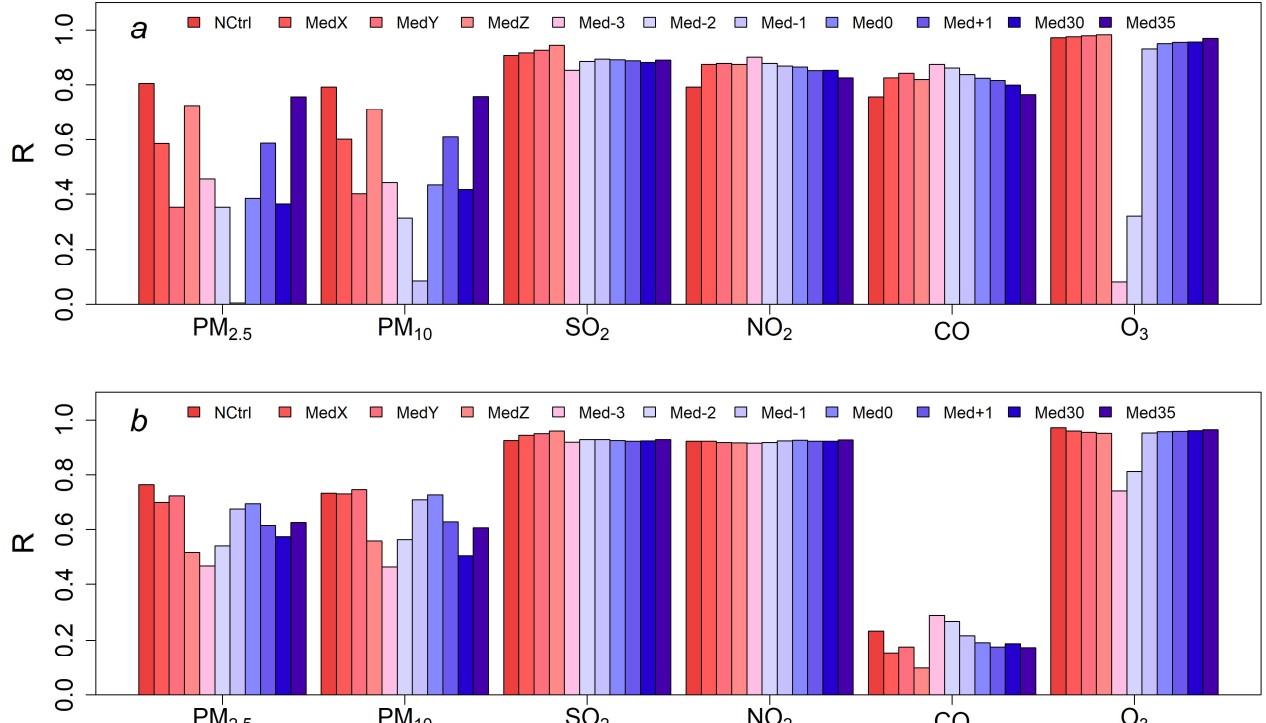

**Figure 9: Correlation coefficient R for responses of FastCTM and CMAQ to different emission scenarios and different air pollutants in January 2023 (panel *a*) and July 2023 (panel *b*).**

FastCTM model used a principles-constrained formulation framework. As shown in Eq.4, atmospheric chemical reactions are in the Atkinson form, which independently estimates the reaction rate from meteorological conditions and polynomials of reactant concentrations in multiple powers. The principle-based formulation should be the reason for the relatively significant and reasonable response simulations of $PM_{2.5}$ and $O_3$ to precursor emissions, even though the FastCTM itself is not trained by emission-controlled CMAQ scenario simulations. The remaining uncertainties should be attributed to the reason that FastCTM only considered environmental chemical reactants in part, compared to that of the CMAQ model (Binkowski and Roselle, 2003).

## 3.3 Internal Processes Analysis with FastCTM

The FastCTM is a principles-guided deep neural network to individually simulate the dominant atmospheric physical and chemical processes as defined in Eq.1. The processes are calculated numerically with critical parameters describing the processes being estimated by deep learning encoders. The hourly concentration changes equal the sum of the changes produced by each process. Figure 11 depicts an example during the nighttime of January 13, 2023, when hourly $PM_{2.5}$ concentration changes significantly. Between the two hours of 18:00 and 19:00, hourly $PM_{2.5}$ concentrations change markedly in neighbouring areas of Shandong, Hebei, and Henan provinces as shown in the red rectangle (denoted as Area A hereafter) in Figure 11c. In this example, strong northern wind prevails, leading pollutants to move southward. For $PM_{2.5}$ concentration changes caused by primary emissions (Figure 8d), it is determined by the primary emission and the mixing volumes determined by PBLH. $PM_{2.5}$ changes are mostly determined by the transport process (Figure 11e) as its spatial pattern most closely resembles total $PM_{2.5}$ concentration changes. In the transport process, air pollutants move from one area to another, determined by the wind fields as shown in Eq.4. When the northern clean air prevails as in Area A, changes should be negative in the upstream direction and positive in the downstream direction. The transport process simulated by

FastCTM sticks to this pattern. As known to us, the diffusion process will bring pollutants from a region of high
concentration to one of low concentration. Its contribution is low as shown in Figure 11f, which is reasonable considering
the relatively large grid cell size of 12km and short simulation period of 1 hour. PM$_{2.5}$ concentration changes caused by the
diffusion process constituted a small proportion compared to other processes. The activities of chemical reactions are
determined by both meteorological conditions and related precursor concentrations. PM$_{2.5}$ contribution changes between
T1 and T2 caused by chemical reactions are lower in the areas to the north of Area A because the cold and clean air in this
area is not favourable for chemical reactions. The deposition is the dominant process that led to PM$_{2.5}$ concentration
reductions where regional transport was not significant. In general, deposition rates were proportional to PM$_{2.5}$
concentrations as shown in Figure 8h (Davis and Swall, 2006). It should be noted that FastCTM simulated air quality in a
2-D domain rather than in 3-D. The deposition may also include the vertical transport of air pollutants to the upper air
above PBL (Zhao et al., 2020).

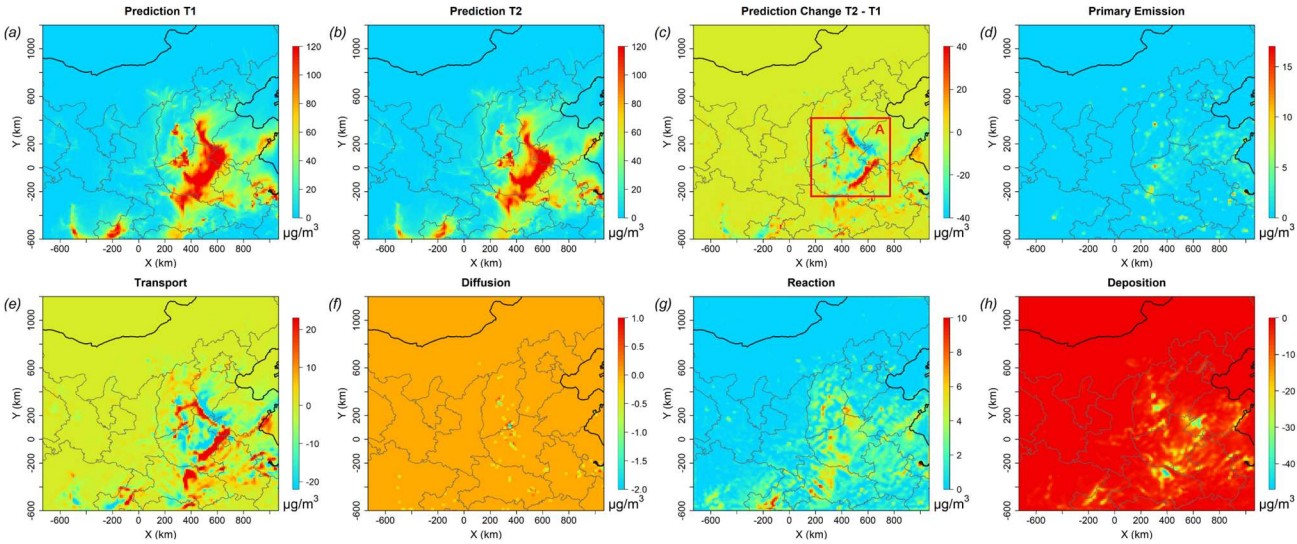

**Figure 10: An example of the PM$_{2.5}$ concentration at T1 (18:00, panel *a*) and T2 (19:00, panel *b*) on January 13, 2023 (with the**
**forecast leading time of 42 hours) and hourly changes (panel *c*). Changes caused by each of the five dominant processes are**
**depicted in panels *d-h*.**
Simulated contributions of five major processes to hourly PM$_{2.5}$ concentration changes are compared between FastCTM
and CMAQ at 139 stations (Figure S15) in the Sichuan-Chongqing region from October 12, 2024, to October 16, 2024, as
shown in boxplots of Figure 11. Overall, the simulation results of the process contributions by FastCTM and its parent
model CMAQ were relatively consistent. Higher degrees of consistency were found in simulations of emissions, advection
processes, and diffusion processes between the two models. Contributions from chemical reactions of FastCTM exhibited
overestimation compared to CMAQ, while contributions from deposition were underestimated. The differences in the
simulated deposition and reaction contributions between the two models could be due to incomplete representation of
influencing factors, given the complexity of the two processes. In general, the consistency between the two models provides
confidence in the reliability of FastCTM for simulating and understanding the complex interplay of atmospheric processes
that govern PM$_{2.5}$ levels.

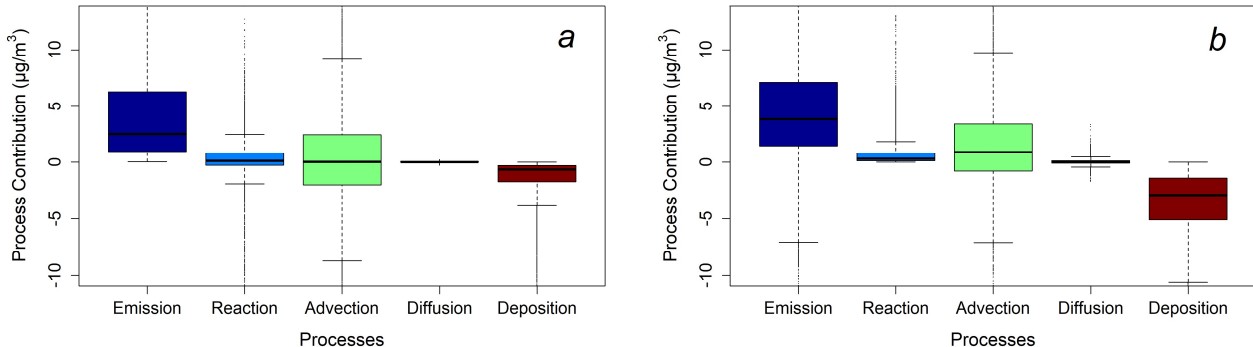

**Figure 11: Boxplots of hourly PM₂.₅ contribution changes from five major atmospheric processes at 139 evaluation stations from October 13, 2024, to October 16, 2024, simulated by (a) CMAQ and (b) FastCTM.**

## 4 Discussions

### 4.1 Model Accuracy and Uncertainty

One debatable concern is the accuracy of neural network (NN)-based components in integrated chemical transport models (CTMs) and the potential for amplified uncertainty when coupling multiple NN modules. Literature precedent suggests that individual NN emulators may exhibit lower accuracy compared to traditional physical parameterizations, but their integration could introduce unexplained uncertainties. This is a valid consideration that aligns with broader discussions in Earth system modeling about the trade-offs between computational efficiency and physical fidelity (Irrgang et al., 2021). In FastCTM, we address this by adopting a principle-informed modular design where each module (transport, chemistry, deposition, etc.) is constrained by governing physical/chemical equations (e.g., Eqs. 3-14). This distinguishes it from unconstrained "black-box" NN models, as each process is guided by known atmospheric dynamics. For example, the transport module explicitly enforces mass conservation via upwind schemes (Eqs. 5-7), and the chemical reaction module links reaction rates to meteorological conditions (Eq. 12) based on kinetic theory. Our evaluation shows that FastCTM maintains high consistency with CMAQ across 119-hour forecasts (Section 3.1), with $R^2$ values exceeding 0.8 for most pollutants, indicating that physical constraints effectively mitigate accuracy losses.

However, we acknowledge that uncertainty can accumulate when coupling modules, particularly for species involved in complex multi-process interactions due to limited chemical constraints in our current training datasets(e.g., $NH_4^+$, Section 3.1). This is partly due to simplifications in FastCTM's chemical mechanism, which omits some aerosol thermodynamics included in CMAQ. Future work will reduce such uncertainties by incorporating additional species (e.g., VOCs) and refining process formulations by adding CMAQ's integrated process rate (IPR) data for supervised training of individual modules.

### 4.2 Choosing Neural Network Components over Traditional Parameterizations

One question might arise about the utility of replacing non-bottleneck CTM components (e.g., deposition) with NN solvers, given the argument that traditional parameterizations may already be accurate and fast. This highlights a critical design choice in FastCTM: balancing computational efficiency with fidelity to the parent model (CMAQ).

It is important to note that even non-bottleneck components in traditional CTMs can benefit from NN acceleration in integrated simulations. For example, CMAQ's deposition module, while not a primary computational burden, relies on

parameterizations based on similarity theory and limited flux measurements (Janhäll, 2015), which may oversimplify complex surface-atmosphere interactions (e.g., vegetation-specific uptake). NN-based parameterizations have shown promise in improving such processes. Silva et al. (2019), for instance, developed a deep learning model for ozone dry deposition that outperformed traditional schemes in independent validation. In FastCTM, the deposition module (Eq. 14) leverages NN to capture nonlinear relationships between meteorology (e.g., wind speed, land cover) and deposition rates, while retaining compatibility with CMAQ's output.

Moreover, FastCTM's modular architecture allows flexible integration of traditional parameterizations as an option. For example, users could replace the NN-based deposition module with CMAQ's original parameterization if higher fidelity to that specific process is prioritized. This hybrid approach addresses concerns about unnecessary replacement of robust components while retaining the overall speed advantage of NN for bottleneck processes (e.g., chemical reactions, which dominate CTM runtime; Xia et al., 2024).

### 4.3 Beyond "Black Boxes": Interpretability and Error Identification

A central goal of FastCTM is to advance beyond opaque deep learning models by enabling process-level interpretability, addressing concerns about error attribution. Traditional "black-box" NN models obscure how individual processes contribute to predictions, hindering error analysis. In contrast, FastCTM's modular design quantifies hourly contributions from transport, diffusion, emissions, chemistry, and deposition separately (Section 3.3), allowing targeted identification of error sources. For example, in the January 2023 pollution episode (Figure 10), transport was found to dominate $PM_{2.5}$ concentration changes, while deposition acted as a secondary sink This process-level attribution aligns well with CMAQ's process analysis (Figure 11), ensuring that uncertainties are traced to specific physical processes rather than being attributed to arbitrary model behavior.

We anticipate that incorporating abundant CMAQ's integrated process rate (IPR) data for supervised training of individual modules will further refine the FastCTM's process level predictions. However, a comprehensive process-oriented error analysis that would further enhancing interpretability, for instance isolating and quantifying whether transport or chemistry drives urban-rural accuracy discrepancies, requires long-term process simulations and systematic perturbations plus observational datasets (e.g., tracer experiments) to validate specific processes predictions from both CMAQ and FastCTM.

### 4.4 Limitations and Future Directions

FastCTM's current limitations include simplified vertical dynamics (2D boundary layer representation) and incomplete chemical mechanisms, which affect performance during vigorous daytime mixing (Section 3.1). A future extension to a 3D framework will improve representation of vertical transport and in-cloud chemistry. Additionally, while FastCTM efficiently reproduces CMAQ simulations, it does not claim superiority over traditional CTMs across all scenarios; rather, it serves as a complementary tool for applications requiring rapid simulations (e.g., ensemble forecasting, emission scenario screening).

By addressing these limitations and engaging with ongoing debates about NN integration in atmospheric modeling, FastCTM aims to bridge the gap between computational efficiency and physical rigor, providing a flexible framework for air quality research and management.

***Data availability.*** The land use and land cover data are available at the Data Sharing and Service Portal of the Chinese

Academy of Science (http://data.casearth.cn/en/sdo/detail/5ebe2a9908415d14083a4c24). The CTM simulation data and
source code files of the exact version used to produce the results used in this paper are available at
https://doi.org/10.5281/zenodo.13757211 on Zenodo (Lyu, 2024). The configuration files for running models of WRF
v3.4.1 and CAMQ v5.0.2 are also available at https://doi.org/10.5281/zenodo.5152621 (Hu, 2021).
*Author contributions.* BL and YH conceived the study. BL developed the model and codes. RH and XW contributed the
CTM simulation data. BL and RH collected the observation data. BL analyzed data and wrote the paper with contributions
from YH, RH, WW, and XW. RH managed the project.
*Competing interests.* The authors declare that they have no conflict of interest.
*Acknowledgements.* This research has been in part supported by the AiMa R&D Project (R#2016-004) of Hangzhou AiMa
Technologies. The findings in this research do not necessarily reflect the views of the sponsors.

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
