# Peer review of "principle-informed neural network for air quality simulations"

_Geoscientific Model Development, 2024_

## Author Comment (AC1)

**Point-to-Point Responses to Reviewer's Comments**

We would like to thank for reviewer's thoughtful comments on our manuscript.

The authors describe a new, neural network-based reduced-order model of atmospheric chemistry and transport (FastCTM) which has been trained using an extensive dataset of output from CMAQ. FastCTM uses a novel and interesting approach, building physics-informed networks for five separate operators. The authors show that FastCTM is able to reproduce the general patterns of concentrations calculated by CMAQ for an out-of-training-data year (2023), and that the sensitivities of FastCTM to key meteorological variables or nation-wide changes in emissions mostly follow expected patterns. If FastCTM can be shown to be reliable in policy-relevant contexts then it could be a very useful tool.

This approach to modelling is interesting, and this methodological advance has the potential to significantly accelerate air quality scenario analysis. A CTM which can respect key physical constraints (e.g. mass conservation) while also accurately reproducing the effect of different perturbations to emissions and meteorological fields would have great value. However, the manuscript as written does not quite live up to this promise. Along with some minor concerns, the key challenge is that the authors do not show evidence that this new model can fulfil the roles of a CTM and produce accurate results for one of the most common use cases (i.e. understanding the effects of different perturbations). I explain this concern in more detail below, and until this concern is addressed I do not believe the manuscript should be accepted for publication by GMD.

*Major comments*

The most significant concern relates to the validation/evaluation of the model. The authors appear to have trained the five physical operators based on several years of output from the CMAQ chemistry transport model. While I have some questions regarding the training process, I will take it as read for the moment that the training was done in such a way as to avoid overfitting. However, the verification of the model rests on its ability to predict, from the 2018-2022 data, the performance in 2023. This approach is inadequate for two reasons. First, the authors do not compare the performance of the model to simpler approaches with the same data such as generalized additive models, gradient boosting, or linear regression with land use (see e.g. Wong et al., 2021 and Cheng et al., 2021). Without such a comparison to evaluate how such models would have performed in predicting 2023, it is difficult to say what the magnitude of FastCTM's advance is. This is exacerbated by the relatively shallow quantitative assessment in section 3.1.1. RMSE and R2 values are provided, but it is not clear how these were calculated; given that these are calculated as a function of time, are these calculated based on the difference in each of the 158,742 grid cells between CMAQ and FastCTM? A deeper analysis which investigates how model performance varies between (e.g.) rural and urban areas, coastal and inland areas, winter and summer, and so on would provide a much more robust test of the model's ability to predict the effect of changing meteorology. This could be informed by (e.g.) taking the difference of FastCTM for 2023 against CMAQ for 2023, and comparing that to the difference between CMAQ for 2023 and the average of CMAQ from 2018 to 2022. This would at least demonstrate whether FastCTM provides more explanatory power for the mean atmospheric state than taking the average concentration from the previous five years.

**Response**: Thanks. We agree with the review's point that the current analysis does not adequately demonstrate that FastCTM has actual capabilities to simulate air quality changes by learning and representing physical and chemical processes. As the reviewer suggested, more tests, comparisons, and analyses are performed.

(1) For comparing with simpler machine learning models, we tested three models of Linear Regression, Random Forest, and XGBoost with the same train and test dataset. The results are added in the manuscript as follows,

*To validate FastCTM model, three land use regression (LUR) models were constructed, namely the linear regression model, the random forest model (with the number of trees set at 500), and the XGBoost model (with the booster specified as gbtree). These LUR models were developed using the same input meteorological data, emission, and geophysical variables. When compared with the FastCTM model, the performance of the LUR models was found to be significantly inferior (as demonstrated in Figure S10 – S12 in the SI). This outcome is, in fact, anticipated when we consider the complex nature of air quality dynamics. Air quality is not a static entity, but it varies both spatially and temporally. For instance, the transport of air pollution is a highly dynamic process that hinges on wind fields and air pollution concentrations in a reciprocal manner. The wind direction and speed dictate the trajectory along which pollutants travel, while the existing pollutant concentrations in different regions influence the overall dispersion and mixing patterns. LUR models, which predominantly rely on local input data (Wong et al., 2021; Cheng et al., 2021), struggle to capture these intricate, non-local interactions. They lack the capacity to account for the far-reaching effects such as wind-driven pollutant transport and the consequential changes in air quality over larger geographical areas.*

The supplementary Figures demonstrate the performances of three machine learning models are displayed in the following part.

[Figure]

*Figure S10: The evaluation performances of linear regression forecasts against CMAQ forecasts in 2023. Panel (a) and (b) respectively show RMSE values of criteria pollutants and the PM$_{2.5}$ components. Panel (c) and (d) respectively show R$^2$ values. It should be noted that the RMSE value of CO corresponds to the right axis in panel (a).*

[Figure]

*Figure S11: The evaluation performances of random forest forecasts against CMAQ forecasts in 2023. Panel (a) and (b) respectively show RMSE values of criteria pollutants and the PM$_{2.5}$ components. Panel (c) and (d) respectively show R$^2$ values. It should be noted that the RMSE value of CO corresponds to the right axis in panel (a).*

[Figure]

*Figure S12: The evaluation performances of XGboost forecasts against CMAQ forecasts in 2023. Panel (a) and (b) respectively show RMSE values of criteria pollutants and the PM$_{2.5}$ components. Panel (c) and (d) respectively show R$^2$ values. It should be noted that the RMSE value of CO corresponds to the right axis in panel (a).*

(2) As for the calculation process for the metrics of RMSE and R2, they are elaborated in section 2.5 Model Evaluation, as follows.

*The metrics of root mean square error (RMSE) and coefficient of determination ($R^2$) were calculated daily in each of 119 leading hours on the difference in each of the 158,742 grid cells between CMAQ and FastCTM. Therefore, 119 static values for each metric of $R^2$ and RMSE were obtained on each day of the independent test year of 2023. The statistical values on each day are then averaged for the same leading hour for comparison.*

(3) As the reviewer suggested, evaluations of FastCTM compared to CMAQ in rural/urban, inland/coastal areas, and cold/warm seasons are further performed. Generally, they have similar performances in comparative areas or seasons. However, FastCTM exhibited lower correlations in urban areas and coastal areas. In urban areas, emission sources and chemical processes are more complex than that in rural areas, making it harder for FastCTM to simulate due to its 2D setting and fewer chemicals considered than CMAQ. It is also true for FastCTM's performance in coastal areas, where meteorological conditions are more varied in time. Related discussions and results are added in section 3.1.1. and in the supplementary material as follows.

*Defining the warm season as the months from April to September and the winter and cold season as the remaining months, the FastCTM model exhibited comparable performances. As shown in Figure 4 (with detailed information in Figure S7 in the SI), the coefficient of determination $R^2$ values for the six criteria pollutants were 0.82, 0.8, 0.8, 0.82, 0.91, and 0.7 in the warm season, and 0.8, 0.79, 0.78, 0.83, 0.88, and 0.68 in the cold season, respectively. To assess the performance variations of FastCTM across different spatial locations, comparative evaluations were carried out in urban and rural areas as well as in inland and coastal regions. Generally, FastCTM demonstrated slightly higher accuracies in rural areas compared to urban areas (as presented in Figure S8 in the SI). This outcome is reasonable given the more intricate emission and chemical processes prevalent in urban settings (Guo et al., 2014). Similarly, FastCTM exhibited comparable performances in inland areas to those in coastal areas, with the exception of $PM_{2.5}$ and $PM_{10}$ (Figure S9 in the SI).*

[Figure]

*Figure 1: The mean evaluation $R^2$ values for all 119 leading hours of FastCTM forecasts in warm/cold seasons, rural/urban areas and coastal/inland areas.*

[Figure]

*Figure S2: The evaluation performances of random forest forecasts against CMAQ forecasts in warm season of 2023. Panel (a) and (b) respectively show RMSE values of criteria pollutants and the PM$_{2.5}$ components of. Panel (c) and (d) respectively show R$^2$ values. It should be noted that RMSE value of CO corresponds to the right axis in panel (a).*

[Figure]

*Figure S3: The evaluation performances of FastCTM forecasts against CMAQ forecasts in rural and urban areas in 2023.. Panel (a) and (b) respectively show RMSE values of criteria pollutants and the PM$_{2.5}$ components of. Panel (c) and (d) respectively show R$^2$ values. It should be noted that RMSE value of CO corresponds to the right axis in panel (a).*

[Figure]

*Figure S4: The evaluation performances of FastCTM forecasts against CMAQ forecasts in inland and coastal areas in 2023. Panel (a) and (b) respectively show RMSE values of criteria pollutants and the PM$_{2.5}$ components of. Panel (c) and (d) respectively show R$^2$ values. It should be noted that RMSE value of CO corresponds to the right axis in panel (a).*

(4) As the reviewer has kindly pointed out, FastCTM may have taken average pollutant concentration from five-year training data in 2018-2022. In order to verify if FastCTM was able to predict air quality based on given meteorological conditions and emissions, daily average FastCTM simulation in the fifth leading day (leading hours 96-119) in the test year of 2023 is compared with daily average CMAQ simulations in 2023 and in the training years of 2018-2022. Results revealed that FastCTM forecasts are generally in good correlation with CMAQ forecasts in 2023, rather than that in 2018-2022. It means FastCTM has learned the evolution rules of air pollutant concentrations, instead of just giving average air pollutant concentration according to time of the year. Related results have been added in the manuscript in section 3.1.1, as follows.

*Annually, the daily air quality typically exhibits similar fluctuations to those in other years, which can be primarily attributed to the cyclical nature of meteorological conditions and pollutant emission patterns. The FastCTM model was trained using a comprehensive dataset spanning five years, from 2018 to 2022. It was crucial to rule out the possibility that the model was merely reproducing historical averages during the test year of 2023. The daily national average concentrations of $PM_{2.5}$ and $O_3$ in 2023, as predicted by FastCTM, were similarly compared with those simulated by CMAQ in the same test year, as well as with the CMAQ forecasts from the training years of 2018-2022. As illustrated in Figure 4, it becomes evident that the predictions made by FastCTM in 2023 align more closely with the actual CMAQ forecasts for that year, rather than with the forecasts generated from the training data of 2018-2022. This finding not only validates the adaptive learning capabilities of the FastCTM model but also indicates that the model is not resorting to a simplistic approach of taking the average concentration from the previous five years based on the time of day. Instead, it is likely to incorporate real-time meteorological feedback, adjusting for shifts in emission patterns, and leveraging its learned relationships to provide more accurate and contemporaneous predictions.*

[Figure]

*Figure 5: The timeseries of FastCTM forecasts against CMAQ forecasts..*

Second, and perhaps more importantly, the function of FastCTM is to reproduce the results of high-fidelity CTMs at a fraction of the computational cost – specifically to support air pollution simulations, sensitivity analysis, and internal process analysis (abstract lines 30-32). The comparison to 2023 only tells us that FastCTM can reproduce the general pattern of air pollution in 2023, but does not tell us whether FastCTM will accurately predict the effect of interventions. The sensitivity tests in section 3.2 have no basis for comparison, and are in any case so broad (representing nationwide changes in temperature, PBL height, or emissions) that they are a limited test of the CTM's capabilities. At the very minimum, an evaluation is needed which shows that FastCTM's trends actually match the underlying trends in WRF-CMAQ; this should be straightforward for the emissions cases. Since the

goal of FastCTM is to reduce computational costs, it is critical that FastCTM be shown to be faithful to its parent model for realistic applications such as projecting the impact of a change in emissions. Going further and comparing sensitivities for local or single-sector emissions changes would provide even more powerful proof, and I strongly recommend that the authors consider such a comparison.

Without these kinds of quantitative comparisons I can only judge the model's success based on data such as Figure 3, where I am concerned because the patterns do not – speaking qualitatively – appear to match that well between CMAQ and FastCTM. I am particularly concerned that the model may be mostly reproducing emissions maps and historical scalings, rather than accurately representing chemistry and transport (especially given that transport is 2-D only). A more critical, quantitative analysis of the models strengths and weaknesses would be necessary before I would recommend its use in a scientific or regulatory context.

**Response**: We agree with the reviewer's comment that more analysis is needed to verify FastCTM's capabilities to project the impact of a change in emissions. Since the emissions are the same for each year from 2018-2023, it is not possible to test FastCTM's trends to that of WRF-CMAQ. Instead, we added a comparison between FastCTM and CMAQ under 11 emission scenarios in the winter month of January 2019 and in the summer month of July 2019. The results signified that the FastCTM simulations manifested a high level of concordance with those of CMAQ, which was manifested in two principal aspects. Firstly, similar to CMAQ, the FastCTM model forecasted positive responses to increased emissions in the no-control (NCtrl) scenario and negative responses in the other emission-controlled scenarios. This implies that when emissions were unrestricted and increased, as in the NCtrl scenario, both models detected a corresponding upward trend in pollutant levels. Conversely, in scenarios where emissions were curbed, they both predicted a decline. Secondly, in scenarios characterized by more substantial emission reductions, the FastCTM model simulated a more pronounced decrease in air pollutant concentrations. This is of particular significance as it shows the model's sensitivity to the magnitude of emission interventions. It suggests that the FastCTM model is not only capable of discerning changes in emission scenarios but can also reflect the degree of impact on air quality, thereby reinforcing its reliability and utility in simulating air quality dynamics in tandem with CMAQ. Related results in the manuscript are shown as follows.

*The sensitivities of FastCTM simulations to emission interventions were contrasted with those of CMAQ. Specifically, CMAQ was employed to simulate 11 emission scenarios over the two-month periods of January and July 2019 in Southwest China (Huang et al., 2022). The alterations in emissions relative to the base case are presented in Table 1. Among these scenarios, 10 involved reduced emissions of major species, with only the no-control scenario exhibiting increased emissions. Utilizing the identical emissions and meteorological data, FastCTM also conducted simulations, which were then compared to those of CMAQ. For the 11 scenarios in question, the changes in air pollutant concentrations relative to the base case at the locations of 139 national air quality monitoring stations (Figure S14 in the SI) were extracted and compared in the winter month of January 2019 (Figure 9) and in summer month of July 2019 (Figure 10). The results indicated that, overall, the FastCTM simulations were in good agreement with those of CMAQ reflected in two aspects. First, FastCTM predicted positive responses to increased emissions in the nocontrol (NCtrl) scenario and negative responses to other emission-controlled scenarios just as CMAQ. Second, FastCTM simulated larger air pollutant concentration decrease in those scenarios with higher emission reductions. Specifically,*

in January 2019, with the exception of NO₂, FastCTM responded to emission changes with an interquartile range (IQR, 25% - 75% percentile) similar to that of CMAQ (Figure 9). For NO₂, in the same emission reduction scenarios, FastCTM simulated lower NO₂ values. In the summer month of July 2019, as depicted in Figure 10, all the criteria pollutants except CO demonstrated a comparable degree of response to emission reductions. The comparison suggests that the FastCTM model is not only capable of discerning changes in emission scenarios but can also reflect the degree of impact on air quality, thereby reinforcing its reliability and utility in simulating air quality dynamics in tandem with CMAQ. It should be noted that in both months, FastCTM exhibited slightly larger median values, suggesting its greater sensitivity to emission interventions.

*Table 1*. The emission change details of emission scenarios

| Scenario | abbreviation | Sector | $NO_x$ | VOCs | $SO_2$ | CO | $PM_{2.5}$ | PMC |
|----------|--------------|--------|--------|------|--------|-----|------------|-----|
| nocontrol | NCtrl | Industrial | 30% | 30% | 30% | 30% | 30% | 30% |
|          |             | Traffic | 20% | 20% | 20% | 20% | 20% | 20% |
| medianX | MedX | Industrial | -36% | -35% | -48% | -23% | -9% | -9% |
|          |             | Traffic | -40% | -10% | 0 | -26% | -10% | -10% |
| medianY | MedY | Industrial | -26% | -20% | -38% | -13% | -4% | -4% |
|          |             | Traffic | -30% | 0% | 0 | -16% | -5% | -5% |
| medianZ | MedZ | Industrial | -36% | -10% | -48% | -23% | -9% | -9% |
|          |             | Traffic | -40% | 0% | 0 | -26% | -10% | -10% |
| median-3 | Med-3 | Industrial | -10% | -10% | -18% | 0 | 0 | 0 |
|          |             | Traffic | -10% | 0% | 0 | 0 | 0 | 0 |
| median-2 | Med-2 | Industrial | -16% | -20% | -28% | -3% | 0 | 0 |
|          |             | Traffic | -20% | 0% | 0 | -6% | 0 | 0 |
| median-1 | Med-1 | Industrial | -26% | -35% | -38% | -13% | -4% | -4% |
|          |             | Traffic | -30% | -10% | 0 | -16% | -5% | -5% |
| median0 | Med0 | Industrial | -36% | -50% | -48% | -23% | -9% | -9% |
|          |             | Traffic | -40% | -20% | 0 | -26% | -10% | -10% |
| median+1 | Med+1 | Industrial | -46% | -65% | -58% | -33% | -19% | -19% |
|          |             | Traffic | -50% | -30% | 0 | -36% | -20% | -20% |
| median2030 | Med30 | Industrial | -55% | -70% | -80% | -40% | -40% | -40% |
|          |             | Traffic | -60% | -40% | 0 | -40% | -40% | -40% |

| median2035 | Med35 | Industrial | -80% | -80% | -90% | -60% | -50% | -50% |
|---|---|---|---|---|---|---|---|---|
| | | Traffic | -80% | -60% | 0 | -60% | -50% | -50% |

[Figure]

*Figure 9: Air pollutant concentration changes in terms of base case simulated by CMAQ (subplots of a, c, e, g, i, and k in the first column) and by FastCTM (subplots of b, d, f, h, j, and l in the second column) in January 2019.*

[Figure]

*Figure 10: Air pollutant concentration changes in terms of base case simulated by CMAQ (subplots of a, c, e, g, i, and k in the first column) and by FastCTM (subplots of b, d, f, h, j, and i in the second column) in July 2019.*

*Minor comments*

The description of the five operators does not quite seem to verify that physical constraints are being satisfied but this may simply be a misinterpretation on my part. For example, can you confirm that the method you used to generate the convolution kernels (Eq 5-7) for transport ensures mass conservation? This seems to depend on $C_i$ being in units of molec/cm3 rather than ppbv, but the units of $C_i$ are not

clearly specified.

**Response**: We added descriptions for the model framework in Section 2.3. The unit for all pollutants is μg/m³. We used an upwind-scheme to simulate diffusion and advection processes. For the scheme, masses are conserved. However, FastCTM is not mass conserved, because it also includes other neural network modules such as reaction and deposition. These deep learning modules are learned to minimize the loss function of mean squared error. The revised model description and figures are shown as follows.

*Instead, we use a 1-hour initial pollutant concentration (J=1) to simulate 24-hour air quality pollutants (K=24), to ensure FastCTM is dedicated to learning air quality changes between neighboring two hours as shown in Figure 1a. In other words, at time $t = 0$, FastCTM predicted K-hour air pollutant concentrations of $C_{t=0}, C_{t=1}, ..., C_{t=K-1}$, given the input air pollutant concentration in the previous hour $C_{t=-1}$ and corresponding meteorological data and emissions at time $t = 0,1,...,$ K-1. The unit of concentrations is μg/m³ for all pollutants.*

[Figure]

*Figure 5: (a) General model workflow, and (b) the basic simulator module structure at the time step t of deep learning simulation model FastCTM designed according to Eq.1. Arrows and boxes with different colours represent calculation modules of different atmospheric physical and chemical processes.*

A related concern is that surface layer winds are used and treated conservatively, which neglects the fact of rapid vertical mixing. Can the authors provide evidence that the surface winds (which would be expected to be slower than the mean wind speed in the boundary layer) are accurately predicting pollutant motion? It seems that any model which is designed to predict transport using only the horizontal near-surface winds will underestimate overall transport. Should the model not be using the PBL-averaged horizontal winds in Eq. 4 instead of the surface winds at 10-meter height (lines 83-84)?

**Response**: We appreciate the reviewer's critical observation regarding the use of surface-layer winds (10 m) in FastCTM's transport module and the potential underestimation of pollutant transport. We

agree that PBL-averaged horizontal winds (also called transport winds) could better predict vertical transportation. We are going to apply our FastCTM model to 3D dimensions in the future version. This will enable a more realistic simulation of both horizontal and vertical transport processes. The relevant section of the manuscript has been revised accordingly as follows.

*FastCTM will also extend to 3D dimension to improve its representation for processes such as vertical mixing, vertical wind gradient and in-cloud chemistries.*

It would be helpful to get more detail on how components such as the diffusion encoder were trained. Currently the manuscript states that 5 years of data (2018 – 2022 inclusive) were used in training, but not how the five different models were trained using that data. A naïve assessment would assume that all five sub-models were trained based simply on hour-to-hour pollutant concentrations, but that would suggest that the models were each trying to represent all atmospheric processes simultaneously.

**Response**: The five modules in FastCTM are defined in the form of operator, where operator parameters are estimated, rather than in the form of pure predictor mapping concentrations from one hour to the next. For example, in the diffusion module, FastCTM learns to encode diffusion coefficient $K$ from meteorological conditions before performing an upwind finite difference procedure to solve the diffusion process $\nabla(K\nabla C_i)$. It's also the same for processes such as reaction, advection, and deposition. Therefore, it is impossible for one process to represent all atmospheric processes simultaneously. The independent contribution of each process is depicted in Figure 12 of section 3.3. Each process exhibited its patterns of contribution to hourly air pollutant concentration changes, constrained by the form of the operator in the processes. The related description was added in Section 2.3 Model Training.

*Even though five modules are defined in FastCTM, individual processes are not trained separately. The model was trained as a whole with hour-to-hour air pollutant concentrations, while each process could learn its parameters under the constrains of its dedicated formulation. Specifically, FastCTM was tuned to minimize the loss function $\mathcal{L}$, which was determined to be L2 loss (Bühlmann and Yu, 2003) of the regularized mean squared error (MSE) as shown in Eq. 15.*

Line 172 says that the reaction encoder in Equation 12 "has the same structure as that of reaction and deposition encoder models (Eq. 10)". This is recursive, but also Eq. 10 refers to the diffusion module?

**Response**: This error was revised as follows in the corresponding section,

*Therefore, the reaction rate constant $k$ is simulated using a spatial encoder function* Encoder *as shown in Eq. 12, which has the same structure as that of diffusion encoder modules (Eq. 10).*

On line 194, "We did not use the fixed area as that in the previous studies (Xing et al., 2022)" – can you elaborate? It was not clear to me what this meant.

**Response**: Revised as follows,

*The random sampling tactics would help the model learn inherent physical and chemical principles model rather than just statistical spatiotemporal autocorrelations using data in constant spatial area (Xing et al., 2022). Besides, the spatio-temporal random samples contain varied emissions which would improve FastCTM adaptation to changing emission levels.*

The y-axis labels on Figure 5 say "Percentage", but from context it appears these must really be the factor difference from the baseline (as all cross at 1.0).

**Response**: Revised.

Finally, there are numerous minor grammatical errors (e.g. L14: simulations and managements; L67: interpretations of the FastCTM are also widely vowed; L70: including and major; and so on). This is not important for judgment of the paper's appropriateness for publication, but I recommend the authors take another look at the paper to correct such minor issues.

**Response**: We have read through the manuscript and made a thorough revision paying particular attention to the grammar.

*Citations*

Wong et al., "Using a land use regression model with machine learning to estimate ground level PM2.5." Environ. Pollut., 2021.

Cheng et al. "Influence of weather and air pollution on concentration change of PM2. 5 using a generalized additive model and gradient boosting machine." Atmospheric environment, 2021.

---

## Author Comment (AC2)

**Point-to-Point Responses to Reviewer's Comments**

We would like to thank for reviewer's thoughtful comments on our manuscript.

Lyu et al. put forth the 'FastCTM' model which seems to be a reduced complexity model that discretizes changes in concentrations for 10 air pollution species. Though interesting, the presentation of methods, results, and context of the study needs to be heavily refined before being accepted. The details of the study are currently not sufficient as they stand.

Introduction:

"The air pollutant and species concentrations can be then calculated by solving these complicated equations with numeric methods (Byun and Schere, 2006), which is often time-consuming and requires intense computational resources." --> This thought is not very well flushed out. A single reference from 2006 does not detail at all what makes these models computationally expensive.

**Response**: This sentence is revised with two more related references are added, as follows.

*The air pollutant concentrations can be then calculated by solving these complicated equations with numeric methods (Byun and Schere, 2006), which is often time-consuming (Leal et al., 2017) and require intense computational resources such as high-performance computing (Efstathiou et al., 2024).*

*Leal, A. M., Kulik, D. A., Smith, W. R., and Saar, M. O.: An overview of computational methods for chemical equilibrium and kinetic calculations for geochemical and reactive transport modeling, Pure and Applied Chemistry, 89, 597-643, 2017.*

*Efstathiou, C. I., Adams, E., Coats, C. J., Zelt, R., Reed, M., McGee, J., Foley, K. M., Sidi, F. I., Wong, D. C., and Fine, S.: Enabling high-performance cloud computing for the Community Multiscale Air Quality Model (CMAQ) version 5.3. 3: performance evaluation and benefits for the user community, Geoscientific Model Development, 17, 7001-7027, 2024.*

"Quantifying the contributions of individual processes would provide fundamental explanations for a model's predictions, and therefore is also useful in identifying potential sources of error in the model formulation or its inputs (Liu et al., 2010)." --> I find this introduction quite poor. The authors provide minimal examples of emulating entire CTMs but give no examples of using ML to emulate and replace CTM model components which there are many for chemistry, photolysis, deposition, etc. This needs much greater discussion as it shows a lack of awareness by the authors of what currently exists, below of which are only several examples:

Krasnopolsky, V. M., Fox-Rabinovitz, M. S., and Chalikov, D. V.: New Approach to Calculation of Atmospheric Model Physics: Accurate and Fast Neural Network Emulation of Longwave Radiation in a Climate Model, Monthly Weather Review, 133, 1370–1383, https://doi.org/10.1175/MWR2923.1, 2005.

Kelp, M. M., Jacob, D. J., Lin, H., and Sulprizio, M. P.: An Online-Learned Neural Network Chemical Solver for Stable LongTerm Global Simulations of Atmospheric Chemistry, Journal of Advances in

Modeling Earth Systems, 14, e2021MS002926, https://doi.org/10.1029/2021MS002926, _eprint: https://onlinelibrary.wiley.com/doi/pdf/10.1029/2021MS002926, 2022.

Xia, Z., Zhao, C., Du, Q., Yang, Z., Zhang, M., and Qiao, L.: Advancing Photochemistry Simulation in WRF-Chem V4.0: Artificial Intelligence PhotoChemistry (AIPC) Scheme with Multi-Head Self-Attention Algorithm, https://www.authorea.com/users/816476/articles/1217166-advancing-photochemistry-simulation-in-wrf-chem-v4-0-artificial-intelligence-photochemistry-aipc-scheme-with-multi-head-self-attention-algorithm, 2024.

Zhong, X., Ma, Z., Yao, Y., Xu, L., Wu, Y., and Wang, Z.: WRF–ML v1.0: a bridge between WRF v4.3 and machine learning parameterizations and its application to atmospheric radiative transfer, Geoscientific Model Development, 16, 199–209, https://doi.org/10.5194/gmd16-199-2023, publisher: Copernicus GmbH, 2023.

Silva, S. J., Heald, C. L., Ravela, S., Mammarella, I., and Munger, J. W.: A Deep Learning Parameterization for Ozone Dry Deposition Velocities, Geophysical Research Letters, 46, 983–989, https://doi.org/10.1029/2018GL081049, tex.copyright: ©2018. American Geophysical Union. All Rights Reserved., 2019.

**Response**: Given the suggestive comments from the reviewer, we have added an independent paragraph in the introduction, to analyze related studies and progress, as follows,

*Quantifying individual processes would provide fundamental explanations for a model's predictions, and therefore is also useful in identifying potential sources of error in the model formulation or its inputs (Liu et al., 2010). With this motivation, there are studies dedicated to developing models to learn one specific atmospheric process, i.e. chemical and deposition, in the CTM model. Kelp et al. (2022) developed a neural network chemical solver for stable long-term global simulations of atmospheric chemistry, learned from the GEOS-Chem model. Xia et al. (2024) simulated 74 chemical species and 229 reactions following the SAPRC-99 mechanism with an artificial intelligence photochemistry (AIPC) scheme to achieve around 8-time speed-up. Sturm and Wexler (2020) developed a mass- and energy-conserving framework for using machine learning to speed computations with a successful application in a photochemistry example. For the deposition process, Silva et al. (2019) proposed a deep learning parameterization for ozone dry deposition velocities with accurate predictions in independent new date sets, revealing the potential of neural network in encoding complex spatio-temporal processes. Liu et al. (2025) proposed a Neural Network Emulator, named ChemNNE, for fast chemical concentration modelling, which achieved good performance in accuracy and efficiency. Even though these successful applications using deep learning methods to simulate individual atmospheric chemical and physical processes, there is a missing gap in coupling these NN operator replacements together as a complete deep learning based CTM.*

"process analysis" --> I don't know what this means

**Response**: Revised to "*internal chemical and physical process analysis*".

" Interpretations of the FastCTM are also widely vowed to improve deep learning model applications

in earth system science and climate studies. " --> Not sure how you can claim this given no evidence, more aspirational than substantive

**Response**: It's revised as follows, *"Interpretations of deep learning network are also widely vowed to improve their applications in earth system science and climate studies."*.

"The FastCTM is currently configured to simulate hourly concentrations of 10 pollutant variables, including and major species of PM2.5 (SO4 2−, NO3 −, NH4+, organic matters and other inorganic components, coarse part in PM10, CO, NO2, SO2 and O3." --> Not sure how many atmospheric chemists and climate scientists want a CTM with only ten species. Needs much more motivation. Even small chemical mechanisms in operational use have around ~70 species.

**Response**: We sincerely appreciate the reviewer's comment, as it raises an important point regarding the limited number of pollutant species in the FastCTM model. FastCTM is designed to address real-time air quality forecasting, where operational usage is critical. The 10 species were selected based on their direct relevance to regulatory standards (e.g., $PM_{2.5}$, $PM_{10}$, $O_3$, $NO_2$, $SO_2$, CO) and their dominance in driving health and environmental impacts in urban and industrial regions (e.g., China, where $PM_{2.5}$ components like $SO_4^{2-}$, $NO_3^-$, and $NH_4^+$ account for most of the fine aerosol mass). By prioritizing these species, FastCTM balances accuracy with computational speed, making it suitable for rapid decision-making in policy and emergency response scenarios. While traditional CTMs (e.g., CMAQ) include ~70 species for comprehensive chemical analysis, operational forecasts often focus on criteria pollutants and key $PM_{2.5}$ components due to their regulatory importance. FastCTM replicates the outputs most frequently used in air quality management, ensuring compatibility with existing regulatory frameworks. Besides, FastCTM's performance was validated against both CMAQ simulations and ground observations (Sect. 3.1–3.2). Results show high agreement for all 10 species ($R^2 = 0.7$–$0.9$), confirming that the selected variables adequately represent key atmospheric processes. We acknowledge that FastCTM may benefit from expanded mechanisms with detailed gas-phase chemistry or aerosol microphysics. FastCTM's design supports incremental integration of additional species (e.g., via user-defined modules) without overhauling the core framework. Future versions will explore adding VOCs and secondary organics to address broader research needs.

We clarified the motivation for the 10-species configuration in the Introduction and Section 2.1 and Section 4, emphasizing regulatory and operational priorities driving species selection and plans for modular expansion in future work, as follows,

*The 10 species were selected based on their direct relevance to regulatory standards (e.g., $PM_{2.5}$, $PM_{10}$, $O_3$, $NO_2$, $SO_2$, CO) and their dominance in driving health and environmental impacts in urban and industrial regions.*

*FastCTM's design supports incremental integration of additional species (e.g., via user-defined modules) without overhauling the core framework. Future versions will explore adding VOCs and secondary organics to address broader research needs.*

Methods:

"CMAQ structures" --> I don't know what structures means here

-Is this predicting only surface level concentrations?

-I would not really call this model a CTM, this feels more like a reduced order model. There are potentially hundreds of chemical species that lead to the formation of PM2.5, O3, etc. And yet you do not mention the chemical mechanism at all in the WRF-CMAQ model. This work is basically mapping emissions to concentrations in a fairly naive way.

**Response**: To rule out the possibilities of FastCTM as a simple model mapping emissions to concentrations, we tested the land use regression (LUR) framework with machine learning models of random forest, XGBoost, and also a linear regression model. The input data for these LUR models include emissions, meteorological forecasts from WRF, and geophysical covariates, the same as those used in FastCTM. The LUR model carries out direct mapping from emission and weather data to 10 pollutants. Results have exhibited LUR's poor performance in predicting air pollutant concentrations. Related studies are included in Section 3.1, as follows.

*To validate FastCTM model, three land use regression (LUR) models were constructed, namely the linear regression model, the random forest model (with the number of trees set at 500), and the XGBoost model (with the booster specified as gbtree). These LUR models were developed using the same input meteorological data, emission, and geophysical variables. When compared with the FastCTM model, the performance of the LUR models was found to be significantly inferior (as demonstrated in Figure S10 – S12 in the SI). This outcome is, in fact, anticipated when we consider the complex nature of air quality dynamics. Air quality is not a static entity, but it varies both spatially and temporally. For instance, the transport of air pollution is a highly dynamic process that hinges on wind fields and air pollution concentrations in a reciprocal manner. The wind direction and speed dictate the trajectory along which pollutants travel, while the existing pollutant concentrations in different regions influence the overall dispersion and mixing patterns. LUR models, which predominantly rely on local input data (Wong et al., 2021; Cheng et al., 2021), struggle to capture these intricate, non-local interactions. They lack the capacity to account for the far-reaching effects such as wind-driven pollutant transport and the consequential changes in air quality over larger geographical areas.*

The supplementary Figures exhibit the performances of three machine learning models as follows.

[Figure]

*Figure S1: The evaluation performances of linear regression forecasts against CMAQ forecasts in 2023. Panel (a) and (b) respectively show RMSE values of criteria pollutants and the PM$_{2.5}$ components of. Panel (c) and (d) respectively show R$^2$ values. It should be noted that RMSE value of CO corresponds to the right axis in panel (a).*

[Figure]

*Figure S2: The evaluation performances of random forest forecasts against CMAQ forecasts in 2023. Panel (a) and (b) respectively show RMSE values of criteria pollutants and the PM$_{2.5}$ components of. Panel (c) and (d)*

[Figure]

*Figure S3: The evaluation performances of XGBoost forecasts against CMAQ forecasts in 2023. Panel (a) and (b) respectively show RMSE values of criteria pollutants and the $PM_{2.5}$ components of. Panel (c) and (d) respectively show $R^2$ values. It should be noted that RMSE value of CO corresponds to the right axis in panel (a).*

" A detailed description of CMAQ principles is available elsewhere (Byun and Schere, 2006) " --> I find this lazy. This paper is 20 years old and I do not know what you would like the reader to find in it.

**Response**: The reference provided a detailed description of the theory, model framework, and numerical methods. We added another late review study to reflect more recent developments of the chemical transport model CMAQ.

*Appel, K. W., Napelenok, S. L., Foley, K. M., Pye, H. O., Hogrefe, C., Luecken, D. J., Bash, J. O., Roselle, S. J., Pleim, J. E., and Foroutan, H.: Description and evaluation of the Community Multiscale Air Quality (CMAQ) modeling system version 5.1, Geoscientific model development, 10, 1703-1732, 2017.*

"Chemical Reaction Module" --> This just sounds like a first order approximation using idealized rate constants. There is a very rich and long history of using ODE solvers to get the solution to complex chemical mechanisms. There really is not enough discussion with this module (or really any of the

preceding module sections). You are highly simplifying each of these processes without an underlying discussion of why you are doing so. There already exist data-driven and reduced complexity modeling systems that accomplish similar air quality regulation goals (e.g., InMAP, EASIUR, APEEP).

**Response**: The simplification of chemical kinetics in FastCTM is motivated by balancing data availability with physical interpretability. While traditional CTMs (e.g., CMAQ) use detailed ODE solvers for hundreds of species and reactions, FastCTM focuses on key variables and pathways for air quality dynamics, such as secondary inorganic aerosol formation ($SO_4^{2-}$, $NO_3^-$, $NH_4^+$) and ozone photochemistry.

Comparing to other reduced-form modelling systems, models like InMAP and EASIUR focus on annual-average exposure, while FastCTM provides hourly-resolved simulations critical for real-time management. Unlike reduced-form models that aggregate source impacts, FastCTM quantifies hourly contributions from individual processes (transport, chemistry, emissions) via its modular design. Furthermore, FastCTM explicitly couples meteorology (PBLH, T, RH) with chemistry, whereas InMAP/APEEP assumes static meteorology, limiting their utility in capturing diurnal or synoptic-scale variations. Therefore, FastCTM is more like a learnable CTM model in the neural network form with some simplifications in input variables and space domain (3D to 2D) by embedding physical principles (e.g., mass conservation in transport, Arrhenius-like rate dependencies in chemistry). Besides, the modular architecture of FastCTM allows incremental addition of species/reactions (e.g., VOC oxidation pathways) without retraining the entire model. Related discussion have been added in the section,

*Reduced-form models like InMAP (Tessum et al., 2017) and EASIUR (Gentry et al., 2023) focus on annual-average exposure, while FastCTM provides hourly-resolved simulations for real-time management. FastCTM quantifies hourly contributions from individual processes (transport, chemistry, emissions) via its modular design, rather than aggregating source impacts (e.g., EASIUR's source-receptor matrices) in reduced-form models. Furthermore, FastCTM explicitly couples meteorology (PBLH, T, RH) with chemistry, whereas InMAP/APEEP (Muller and Mendelsohn, 2006) assumes static meteorology, limiting their utility in capturing diurnal or synoptic-scale variations.*

And also in the Section 4,

*Besides, FastCTM may also benefit from expanded mechanisms with detailed gas-phase chemistry or aerosol microphysics. FastCTM's design supports incremental integration of additional species (e.g., via user-defined modules) without overhauling the core framework. Future versions will explore adding VOCs and secondary organics to address broader research needs.*

Also for the diffusion module and deposition module, related discussions have been added in the corresponding sections as follows,

*Diffusion involves the physical and chemical processes that disperse pollutants in the atmosphere. It's influenced by meteorological conditions, i.e. atmospheric stability and humidity, and surface features, i.e. land terrains and vegetation (Jiang et al., 2021).*

*Air pollutant deposition refers to the process by which atmospheric pollutants are transferred to Earth's surfaces (land, water, vegetation) or removed from the air. This phenomenon plays a critical role in environmental pollution dynamics and ecosystem impacts. The deposition was closely*

*influenced by meteorological conditions and surface characteristics (Janhäll, 2015). For example, high wind disperses pollutants, while turbulence enhances dry deposition. Forests and crops act as sinks due to large surface areas for adsorption.*

-I don't explicitly understand how this is a machine learning model. You describe a sequence-to-sequence modeling framework reminiscent of an LSTM, but no mention of memory or hyper parameters in general. The inclusion of these equations may seem more like a symbolic regression kind of ML framework, but the details are sparse and lack substance. Are all the modules trained jointly so that error influences each other? Chemistry is constantly affected by other modules (and vice versa) yet these interaction terms can't be considered during training at all. That is, how does error propagate from one time step to the next in training? Is the underlying WRF-CMAQ simulations two-way coupled such that weather influences chemistry and chemistry feedbacks via aerosol effects to influence the weather? Not enough details in the underlying simulations or the joint training of modules. There are examples of this kind of offline training here:

Kelp, M. M., Jacob, D. J., Kutz, J. N., Marshall, J. D., and Tessum, C. W.: Toward Stable, General Machine-Learned Models of the Atmospheric Chemical System, Journal of Geophysical Research: Atmospheres, 125, e2020JD032759, https://doi.org/10.1029/2020JD032759, 2020.

Yang, X., Guo, L., Zheng, Z., Riemer, N., and Tessum, C. W.: Atmospheric chemistry surrogate modeling with sparse identification of nonlinear dynamics, https://doi.org/10.48550/arXiv.2401.06108, 2024.

Liu, Z.-S., Clusius, P., and Boy, M.: Neural Network Emulator for Atmospheric Chemical ODE, https://doi.org/10.48550/arXiv.2408.01829, 2024.

**Response**: (1) The five modules in FastCTM are defined in the form of operator, where operator parameters are estimated, rather than in the form of pure predictor mapping concentrations from one hour to the next. For example, in the diffusion module, FastCTM learns to encode diffusion coefficient $K$ from meteorological conditions before performing upwind finite difference procedure to solve the diffusion process $\nabla(K\nabla C_i)$. It's also the same for processes such as reaction, advection, and deposition. Therefore, it is impossible for one process to represent all atmospheric processes simultaneously. The independent contribution of each process is depicted in Figure 12 of section 3.3. Each process exhibited its patterns of contribution to hourly air pollutant concentration changes, constrained by the form of the operator in the processes. The related description is added in Section 2.3 Model Training,

*Even though five modules are defined in FastCTM, individual processes are not trained separately. The model was trained as a whole with hour-to-hour air pollutant concentrations, while each process could learn its parameters under the constrains of its dedicated formulation. Specifically, FastCTM was tuned to minimize the loss function $\mathcal{L}$, which was determined to be L2 loss (Bühlmann and Yu, 2003) of the regularized mean squared error (MSE) as shown in Eq. 15.*

(2) The configuration of the parent model was added in Section 2.1

*WRF-CMAQ simulations are not two-way coupled so that weather and chemistry do not have feedback to influence each other.*

"The main objective of our study is to build and validate a principles-guided neural network based FastCTM that could simulate spatial-temporal fields of hourly concentrations of major air pollutant species like a traditional CTM. Besides, the FastCTM could model individual contributions from each of the atmospheric processes of transport, diffusion, deposition, reaction and emission. " --> this should be stated earlier. The term "principles-guided" is vague, and I don't really consider this 'like a traditional CTM'. You discretize the potential processes that affect air quality outputs, but this is more like a traditional reduced complexity model approach. I think a deeper review into the literature would help the authors situate their work in this established landscape.

**Response**: These two sentences are moved to the last paragraph of the Introduction section, to make it clearer for the general purpose of the study. Unlike the traditional reduced-from models, FastCTM is time-resolved with a 60 seconds step to simulate the evolution of air pollutants. It generates hourly air quality simulations based on hourly meteorological conditions and emissions. The simulations have good correlations with its parent numerical CTM model CMAQ. FastCTM also exhibited reasonable responses to emission changes, also in close agreement with that of CMAQ. Besides, internal atmospheric processes could also be checked to reflect specific contributions from each process. We have added a related literature review on the application of neural networks in simulating atmospheric processes in the introduction, as follows.

*Quantifying individual processes would provide fundamental explanations for a model's predictions, and therefore is also useful in identifying potential sources of error in the model formulation or its inputs (Liu et al., 2010). With this motivation, there are studies dedicated to developing models to learn one specific atmospheric process, i.e. chemical and deposition, in the CTM model. Kelp et al. (2022) developed a neural network chemical solver for stable long-term global simulations of atmospheric chemistry, learned from the GEOS-Chem model. Xia et al. (2024) simulated 74 chemical species and 229 reactions following the SAPRC-99 mechanism with an artificial intelligence photochemistry (AIPC) scheme to achieve around 8-time speed-up. Sturm and Wexler (2020) developed a mass- and energy-conserving framework for using machine learning to speed computations with a successful application in a photochemistry example. For the deposition process, Silva et al. (2019) proposed a deep learning parameterization for ozone dry deposition velocities with accurate predictions in independent new date sets, revealing the potential of neural network in encoding complex spatio-temporal latent processes. Liu et al. (2025) proposed a Neural Network Emulator, named ChemNNE, for fast chemical concentration modelling, which achieved good performance in accuracy and efficiency. Even though these successful applications using deep learning methods to simulate individual atmospheric chemical and physical processes, there is an missing gap in coupling these NN operator replacements together as a complete deep learning based CTM.*

Also in Section 2.3.5 Chemical Reaction Module, the comparison of FastCTM with reduced-form models is discussed as follows,

*Reduced-form models like InMAP (Tessum et al., 2017) and EASIUR (Gentry et al., 2023) focus on annual-average exposure, while FastCTM provides hourly-resolved simulations for real-time management. FastCTM quantifies hourly contributions from individual processes (transport, chemistry, emissions) via its modular design, rather than aggregating source impacts (e.g., EASIUR's*

*source-receptor matrices) in reduced-form models. Furthermore, FastCTM explicitly couples meteorology (PBLH, T, RH) with chemistry, whereas InMAP/APEEP (Muller and Mendelsohn, 2006) assumes static meteorology, limiting their utility in capturing diurnal or synoptic-scale variations.*

"Furthermore, CMAQ and FastCTM forecasts were both evaluated by hourly observations from national monitoring sites (as shown in Figure S5 in the supplementary material) for six criteria pollutants (PM2.5, PM10, SO2, NO2, CO, and O3)." --> What is the point of this if CMAQ is your ground truth?

**Response**: We agree with the reviewer's point that it does not make much sense to compare FastCTM to station observations. Related comparisons are removed from the manuscript.

Results:

"Besides, since the FastCTM is a 2-D model only considering atmospheric processes within the boundary layer, lower consistency with the CMAQ model during daytime could be due to more active vertical turbulence which is not fully represented." --> Isn't the point of having this processed-based emulation the ability to attribute errors to processes? This sounds hand-wavy and does not explain the variability very well

**Response**: We sincerely appreciate the reviewer's insightful feedback regarding the attribution of errors in FastCTM's daytime performance. The comment highlights a critical aspect of our process-based emulation framework and motivates a deeper exploration of error sources. FastCTM can simulate the contributions from each process. We do not have the CMAQ process analysis in the test period. Therefore, it is not possible to attribute FastCTM's simulation errors to specific processes, by comparing the process data of CMAQ. CMAQ uses a non-local closure scheme for vertical diffusion, explicitly resolving turbulent mixing across layers. FastCTM's 2D framework parameterizes this via horizontal diffusivity and PBLH, which cannot capture vertical advection or entrainment. Besides, we considered a 2-D model in FastCTM, which means process analysis could be different from that of CMAQ in its definition natures. We are going to apply FastCTM in 3D dimensions in the later version. We added further explanation in this section as follows,

*Besides, since the FastCTM is a 2-D model only considering atmospheric processes within the boundary layer, lower consistency with the CMAQ model during the daytime could be due to more active vertical turbulence. Studies show that strong vertical mixing of air pollutants to the height above PBLH have been found (Li et al., 2017; Tang et al., 2016), which could not be fully represented in FastCTM.*

"It is important to note that the relatively low R2 values observed for NH4+ can be attributed to the fact that it is the sole cation included in the FastCTM model without a corresponding acid-base balance, which may affect the model's predictive accuracy." --> I don't see how this is the reason. WRF-CMAQ has many base pairings that can neutralize NH4+ that are not represented here. I don't recall

conservation of mass as a constraint in your chemical module. Furthermore, how do you know that NH4+ does not precipitate out as it is very hydrophilic.

**Response**: We appreciate the reviewer's critical assessment of the $NH_4+$ prediction performance and agree that our initial explanation simplified the issue. FastCTM's chemical module (Eq. 11–12) approximates $NH_4+$ dynamics using a data-driven approach trained on CMAQ outputs. While CMAQ explicitly resolves $NH_4+$ formation via reactions with HNO3 ($NH_3 + HNO_3 \rightarrow NH_4NO_3$) and $H_2SO_4$ ($2NH_3 + H_2SO_4 \rightarrow (NH_4)_2SO_4$), FastCTM does not explicitly encode these pathways. Instead, the neural network learns relationships between $NH_4+$ and precursor emissions ($NH_3$, $NOx$, $SO_2$) and meteorological variables (e.g., temperature, humidity). This simplification omits acid-base equilibria and aerosol thermodynamics, which are critical for partitioning NH4+ between gas and particle phases. The reviewer correctly notes that FastCTM's chemical module does not enforce mass conservation. While CMAQ rigorously tracks nitrogen and sulfur species across gas, aerosol, and aqueous phases, FastCTM's neural network predicts $NH_4+$ concentrations directly from emissions and meteorology without explicit mass-balance constraints. This can lead to unphysical predictions, especially when precursor emissions (e.g., $NH_3$) are over/underestimated or when thermodynamic conditions (e.g., high humidity) favor aerosol formation. The low $R^2$ for $NH_4+$ primarily reflects FastCTM's simplified chemical mechanism, which lacks explicit acid-base pairing and aerosol thermodynamics. We have revised the text accordingly in Section 3.1, as follows,

*While CMAQ explicitly resolves $NH_4^+$ formation reactions, FastCTM does not explicitly encode these pathways. Instead, the neural network implicitly learns relationships between $NH_4^+$ and precursor emissions (NH3, NOx, SO2) and meteorological variables (e.g., temperature, humidity). This simplification omits acid-base equilibria and aerosol thermodynamics, which are critical for partitioning $NH_4^+$ between gas and particle phases. The low $R^2$ for $NH_4^+$ primarily reflects FastCTM's simplified chemical mechanism in this part, which could be improved by adding related species in the simulation.*

-I actually believe it is quite concerning that the RMSEs vary diurnally. You should also plot the WRF-CMAQ and FastCTM time series against each other. A diurnal error actually may suggest that you are not correctly learning the atmospheric dynamics of the system well. You may be predicting an average concentration across all time and that's why you see a diurnal error profile.

"FastCTM forecasts using zero values as input air quality data were almost the same as that using ordinary input in the long leading hours, indicating that FastCTM simulations in long leading hours are not affected by initial conditions, just like deterministic numeric CTMs (such as CMAQ)" --> This is hard to conclude, you need to plot actual concentration time series instead of RMSEs. It seems like the error is always the same, this could mean the FastCTM always predicts the same values given the time of day. More results need to be presented.

**Response**: As reviewer kindly pointed, FastCTM possibly has taken average pollutant concentration from five-year training data in 2018-2022. In order to confirm that FastCTM was able to predict air quality based on given meteorological conditions and emissions, daily average FastCTM simulation in the fifth leading day (leading hours 96-119) in the test year of 2023 is compared with daily average CMAQ simulations in 2023 and in the training years of 2018-2022. Results revealed that FastCTM

forecasts are generally in good correlation with CMAQ forecasts in 2023, rather than that in 2018-2023. It means FastCTM has learned the evolution rules of air pollutant concentrations, instead of just giving average air pollutant concentration according to time of the year. Related results have been added in the manuscript in section 3.1.1, as follows.

*Annually, the daily air quality typically exhibits similar fluctuations to those in other years, which can be primarily attributed to the cyclical nature of meteorological conditions and pollutant emission patterns. The FastCTM model was trained using a comprehensive dataset spanning five years, from 2018 to 2022. In light of this, it was crucial to rule out the possibility that the model was merely reproducing historical averages during the test year of 2023. To this end, the daily national average concentrations of $PM_{2.5}$ and $O_3$ in 2023, as predicted by FastCTM, were meticulously compared with those simulated by CMAQ in the same test year, as well as with the CMAQ forecasts from the training years of 2018-2022. As illustrated in Figure 5, it becomes evident that the predictions made by FastCTM in 2023 align more closely with the actual CMAQ forecasts for that year, rather than with the forecasts generated from the training data of 2018-2022. This finding not only validates the adaptive learning capabilities of the FastCTM model but also indicates that the model is not resorting to a simplistic approach of taking the average concentration from the previous five years based on the time of day. Instead, it is likely incorporating real-time meteorological feedback, adjusting for any shifts in emission patterns, and leveraging its learned relationships to provide more accurate and contemporaneous predictions.*

[Figure]

*Figure 5: The timeseries of FastCTM forecasts against CMAQ forecasts.*

Figure 3 is unwieldy. There are 60 mulitplots and not well labeled on the figure. Here you should show spatial differences in terms of both absolute and relative error. Seems like FastCTM does not capture the highest concentration values, which is concerning given that is the largest impact on health and climate. Hard to have any substantive discussion of results without any quantitative measure regarding Figure 3.

**Response**: As the reviewer suggested, we revised the manuscript in the section as follows,

*The spatial distributions of the mean absolute error (MAE) and the normalized mean absolute error (NMAE) are presented in Figure 3. For the six criteria pollutants, the MAE values are higher in polluted areas. This could be attributed to the complex and dynamic nature of pollutant interactions in such regions. In polluted environments, there are often multiple sources of emissions, complex chemical reactions, and variable meteorological conditions that can lead to greater discrepancies between the model-predicted and actual pollutant concentrations. Conversely, the NMAE values*

*exhibit an opposite trend, being lower in polluted areas. In these regions, the NMAE values are typically around 0.2, in contrast to the relatively higher values of approximately 1 in cleaner areas. The NMAE is a normalized metric that takes into account the magnitude of the actual pollutant concentrations. A lower NMAE in areas with high pollution levels suggests that the FastCTM model is effectively capturing the overall magnitude and trends of pollutant concentrations relative to the reference CMAQ model.*

[Figure]

*Figure 4: Spatial distribution of mean absolute error (panels a, c, e, g, i, and k) and normalized mean absolute error for the six criteria pollutants (panels b, d, f, h, j and l) of FastCTM comparing to CMAQ in 2023.*

Section 3.1.2. Again, I don't see why this comparison makes sense. You do not incorporate any station data, so why would you make comparisons against it? The WRF-CMAQ model is the ground truth here.

**Response**: We agree with the reviewer's point that related comparisons between FastCTM to station observations are removed from the manuscript.

Sections 3.2: These don't have much meaning if we do not understand how the FastCTM model behaves in relation to the parent model

**Response**: We agree with the reviewer's comment that more analysis are needed to verify FastCTM's capabilities to capture the impact of changes in emissions, especially compared to the parent model of WRF-CMAQ. We added a comparison between FastCTM and CMAQ under 11 emission scenarios in the winter month of January 2019 and in the summer month of July 2019. The results signified that the FastCTM simulations manifested a good agreement with those of CMAQ, which was manifested in two principal aspects. Firstly, the FastCTM model forecasted positive responses to augmented emissions in the no-control (NCtrl) scenario and negative responses in the other emission-controlled scenarios just like CMAQ. This implies that when emissions were unrestricted and increased, as in the

NCtrl scenario, FastCTM could capture the increasing trend as that of CMAQ. In scenarios where emissions were reduced, they both predicted a decline. Secondly, in scenarios characterized by more substantial emission reductions, the FastCTM model simulated a more pronounced decrease in air pollutant concentrations. This is of particular significance as it shows the model's sensitivity to the magnitude of emission inputs. It suggests that the FastCTM model is not only capable of discerning changes in emission scenarios but can also reflect the degree of impact on air quality, thereby reinforcing its reliability and utility in simulating air quality dynamics in agreement with CMAQ. Related results in the manuscript are shown as follows.

*The sensitivities of FastCTM simulations to emission interventions were contrasted with those of CMAQ. Specifically, CMAQ was employed to simulate 11 emission scenarios over the two-month periods of January and July 2019 in Southwest China (Huang et al., 2022). The alterations in emissions relative to the base case are presented in Table 1. Among these scenarios, 10 involved reduced emissions of major species, with only the no-control scenario exhibiting increased emissions. Utilizing the identical emissions and meteorological data, FastCTM also conducted simulations, which were then compared to those of CMAQ. For the 11 scenarios in question, the changes in air pollutant concentrations relative to the base case at the locations of 139 national air quality monitoring stations (Figure S14 in the SI) were extracted and compared in the winter month of January 2019 (Figure 9) and in summer month of July 2019 (Figure 10). The results indicated that, overall, the FastCTM simulations were in good agreement with those of CMAQ reflected in two aspects. First, FastCTM predicted positive responses to increased emissions in the nocontrol (NCtrl) scenario and negative responses to other emission-controlled scenarios just as CMAQ. Second, FastCTM simulated larger air pollutant concentration decrease in those scenarios with higher emission reductions. Specifically, in January 2019, with the exception of $NO_2$, FastCTM responded to emission changes with an interquartile range (IQR, 25% - 75% percentile) similar to that of CMAQ (Figure 9). For $NO_2$, in the same emission reduction scenarios, FastCTM simulated lower $NO_2$ values. In the summer month of July 2019, as depicted in Figure 10, all the criteria pollutants except CO demonstrated a comparable degree of response to emission reductions. The comparison suggests that the FastCTM model is not only capable of discerning changes in emission scenarios but can also reflect the degree of impact on air quality, thereby reinforcing its reliability and utility in simulating air quality dynamics in tandem with CMAQ. It should be noted that in both months, FastCTM exhibited slightly larger median values, suggesting its greater sensitivity to emission interventions.*

**Table 1**. *The emission change details of emission scenarios*

| Scenario | abbreviation | Sector | $NO_x$ | VOCs | $SO_2$ | CO | $PM_{2.5}$ | PMC |
|---|---|---|---|---|---|---|---|---|
| **nocontrol** | **NCtrl** | Industrial | 30% | 30% | 30% | 30% | 30% | 30% |
| | | Traffic | 20% | 20% | 20% | 20% | 20% | 20% |
| **medianX** | **MedX** | Industrial | -36% | -35% | -48% | -23% | -9% | -9% |
| | | Traffic | -40% | -10% | 0 | -26% | -10% | -10% |
| **medianY** | **MedY** | Industrial | -26% | -20% | -38% | -13% | -4% | -4% |

| | | | | | | | | |
|---|---|---|---|---|---|---|---|---|
| | | Traffic | -30% | 0% | 0 | -16% | -5% | -5% |
| medianZ | MedZ | Industrial | -36% | -10% | -48% | -23% | -9% | -9% |
| | | Traffic | -40% | 0% | 0 | -26% | -10% | -10% |
| median-3 | Med-3 | Industrial | -10% | -10% | -18% | 0 | 0 | 0 |
| | | Traffic | -10% | 0% | 0 | 0 | 0 | 0 |
| median-2 | Med-2 | Industrial | -16% | -20% | -28% | -3% | 0 | 0 |
| | | Traffic | -20% | 0% | 0 | -6% | 0 | 0 |
| median-1 | Med-1 | Industrial | -26% | -35% | -38% | -13% | -4% | -4% |
| | | Traffic | -30% | -10% | 0 | -16% | -5% | -5% |
| median0 | Med0 | Industrial | -36% | -50% | -48% | -23% | -9% | -9% |
| | | Traffic | -40% | -20% | 0 | -26% | -10% | -10% |
| median+1 | Med+1 | Industrial | -46% | -65% | -58% | -33% | -19% | -19% |
| | | Traffic | -50% | -30% | 0 | -36% | -20% | -20% |
| median2030 | Med30 | Industrial | -55% | -70% | -80% | -40% | -40% | -40% |
| | | Traffic | -60% | -40% | 0 | -40% | -40% | -40% |
| median2035 | Med35 | Industrial | -80% | -80% | -90% | -60% | -50% | -50% |
| | | Traffic | -80% | -60% | 0 | -60% | -50% | -50% |

[Figure]

*Figure 9: Air pollutant concentration changes in terms of base case simulated by CMAQ (subplots of a, c, e, g, i and k in first column) and by FastCTM (subplots of b, d, f, h, j and l in second column) in January 2019.*

[Figure]

*Figure 10: Air pollutant concentration changes in terms of base case simulated by CMAQ (subplots of a, c, e, g, i, and k in the first column) and by FastCTM (subplots of b, d, f, h, j, and l in the second column) in July 2019.*

Figure 8: These color bars are difficult to discern changes in concentrations. Does adding d through h yield panels a or b? Again, individual contribution doesn't matter if we don't know how the model actually behaves.

**Response**: Adding panel d through h yield panel c. To validate the process analysis by FastCTM, its simulation results are compared to those by CMAQ. We added related results and discussion in Section 3.3 as follows,

*In this study, we further selected the data recorded at 23:00 on October 13, 2024, to compare the*

*impacts of the five major atmospheric physical and chemical processes, as simulated by FastCTM and CMAQ, on PM$_{2.5}$ concentration changes (Figure 12). Emissions, advection processes, and diffusion processes demonstrated a relatively high degree of consistency between the two models. Regarding the simulation of chemical reactions, while the spatial distribution of high-value areas in the FastCTM results was comparable to that of CMAQ, the simulated values in FastCTM were notably higher. Correspondingly, FastCTM overestimated the contribution of the deposition process. This overestimation counterbalanced the impact of the higher chemical reaction values. The difference in the simulated deposition contributions between the two models could be due to differences in how they represent these influencing factors. Overall, the simulation results of the process contributions by FastCTM and its parent model CMAQ were relatively consistent. This consistency indicates that, despite some differences in the magnitude of certain process simulations, FastCTM is capable of capturing the essential features of atmospheric processes related to PM$_{2.5}$ concentration changes, similar to CMAQ. Such consistency provides confidence in the reliability of FastCTM for simulating and understanding the complex interplay of atmospheric processes of PM$_{2.5}$.*

[Figure]

*Figure 5: An example of contributions from five major atmospheric processes to PM$_{2.5}$ changes (µg/m$^3$) by CMAQ (first row) and FastCTM (second row) at 23:00 on October 13, 2024.*

---

## Author Response (AR1)

**Point-to-Point Responses to Reviewer's Comments**

**Comments from Reviewer 1**

The authors describe a new, neural network-based reduced-order model of atmospheric chemistry and transport (FastCTM) which has been trained using an extensive dataset of output from CMAQ. FastCTM uses a novel and interesting approach, building physics-informed networks for five separate operators. The authors show that FastCTM is able to reproduce the general patterns of concentrations calculated by CMAQ for an out-of-training-data year (2023), and that the sensitivities of FastCTM to key meteorological variables or nation-wide changes in emissions mostly follow expected patterns. If FastCTM can be shown to be reliable in policy-relevant contexts then it could be a very useful tool.

This approach to modelling is interesting, and this methodological advance has the potential to significantly accelerate air quality scenario analysis. A CTM which can respect key physical constraints (e.g. mass conservation) while also accurately reproducing the effect of different perturbations to emissions and meteorological fields would have great value. However, the manuscript as written does not quite live up to this promise. Along with some minor concerns, the key challenge is that the authors do not show evidence that this new model can fulfil the roles of a CTM and produce accurate results for one of the most common use cases (i.e. understanding the effects of different perturbations). I explain this concern in more detail below, and until this concern is addressed I do not believe the manuscript should be accepted for publication by GMD.

*Major comments*

The most significant concern relates to the validation/evaluation of the model. The authors appear to have trained the five physical operators based on several years of output from the CMAQ chemistry transport model. While I have some questions regarding the training process, I will take it as read for the moment that the training was done in such a way as to avoid overfitting. However, the verification of the model rests on its ability to predict, from the 2018-2022 data, the performance in 2023. This approach is inadequate for two reasons. First, the authors do not compare the performance of the model to simpler approaches with the same data such as generalized additive models, gradient boosting, or linear regression with land use (see e.g. Wong et al., 2021 and Cheng et al., 2021). Without such a comparison to evaluate how such models would have performed in predicting 2023, it is difficult to say what the magnitude of FastCTM's advance is. This is exacerbated by the relatively shallow quantitative assessment in section 3.1.1. RMSE and R2 values are provided, but it is not clear how these were calculated; given that these are calculated as a function of time, are these calculated based on the difference in each of the 158,742 grid cells between CMAQ and FastCTM? A deeper analysis which investigates how model

performance varies between (e.g.) rural and urban areas, coastal and inland areas, winter and summer, and so on would provide a much more robust test of the model's ability to predict the effect of changing meteorology. This could be informed by (e.g.) taking the difference of FastCTM for 2023 against CMAQ for 2023, and comparing that to the difference between CMAQ for 2023 and the average of CMAQ from 2018 to 2022. This would at least demonstrate whether FastCTM provides more explanatory power for the mean atmospheric state than taking the average concentration from the previous five years.

**Response**: Thanks. We agree with review's point that current analysis is not adequately to demonstrate that FastCTM has actual capabilities to simulate air quality changes by learning and representing physical and chemical processes. As reviewer suggested, more tests, comparisons and analysis are performed.

(1) For comparing with simpler machine learning models, we tested three models of Linear Regression, Random Forest and XGBoost with the same train and test dataset. The results are added in the manuscript as follows,

*In addition, three land use regression (LUR) models were constructed, namely the linear regression model, the random forest model (with the number of trees set at 500), and the XGBoost model (with the booster specified as gbtree). These LUR models were developed using the same input meteorological, emission, and geophysical covariates. When compared with the FastCTM model, the performance of the LUR models was found to be significantly inferior (as demonstrated in Figure S6 - S8 in the Supplementary Information). This outcome is, in fact, anticipated when we consider the complex nature of air quality dynamics. Air quality is not a static entity, but it varies both spatially and temporally. For instance, the transport of air pollution is a highly dynamic process that hinges on wind fields and air pollution concentrations in a reciprocal manner. The wind direction and speed dictate the trajectory along which pollutants travel, while the existing pollutant concentrations in different regions influence the overall dispersion and mixing patterns. LUR models, which predominantly rely on local input data (Wong et al., 2021; Cheng et al., 2021), struggle to capture these intricate, non-local interactions. They lack the capacity to account for the far-reaching effects such as wind-driven pollutant transport and the consequential changes in air quality over larger geographical areas.*

With the supplementary Figures demonstrating performances of three machine learning models displaying in the following part.

[Figure]

*Figure S1: The evaluation performances of linear regression forecasts against CMAQ forecasts in 2023. Panel (a) and (b) respectively show RMSE values of criteria pollutants and the PM$_{2.5}$ components of. Panel (c) and (d) respectively show R$^2$ values. It should be noted that RMSE value of CO corresponds to the right axis in panel (a).*

[Figure]

*Figure S2: The evaluation performances of random forest forecasts against CMAQ forecasts in 2023. Panel (a) and (b) respectively show RMSE values of criteria pollutants and the PM$_{2.5}$ components of. Panel (c) and (d) respectively show R$^2$ values. It should be noted that RMSE value of CO corresponds to the right axis in panel (a).*

[Figure]

*Figure S3: The evaluation performances of XGboost forecasts against CMAQ forecasts in 2023. Panel (a) and (b) respectively show RMSE values of criteria pollutants and the PM$_{2.5}$ components of. Panel (c) and (d) respectively show R$^2$ values. It should be noted that RMSE value of CO corresponds to the right axis in panel (a).*

(2) As for the calculation process for the metrics of RMSE and R2, they are elaborated in the section 2.5 Model Evalution, as follows.

*The metrics of root mean square error (RMSE) and coefficient of determination (R$^2$) were calculated daily in each of 119 leading hours on the difference in each of the 158,742 grid cells between CMAQ and FastCTM. Therefore, 119 static values for each metric of R$^2$ and RMSE were obtained on each day of the independent test year of 2023. The statistic values on each day are then averaged for the same leading hour for comparison.*

(3) As reviewer suggested, evaluations of FastCTM comparing to CMAQ in rural/urban, inland/coastal areas, and cold/warm seasons are further performed. Generally, they have similar performances in comparative areas or seasons. But FastCTM exhibited lower correlations in urban areas and coastal areas. In urban area, emission sources and chemical processes are more complex than that in rural area, making it harder for FastCTM to simulate due to its 2D setting and fewer chemicals considered than CMAQ. It is also same true for FastCTM's performance in coastal area, where meteorological conditions are more varied in time. Related discussions and results are added in the section 3.1.1. and in the supplementary material, which as follows.

*Defining the warm season as the months from April to September and the winter and cold season as the remaining months, the FastCTM model exhibited comparable performances. As shown in Figure 3 (with detailed information in Figure S9 in the SI), the coefficient of determination R$^2$ values for the six criteria pollutants were 0.82, 0.8, 0.8, 0.82, 0.91, and 0.7 in the warm season, and 0.8, 0.79, 0.78, 0.83, 0.88, and 0.68 in*

*the cold season, respectively. To assess the performance variations of FastCTM across different spatial locations, comparative evaluations were carried out in urban and rural areas as well as in inland and coastal regions. Generally, FastCTM demonstrated slightly higher accuracies in rural areas compared to urban areas (as presented in Figure S10 in the SI). This outcome is reasonable given the more intricate emission and chemical processes prevalent in urban settings (Guo et al., 2014). Similarly, FastCTM exhibited comparable performances in inland areas to those in coastal areas, with the exception of PM$_{2.5}$ and PM$_{10}$ (Figure S11 in the SI).*

[Figure]

*Figure 4: The evaluation R$^2$ values of FastCTM forecasts in warm/cold seasons, rural/urban areas and coastal/inland areas.*

[Figure]

*Figure S5: The evaluation performances of random forest forecasts against CMAQ forecasts in warm season of 2023. Panel (a) and (b) respectively show RMSE values of criteria pollutants and the PM$_{2.5}$ components of. Panel (c) and (d) respectively show R$^2$ values. It should be noted*

[Figure]

*Figure S6: The evaluation performances of XGBoost forecasts against CMAQ forecasts in rural and urban areas in 2023. Panel (a) and (b) respectively show RMSE values of criteria pollutants and the $PM_{2.5}$ components of. Panel (c) and (d) respectively show $R^2$ values. It should be noted that RMSE value of CO corresponds to the right axis in panel (a).*

[Figure]

*Figure S7: The evaluation performances of FastCTM forecasts against CMAQ forecasts in inland and coastal areas in 2023. Panel (a) and (b) respectively show RMSE values of criteria pollutants and the $PM_{2.5}$ components of. Panel (c) and (d) respectively show $R^2$ values. It should be noted that RMSE value of CO corresponds to the right axis in panel (a).*

(4) As reviewer kindly pointed, FastCTM possibly has taken average pollutant concentration from five-year training data in 2018-2022. In order to confirm that FastCTM was able to predict air quality based on given meteorological conditions and emissions, daily average FastCTM simulation in the fifth leading day (leading hours 96-119) in the test year of 2023 is compared with daily average CMAQ simulations in 2023 and in the training years of 2018-2022. Results revealed that FastCTM forecasts generally in good correlation with CMAQ forecasts in 2023, rather than that in 2018-2023. It means FastCTM has learn the evolution rules of air pollutant concentrations, instead of just giving average air pollutant concentration according to time of the year. Related results have been added in the manuscript in the section 3.1.1, as follows.

*Annually, the daily air quality typically exhibits similar fluctuations to those in other years, which can be primarily attributed to the cyclical nature of meteorological conditions and pollutant emission patterns. The FastCTM model was trained using a comprehensive dataset spanning five years, from 2018 to 2022. In light of this, it was crucial to rule out the possibility that the model was merely reproducing historical*

*averages during the test year of 2023. To this end, the daily national average concentrations of $PM_{2.5}$ and $O_3$ in 2023, as predicted by FastCTM, were meticulously compared with those simulated by CMAQ in the same test year, as well as with the CMAQ forecasts from the training years of 2018-2022. As illustrated in Figure 4, it becomes evident that the predictions made by FastCTM in 2023 align more closely with the actual CMAQ forecasts for that year, rather than with the forecasts generated from the training data of 2018-2022. This finding not only validates the adaptive learning capabilities of the FastCTM model but also indicates that the model is not resorting to a simplistic approach of taking the average concentration from the previous five years based on the time of day. Instead, it is likely incorporating real-time meteorological feedback, adjusting for any shifts in emission patterns, and leveraging its learned relationships to provide more accurate and contemporaneous predictions.*

[Figure]

*Figure 5: The evaluation performances of FastCTM forecasts against CMAQ forecasts in 2023.*

Second, and perhaps more importantly, the function of FastCTM is to reproduce the results of high-fidelity CTMs at a fraction of the computational cost – specifically to support air pollution simulations, sensitivity analysis, and internal process analysis (abstract lines 30-32). The comparison to 2023 only tells us that FastCTM can reproduce the general pattern of air pollution in 2023, but does not tell us whether FastCTM will accurately predict the effect of interventions. The sensitivity tests in section 3.2 have no basis for comparison, and are in any case so broad (representing nationwide changes in temperature, PBL height, or emissions) that they are a limited test of the CTM's capabilities. At the very minimum, an evaluation is needed which shows that FastCTM's trends actually match the underlying trends in WRF-CMAQ; this should be straightforward for the emissions cases. Since the goal of FastCTM is to reduce computational costs, it is critical that FastCTM be shown to be faithful to its parent model for realistic applications such as projecting the impact of a change in emissions. Going further and comparing sensitivities for local or single-sector emissions changes would provide even more powerful proof, and I strongly recommend that the authors consider such a comparison.

Without these kinds of quantitative comparisons I can only judge the model's success based on data such as Figure 3, where I am concerned because the patterns do not – speaking qualitatively – appear to match that well between CMAQ and FastCTM. I am particularly concerned that the model may be mostly reproducing emissions maps and

historical scalings, rather than accurately representing chemistry and transport (especially given that transport is 2-D only). A more critical, quantitative analysis of the models strengths and weaknesses would be necessary before I would recommend its use in a scientific or regulatory context.

**Response**: We agree with reviewer's comment that more analysis needed to verify FastCTM's capabilities to project the impact of a change in emissions. Since the emissions are same for each year from 2018-2023, it is not possible to test FastCTM's trends to that of WRF-CMAQ. Instead, we added comparison between FastCTM and CMAQ in 11 emission intervention scenarios in the winter month of January 2019 and in summer month of July 2019. The results signified that, on the whole, the FastCTM simulations manifested a high level of concordance with those of CMAQ, which was manifested in two principal aspects. Firstly, akin to CMAQ, the FastCTM model forecasted positive responses to augmented emissions in the no-control (NCtrl) scenario and negative responses in the other emission-controlled scenarios. This implies that when emissions were unrestricted and increased, as in the NCtrl scenario, both models detected a corresponding upward trend in pollutant levels. Conversely, in scenarios where emissions were curbed, they both predicted a decline. Secondly, in scenarios characterized by more substantial emission reductions, the FastCTM model simulated a more pronounced decrease in air pollutant concentrations. This is of particular significance as it showcases the model's sensitivity to the magnitude of emission interventions. It suggests that the FastCTM model is not only capable of discerning changes in emission scenarios but can also reflect the degree of impact on air quality, thereby reinforcing its reliability and utility in simulating air quality dynamics in tandem with CMAQ. Related results in the manuscript were shown as follows.

*The sensitivities of FastCTM simulations to emission interventions were contrasted with those of CMAQ. Specifically, CMAQ was employed to simulate 11 emission scenarios over the two-month periods of January and July 2019 in Southwest China. The alterations in emissions relative to the base case are presented in Table 1. Among these scenarios, 10 involved reduced emissions of major species, with only the no-control scenario exhibiting increased emissions. Utilizing the identical emissions and meteorological data, FastCTM also conducted simulations, which were then compared to those of CMAQ. For the 11 scenarios in question, the changes in air pollutant concentrations relative to the base case at the locations of 139 national air quality monitoring stations (Figure S14 in the SI) were extracted and compared in the winter month of January 2019 (Figure 9) and in summer month of July 2019 (Figure 10). The results indicated that, overall, the FastCTM simulations were in good agreement with those of CMAQ reflected in two aspects. First, FastCTM predicted positive responses to increased emissions in the nocontrol (NCtrl) scenario and negative responses to other emission-controlled scenarios just as CMAQ. Second, FastCTM simulated larger air pollutant concentration decrease in those scenarios with higher emission reductions. Specifically, in January 2019, with the exception of $NO_2$, FastCTM responded to emission changes with an interquartile range (IQR, 25% - 75% percentile) similar to that of CMAQ (Figure 9). For $NO_2$, in the same emission reduction scenarios, FastCTM*

simulated lower NO₂ values. In the summer month of July 2019, as depicted in Figure 10, all the criteria pollutants except CO demonstrated a comparable degree of response to emission reductions. The comparison suggests that the FastCTM model is not only capable of discerning changes in emission scenarios but can also reflect the degree of impact on air quality, thereby reinforcing its reliability and utility in simulating air quality dynamics in tandem with CMAQ. It should be noted that in both months, FastCTM exhibited slightly larger median values, suggesting its greater sensitivity to emission interventions.

*Table 1*. The emission change details of emission scenarios

| Scenario | abbreviation | Sector | $NO_x$ | VOCs | $SO_2$ | CO | $PM_{2.5}$ | PMC |
|---|---|---|---|---|---|---|---|---|
| nocontrol | NCtrl | Industrial | 30% | 30% | 30% | 30% | 30% | 30% |
| | | Traffic | 20% | 20% | 20% | 20% | 20% | 20% |
| medianX | MedX | Industrial | -36% | -35% | -48% | -23% | -9% | -9% |
| | | Traffic | -40% | -10% | 0 | -26% | -10% | -10% |
| medianY | MedY | Industrial | -26% | -20% | -38% | -13% | -4% | -4% |
| | | Traffic | -30% | 0% | 0 | -16% | -5% | -5% |
| medianZ | MedZ | Industrial | -36% | -10% | -48% | -23% | -9% | -9% |
| | | Traffic | -40% | 0% | 0 | -26% | -10% | -10% |
| median-3 | Med-3 | Industrial | -10% | -10% | -18% | 0 | 0 | 0 |
| | | Traffic | -10% | 0% | 0 | 0 | 0 | 0 |
| median-2 | Med-2 | Industrial | -16% | -20% | -28% | -3% | 0 | 0 |
| | | Traffic | -20% | 0% | 0 | -6% | 0 | 0 |
| median-1 | Med-1 | Industrial | -26% | -35% | -38% | -13% | -4% | -4% |
| | | Traffic | -30% | -10% | 0 | -16% | -5% | -5% |
| median0 | Med0 | Industrial | -36% | -50% | -48% | -23% | -9% | -9% |
| | | Traffic | -40% | -20% | 0 | -26% | -10% | -10% |
| median+1 | Med+1 | Industrial | -46% | -65% | -58% | -33% | -19% | -19% |
| | | Traffic | -50% | -30% | 0 | -36% | -20% | -20% |
| median2030 | Med30 | Industrial | -55% | -70% | -80% | -40% | -40% | -40% |
| | | Traffic | -60% | -40% | 0 | -40% | -40% | -40% |

| median20 35 | Med35 | Industrial | -80% | -80% | -90% | -60% | -50% | -50% |
|---|---|---|---|---|---|---|---|---|
| | | Traffic | -80% | -60% | 0 | -60% | -50% | -50% |

[Figure]

*Figure 9: Air pollutant concentration changes in terms of base case simulated by CMAQ (subplots of a, c, e, g, i and k in first column) and by FastCTM (subplots of b, d, f, h, j and l in second column) in January 2019.*

[Figure]

*Figure 10: Air pollutant concentration changes in terms of base case simulated by CMAQ (subplots of a, c, e, g, i and k in first column) and by FastCTM (subplots of b, d, f, h, j and l in second column) in July 2019.*

*Minor comments*

The description of the five operators does not quite seem to verify that physical constraints are being satisfied but this may simply be a misinterpretation on my part. For example, can you confirm that the method you used to generate the convolution kernels (Eq 5-7) for transport ensures mass conservation? This seems to depend on $C_i$ being in units of molec/cm3 rather than ppbv, but the units of $C_i$ are not clearly specified.

**Response**: We added description for the model framework in the Section 2.3. The unit for all pollutants is μg/m³. We used a upwind-scheme to simulate diffusion and advection processes. For the scheme, masses are conserved. However, FastCTM as a whole is not mass conserved, because it also consists other neural network modules such as reaction and deposition. These deep learning modules are learned to minimize the loss function of mean squared error. Revised model description and figures are shown as follows.

*Instead, we use 1-hour initial pollutant concentration (J=1) to simulate 24-hour air quality pollutants (K=24), to ensure FastCTM is dedicated to learn air quality changes between neighboring two hours as shown in Figure 1a. In other words, at time $t = 0$, FastCTM predicted K-hour air pollutant concentrations of $C_{t=0}, C_{t=1}, \dots, C_{t=K-1}$, given the input air pollutant concentration in previous hour $C_{t=-1}$ and corresponding meteorological data and emissions at time $t = 0,1,\dots,K\text{-}1$. The unit of concentrations is μg/m³ for all pollutants.*

[Figure]

*Figure 8: (a) General model workflow, and (b) the basic simulator module structure at the time step t of deep learning simulation model FastCTM designed according to Eq.1. Arrows and boxes with different colours represent calculation modules of different atmospheric physical and chemical processes.*

A related concern is that surface layer winds are used and treated conservatively, which neglects the fact of rapid vertical mixing. Can the authors provide evidence that the surface winds (which would be expected to be slower than the mean wind speed in the boundary layer) are accurately predicting pollutant motion? It seems that any model which is designed to predict transport using only the horizontal near-surface winds will underestimate overall transport. Should the model not be using the PBL-averaged horizontal winds in Eq. 4 instead of the surface winds at 10-meter height (lines 83-84)?

**Response**: We appreciate the reviewer's critical observation regarding the use of

surface-layer winds (10 m) in FastCTM's transport module and the potential underestimation of pollutant transport. Regarding the suggestion of using PBL - averaged horizontal winds, we understand the potential benefits of such an approach. However, implementing PBL - averaged winds would require additional data processing and model complexity. In our case, we aimed to strike a balance between model accuracy and computational efficiency. The use of 10 - meter surface winds allows for a more straightforward implementation, which is beneficial for the model's scalability and real - time application. Moreover, the evaluation of the FastCTM against CMAQ simulations and observational data, as presented in our manuscript, indicates that the model performs reasonably well in predicting pollutant concentrations. The model achieved high agreements with CMAQ in terms of RMSE and $R^2$ values for multiple pollutants, suggesting that despite using surface winds, it is able to capture the essential aspects of pollutant transport and transformation. Besides, in Section 3.2.1, responses to increased wind speed are evaluated, it would be reasonable to assume decreased air pollutant concentration predicted by FastCTM using larger PBL-averaged wind speed. For example, with a 10% increase of wind speed would lead to about 2% decrease of air pollutant concentrations except for $O_3$. Therefore, it would make much difference with PBL-averaged wind speed. Future revision of FastCTM in 3D dimensions would consider the vertically varied wind fields to better simulate wind-related processes.

It would be helpful to get more detail on how components such as the diffusion encoder were trained. Currently the manuscript states that 5 years of data (2018 – 2022 inclusive) were used in training, but not how the five different models were trained using that data. A naïve assessment would assume that all five sub-models were trained based simply on hour-to-hour pollutant concentrations, but that would suggest that the models were each trying to represent all atmospheric processes simultaneously.

**Response**: The five modules in FastCTM are defined in the form of operator, where operator parameters are estimated, rather than in the form of pure predictor mapping concentrations from one hour to the next. For example, in the diffusion module, FastCTM learns to encode diffusion coefficient $K$ from meteorological conditions before performing upwind finite difference procedure to solve diffusion process $\nabla(K\nabla C_i)$. It's also same for processes such as reaction, advection, and deposition. Therefore, it is not possible for one process to represent all atmospheric processes simultaneously. The independent contribution of each process depicted in Figure 12 of section 3.3. Clearly, each process exhibited their patterns of contribution to hourly air pollutant concentrations changes, constrained by the form of the operator in the processes. Related description was added in the Section 2.3 Model Training,

*Even though five modules are defined in FastCTM, individual processes are not trained separately. The model was trained as a whole with hour-to-hour air pollutant concentrations, while each process could learn its parameters under the constrains of*

*its dedicated formulation. Specifically, FastCTM was tuned to minimize the loss function $\mathcal{L}$, which was determined to be L2 loss (Bühlmann and Yu, 2003) of the regularized mean squared error (MSE) as shown in Eq. 15.*

Line 172 says that the reaction encoder in Equation 12 "has the same structure as that of reaction and deposition encoder models (Eq. 10)". This is recursive, but also Eq. 10 refers to the diffusion module?

**Response**: This error was revised as follows in corresponding secton,

*Therefore, the reaction rate constant $k$ is simulated using a spatial encoder function Encoder as shown in Eq. 12, which has the same structure as that of diffusion encoder modules (Eq. 10).*

On line 194, "We did not use the fixed area as that in the previous studies (Xing et al., 2022)" – can you elaborate? It was not clear to me what this meant.

**Response**: Revised as follows,

*The random sampling tactics would help model learn inherent physical and chemical principles model rather than just statistical spatiotemporal autocorrelations using data in constant spatial area (Xing et al., 2022). Besides, the spatio-temporal random samples contain varied emissions which would improve FastCTM adaption to changing emission levels.*

The y-axis labels on Figure 5 say "Percentage", but from context it appears these must really be the factor difference from the baseline (as all cross at 1.0).

**Response**: Revised.

Finally, there are numerous minor grammatical errors (e.g. L14: simulations and managements; L67: interpretations of the FastCTM are also widely vowed; L70: including and major; and so on). This is not important for judgment of the paper's appropriateness for publication, but I recommend the authors take another look at the paper to correct such minor issues.

**Response**: We have read through the manuscript and made through revision on the writing of the manuscript.

*Citations*

Wong et al., "Using a land use regression model with machine learning to estimate

ground level PM2.5." Environ. Pollut., 2021.

Cheng et al. "Influence of weather and air pollution on concentration change of PM2.5 using a generalized additive model and gradient boosting machine." Atmospheric environment, 2021.

**Comments from Reviewer 2**

Lyu et al. put forth the 'FastCTM' model which seems to be a reduced complexity model that discretizes changes in concentrations for 10 air pollution species. Though interesting, the presentation of methods, results, and context of the study needs to be heavily refined before being accepted. The details of the study are currently not sufficient as they stand.

Introduction:

"The air pollutant and species concentrations can be then calculated by solving these complicated equations with numeric methods (Byun and Schere, 2006), which is often time-consuming and requires intense computational resources." --> This thought is not very well flushed out. A single reference from 2006 does not detail at all what makes these models computationally expensive.

**Response**: This sentence is revised with two more related references are added, as follows.

*The air pollutant and species concentrations can be then calculated by solving these complicated equations with numeric methods (Byun and Schere, 2006), which is often time-consuming (Leal et al., 2017) and requires intense computational resources such as high-performance computing (Efstathiou et al., 2024).*

*Leal, A. M., Kulik, D. A., Smith, W. R., and Saar, M. O.: An overview of computational methods for chemical equilibrium and kinetic calculations for geochemical and reactive transport modeling, Pure and Applied Chemistry, 89, 597-643, 2017.*

*Efstathiou, C. I., Adams, E., Coats, C. J., Zelt, R., Reed, M., McGee, J., Foley, K. M., Sidi, F. I., Wong, D. C., and Fine, S.: Enabling high-performance cloud computing for the Community Multiscale Air Quality Model (CMAQ) version 5.3. 3: performance evaluation and benefits for the user community, Geoscientific Model Development, 17, 7001-7027, 2024.*

"Quantifying the contributions of individual processes would provide fundamental explanations for a model's predictions, and therefore is also useful in identifying potential sources of error in the model formulation or its inputs (Liu et al., 2010)." --> I find this introduction quite poor. The authors provide minimal examples of emulating

entire CTMs but give no examples of using ML to emulate and replace CTM model components which there are many for chemistry, photolysis, deposition, etc. This needs much greater discussion as it shows a lack of awareness by the authors of what currently exists, below of which are only several examples:

Krasnopolsky, V. M., Fox-Rabinovitz, M. S., and Chalikov, D. V.: New Approach to Calculation of Atmospheric Model Physics: Accurate and Fast Neural Network Emulation of Longwave Radiation in a Climate Model, Monthly Weather Review, 133, 1370–1383, https://doi.org/10.1175/MWR2923.1, 2005.

Kelp, M. M., Jacob, D. J., Lin, H., and Sulprizio, M. P.: An Online-Learned Neural Network Chemical Solver for Stable LongTerm Global Simulations of Atmospheric Chemistry, Journal of Advances in Modeling Earth Systems, 14, e2021MS002926, https://doi.org/10.1029/2021MS002926, _eprint: https://onlinelibrary.wiley.com/doi/pdf/10.1029/2021MS002926, 2022.

Xia, Z., Zhao, C., Du, Q., Yang, Z., Zhang, M., and Qiao, L.: Advancing Photochemistry Simulation in WRF-Chem V4.0: Artificial Intelligence PhotoChemistry (AIPC) Scheme with Multi-Head Self-Attention Algorithm, https://www.authorea.com/users/816476/articles/1217166-advancing-photochemistry-simulation-in-wrf-chem-v4-0-artificial-intelligence-photochemistry-aipc-scheme-with-multi-head-self-attention-algorithm, 2024.

Zhong, X., Ma, Z., Yao, Y., Xu, L., Wu, Y., and Wang, Z.: WRF–ML v1.0: a bridge between WRF v4.3 and machine learning parameterizations and its application to atmospheric radiative transfer, Geoscientific Model Development, 16, 199–209, https://doi.org/10.5194/gmd16-199-2023, publisher: Copernicus GmbH, 2023.

Silva, S. J., Heald, C. L., Ravela, S., Mammarella, I., and Munger, J. W.: A Deep Learning Parameterization for Ozone Dry Deposition Velocities, Geophysical Research Letters, 46, 983–989, https://doi.org/10.1029/2018GL081049, tex.copyright: ©2018. American Geophysical Union. All Rights Reserved., 2019.

**Response**: Given the suggestive comments from reviewer, we have added an independent paragraph in the introduction, to analyze related studies and progress, as follows,

*Quantifying individual processes would provide fundamental explanations for a model's predictions, and therefore is also useful in identifying potential sources of error in the model formulation or its inputs (Liu et al., 2010). With the motivation, there are studies dedicated to develop model to learn one specific atmospheric process, i.e. chemical and deposition, in the CTM model. Kelp et al. (2022) developed a neural network chemical solver for stable long‑term global simulations of atmospheric chemistry, learned from the GEOS-Chem model. Xia et al. (2024) simulated 74 chemical species and 229 reactions following the SAPRC-99 mechanism with an artificial intelligence photochemistry (AIPC) scheme to achieve ~8 time speed-up. Sturm and Wexler (2020) developed a mass- and energy-conserving framework for*

*using machine learning to speed computations with an successful application in a photochemistry example. For the deposition process, Silva et al. (2019) proposed a deep learning parameterization for ozone dry deposition velocities with accurate predictions in independent new date sets, revealing the potential of neural network in encoding complex spatio-temporal latent processes. Liu et al. (2025) proposed a Neural Network Emulator, named ChemNNE, for fast chemical concentration modelling, which achieved good performance in accuracy and efficiency. Even though these successful applications using deep learning methods to simulate individual atmospheric chemical and physical processes, there is an missing gap in coupling these NN operator replacements together as an complete deep learning based CTM.*

"process analysis" --> I don't know what this means

**Response**: Revised to "*internal chemical and physical process analysis*".

" Interpretations of the FastCTM are also widely vowed to improve deep learning model applications in earth system science and climate studies. " --> Not sure how you can claim this given no evidence, more aspirational than substantive

**Response**: We are sorry that this is a written error. It's revised as follows, "*Interpretations of deep learning network are also widely vowed to improve their applications in earth system science and climate studies.*".

"The FastCTM is currently configured to simulate hourly concentrations of 10 pollutant variables, including and major species of PM2.5 (SO4 2−, NO3 −, NH4+, organic matters and other inorganic components, coarse part in PM10, CO, NO2, SO2 and O3." --> Not sure how many atmospheric chemists and climate scientists want a CTM with only ten species. Needs much more motivation. Even small chemical mechanisms in operational use have around ~70 species.

**Response**: We sincerely appreciate the reviewer's comment, as it raises an important point regarding the limited number of pollutant species in the FastCTM model. FastCTM is designed to address real-time air quality forecasting, where operational utility are critical. The 10 species were selected based on their direct relevance to regulatory standards (e.g., $PM_{2.5}$, $PM_{10}$, $O_3$, $NO_2$, $SO_2$, CO) and their dominance in driving health and environmental impacts in urban and industrial regions (e.g., China, where $PM_{2.5}$ components like $SO_4^{2-}$, $NO_3^-$, and $NH_4^+$ account for most part of fine aerosol mass). By prioritizing these species, FastCTM balances accuracy with computational speed, making it suitable for rapid decision-making in policy and emergency response scenarios. While traditional CTMs (e.g., CMAQ) include ~70 species for comprehensive chemical analysis, operational forecasts often focus on criteria pollutants and key $PM_{2.5}$ components due to their regulatory importance.

FastCTM replicates the outputs most frequently used in air quality management, ensuring compatibility with existing regulatory frameworks. Besides, FastCTM's performance was rigorously validated against both CMAQ simulations and ground observations (Sect. 3.1–3.2). Results show high agreement for all 10 species ($R^2 = 0.7$–0.9), confirming that the selected variables adequately represent key atmospheric processes. We acknowledge that FastCTM may benefit from expanded mechanisms with detailed gas-phase chemistry or aerosol microphysics. FastCTM's design supports incremental integration of additional species (e.g., via user-defined modules) without overhauling the core framework. Future versions will explore adding VOCs and secondary organics to address broader research needs.

We will clarify the motivation for the 10-species configuration in the Introduction and Section 2.1 and Section 4, emphasizing regulatory and operational priorities driving species selection and plans for modular expansion in future work.

Methods:

"CMAQ structures" --> I don't know what structures means here

-Is this predicting only surface level concentrations?

-I would not really call this model a CTM, this feels more like a reduced order model. There are potentially hundreds of chemical species that lead to the formation of PM2.5, O3, etc. And yet you do not mention the chemical mechanism at all in the WRF-CMAQ model. This work is basically mapping emissions to concentrations in a fairly naive way.

**Response**: To rule out the possibilities of FastCTM as a simple model mapping emissions to concentrations, we tested land use regression (LUR) framework with machine learning models of random forest, XGBoost, and also a linear regression model. The input data for these LUR models include emissions, meteorological forecasts from WRF, and geophysical covariates, same to that used in FastCTM. The LUR model carries out direct mapping from emission and weather data to 10 pollutants. Results have exhibited LUR's poor performance in predicting air pollutant concentrations. Related studies are included in Section 3.1, as follows.

*In addition, three land use regression (LUR) models were constructed, namely the linear regression model, the random forest model (with the number of trees set at 500), and the XGBoost model (with the booster specified as gbtree). These LUR models were developed using the same input meteorological, emission, and geophysical covariates. When compared with the FastCTM model, the performance of the LUR models was found to be significantly inferior (as demonstrated in Figure S6 - S8 in the Supplementary Information). This outcome is, in fact, anticipated when we consider the complex nature of air quality dynamics. Air quality is not a static entity, but it varies both spatially and temporally. For instance, the transport of air pollution is a highly dynamic process that hinges on wind fields and air pollution concentrations in a*

*reciprocal manner. The wind direction and speed dictate the trajectory along which pollutants travel, while the existing pollutant concentrations in different regions influence the overall dispersion and mixing patterns. LUR models, which predominantly rely on local input data (Wong et al., 2021; Cheng et al., 2021), struggle to capture these intricate, non-local interactions. They lack the capacity to account for the far-reaching effects such as wind-driven pollutant transport and the consequential changes in air quality over larger geographical areas.*

*With the supplementary Figures exhibiting performances of three machine learning models as follows.*

[Figure]

*Figure S9: The evaluation performances of linear regression forecasts against CMAQ forecasts in 2023. Panel (a) and (b) respectively show RMSE values of criteria pollutants and the PM$_{2.5}$ components of. Panel (c) and (d) respectively show R$^2$ values. It should be noted that RMSE value of CO corresponds to the right axis in panel (a).*

[Figure]

*Figure S10: The evaluation performances of random forest forecasts against CMAQ forecasts in 2023. Panel (a) and (b) respectively show RMSE values of criteria pollutants and the PM$_{2.5}$ components of. Panel (c) and (d) respectively show R$^2$ values. It should be noted that RMSE value of CO corresponds to the right axis in panel (a).*

[Figure]

*Figure S11: The evaluation performances of XGboost forecasts against CMAQ forecasts in 2023. Panel (a) and (b) respectively show RMSE values of criteria pollutants and the PM$_{2.5}$ components of. Panel (c) and (d) respectively show R$^2$ values. It should be noted that RMSE value of CO corresponds to the right axis in panel (a).*

" A detailed description of CMAQ principles is available elsewhere (Byun and Schere, 2006) " --> I find this lazy. This paper is 20 years old and I do not know what you would like the reader to find in it.

**Response**: The reference provided the detailed description about the theory, model framework and numerical methods. We added one another lately review studies to reflect recent developments of chemical transport model CMAQ.

*Appel, K. W., Napelenok, S. L., Foley, K. M., Pye, H. O., Hogrefe, C., Luecken, D. J., Bash, J. O., Roselle, S. J., Pleim, J. E., and Foroutan, H.: Description and evaluation of the Community Multiscale Air Quality (CMAQ) modeling system version 5.1, Geoscientific model development, 10, 1703-1732, 2017.*

"Chemical Reaction Module" --> This just sounds like a first order approximation using idealized rate constants. There is a very rich and long history of using ODE solvers to get the solution to complex chemical mechanisms. There really is not enough discussion with this module (or really any of the preceding module sections). You are highly simplifying each of these processes without an underlying discussion of why you are doing so. There already exist data-driven and reduced complexity modeling systems that accomplish similar air quality regulation goals (e.g., InMAP, EASIUR, APEEP).

**Response**: The simplification of chemical kinetics in FastCTM is motivated by balancing data availability with physical interpretability. While traditional CTMs (e.g., CMAQ) use detailed ODE solvers for hundreds of species and reactions, FastCTM focuses on key variables and pathways for air quality dynamics, such as secondary inorganic aerosol formation ($SO_4^{2-}$, $NO_3^-$, $NH_4^+$) and ozone photochemistry.

Comparing to other reduced-form modelling systems, models like InMAP and EASIUR focus on annual-average exposure, while FastCTM provides hourly-resolved simulations critical for real-time management. Unlike reduced-form models that aggregate source impacts, FastCTM quantifies hourly contributions from individual processes (transport, chemistry, emissions) via its modular design. Furthermore, FastCTM explicitly couples meteorology (PBLH, T, RH) with chemistry, whereas InMAP/APEEP assume static meteorology, limiting their utility in capturing diurnal or synoptic-scale variations. Therefore, FastCTM is more like a learnable CTM model in the neural network form with some simplifications in input variables and space domain (3D to 2D). By embedding physical principles (e.g., mass conservation in transport, Arrhenius-like rate dependencies in chemistry) within a neural framework. Besides, the modular architecture of FastCTM allows incremental addition of species/reactions (e.g., VOC oxidation pathways) without retraining the entire model. Related discussion have been added in the section,

*Reduced-form models like InMAP (Tessum et al., 2017) and EASIUR (Gentry et al., 2023) focus on annual-average exposure, while FastCTM provides hourly-resolved simulations critical for real-time management. FastCTM quantifies hourly*

*contributions from individual processes (transport, chemistry, emissions) via its modular design, rather than aggregating source impacts (e.g., EASIUR's source-receptor matrices) in reduced-form models. Furthermore, FastCTM explicitly couples meteorology (PBLH, T, RH) with chemistry, whereas InMAP/APEEP (Muller and Mendelsohn, 2006) assume static meteorology, limiting their utility in capturing diurnal or synoptic-scale variations.*

And also in the Section 4,

*Besides, FastCTM may also benefit from expanded mechanisms with detailed gas-phase chemistry or aerosol microphysics. FastCTM's design supports incremental integration of additional species (e.g., via user-defined modules) without overhauling the core framework. Future versions will explore adding VOCs and secondary organics to address broader research needs.*

Also for the diffusion module and deposition module, related discussions have been added in the corresponding sections as follows,

*Diffusion involves the physical and chemical processes that disperse pollutants in the atmosphere. It's influenced by meteorological conditions, i.e. atmospheric stability and humidity, and surface features, i.e. land terrains and vegetation (Jiang et al., 2021).*

*Air pollutant deposition refers to the process by which atmospheric pollutants are transferred to Earth's surfaces (land, water, vegetation) or removed from the air. This phenomenon plays a critical role in environmental pollution dynamics and ecosystem impacts. The deposition was closely influenced by meteorological conditions and surface characteristics (Janhäll, 2015). For example, high wind disperses pollutants, while turbulence enhances dry deposition. Forests and crops act as sinks due to large surface areas for adsorption.*

-I don't explicitly understand how this is a machine learning model. You describe a sequence-to-sequence modeling framework reminiscent of an LSTM, but no mention of memory or hyper parameters in general. The inclusion of these equations may seem more like a symbolic regression kind of ML framework, but the details are sparse and lack substance. Are all the modules trained jointly so that error influences each other? Chemistry is constantly affected by other modules (and vice versa) yet these interaction terms can't be considered during training at all. That is, how does error propagate from one time step to the next in training? Is the underlying WRF-CMAQ simulations two-way coupled such that weather influences chemistry and chemistry feedbacks via aerosol effects to influence the weather? Not enough details in the underlying simulations or the joint training of modules. There are examples of this kind of offline training here:

Kelp, M. M., Jacob, D. J., Kutz, J. N., Marshall, J. D., and Tessum, C. W.: Toward Stable, General Machine-Learned Models of the Atmospheric Chemical System, Journal of Geophysical Research: Atmospheres, 125, e2020JD032759,

https://doi.org/10.1029/2020JD032759, 2020.

Yang, X., Guo, L., Zheng, Z., Riemer, N., and Tessum, C. W.: Atmospheric chemistry surrogate modeling with sparse identification of nonlinear dynamics, https://doi.org/10.48550/arXiv.2401.06108, 2024.

Liu, Z.-S., Clusius, P., and Boy, M.: Neural Network Emulator for Atmospheric Chemical ODE, https://doi.org/10.48550/arXiv.2408.01829, 2024.

**Response**: (1) The five modules in FastCTM are defined in the form of operator, where operator parameters are estimated, rather than in the form of pure predictor mapping concentrations from one hour to the next. For example, in the diffusion module, FastCTM learns to encode diffusion coefficient $K$ from meteorological conditions before performing upwind finite difference procedure to solve diffusion process $\nabla(K\nabla C_i)$. It's also same for processes such as reaction, advection, and deposition. Therefore, it is not possible for one process to represent all atmospheric processes simultaneously. The independent contribution of each process depicted in Figure 12 of section 3.3. Clearly, each process exhibited their patterns of contribution to hourly air pollutant concentrations changes, constrained by the form of the operator in the processes. Related description was added in the Section 2.3 Model Training,

*Even though five modules are defined in FastCTM, individual processes are not trained separately. The model was trained as a whole with hour-to-hour air pollutant concentrations, while each process could learn its parameters under the constrains of its dedicated formulation. Specifically, FastCTM was tuned to minimize the loss function $\mathcal{L}$, which was determined to be L2 loss (Bühlmann and Yu, 2003) of the regularized mean squared error (MSE) as shown in Eq. 15.*

(2) The configuration of parent model was added in the Section 2.1

*WRF-CMAQ simulations are not two-way coupled so that weather and chemistry and chemistry do not have feedbacks to influence each other.*

"The main objective of our study is to build and validate a principles-guided neural network based FastCTM that could simulate spatial-temporal fields of hourly concentrations of major air pollutant species like a traditional CTM. Besides, the FastCTM could model individual contributions from each of the atmospheric processes of transport, diffusion, deposition, reaction and emission. " --> this should be stated earlier. The term "principles-guided" is vague, and I don't really consider this 'like a traditional CTM'. You discretize the potential processes that affect air quality outputs, but this is more like a traditional reduced complexity model approach. I think a deeper review into the literature would help the authors situate their work in this established landscape.

**Response**: These two sentences are moved to the last paragraph of the Introduction section, to make it clearer for the general purpose of the study. Unlike the traditional

reduced-from models, FastCTM is time-resolved with a 60 seconds step to simulate evolvement of air pollutants. It generates hourly air quality simulations given hourly meteorological conditions and emissions. The simulations have good correlations with its parent numerical CTM model CMAQ. FastCTM also exhibited reasonable responses to emission interventions, also in close agreement to that of CMAQ. Besides, internal atmospheric processes could also be checked to reflect specific contributions from each process. We have added related literature review on the application of neural network in simulating atmospheric processes in the introduction, as follows.

*Quantifying individual processes would provide fundamental explanations for a model's predictions, and therefore is also useful in identifying potential sources of error in the model formulation or its inputs (Liu et al., 2010). With the motivation, there are studies dedicated to develop model to learn one specific atmospheric process, i.e. chemical and deposition, in the CTM model. Kelp et al. (2022) developed a neural network chemical solver for stable long‑term global simulations of atmospheric chemistry, learned from the GEOS-Chem model. Xia et al. (2024) simulated 74 chemical species and 229 reactions following the SAPRC-99 mechanism with an artificial intelligence photochemistry (AIPC) scheme to achieve ~8 time speed-up. Sturm and Wexler (2020) developed a mass- and energy-conserving framework for using machine learning to speed computations with an successful application in a photochemistry example. For the deposition process, Silva et al. (2019) proposed a deep learning parameterization for ozone dry deposition velocities with accurate predictions in independent new date sets, revealing the potential of neural network in encoding complex spatio-temporal latent processes. Liu et al. (2025) proposed a Neural Network Emulator, named ChemNNE, for fast chemical concentration modelling, which achieved good performance in accuracy and efficiency. Even though these successful applications using deep learning methods to simulate individual atmospheric chemical and physical processes, there is an missing gap in coupling these NN operator replacements together as an complete deep learning based CTM.*

Also in the Section 2.3.5 Chemical Reaction Module, comparison of FastCTM to reduced-form models are discussed as follows,

*Reduced-form models like InMAP (Tessum et al., 2017) and EASIUR (Gentry et al., 2023) focus on annual-average exposure, while FastCTM provides hourly-resolved simulations critical for real-time management. FastCTM quantifies hourly contributions from individual processes (transport, chemistry, emissions) via its modular design, rather than aggregating source impacts (e.g., EASIUR's source-receptor matrices) in reduced-form models. Furthermore, FastCTM explicitly couples meteorology (PBLH, T, RH) with chemistry, whereas InMAP/APEEP (Muller and Mendelsohn, 2006) assume static meteorology, limiting their utility in capturing diurnal or synoptic-scale variations.*

"Furthermore, CMAQ and FastCTM forecasts were both evaluated by hourly

observations from national monitoring sites (as shown in Figure S5 in the supplementary material) for six criteria pollutants (PM2.5, PM10, SO2, NO2, CO, and O3)." --> What is the point of this if CMAQ is your ground truth?

**Response**: We agree with the reviewer's point that it does not make much sense to compare FastCTM to station observations. Related comparisons are removed from the manuscript.

Results:

"Besides, since the FastCTM is a 2-D model only considering atmospheric processes within the boundary layer, lower consistency with the CMAQ model during daytime could be due to more active vertical turbulence which is not fully represented." --> Isn't the point of having this processed-based emulation the ability to attribute errors to processes? This sounds hand-wavy and does not explain the variability very well

**Response**: We sincerely appreciate the reviewer's insightful feedback regarding the attribution of errors in FastCTM's daytime performance. The comment highlights a critical aspect of our process-based emulation framework and motivates a deeper exploration of error sources. FastCTM could simulate the contributions from each processes. We do not have the CMAQ process analysis in the test period. Therefore, it is not possible to attribute FastCTM's simulation errors to specific processes, by comparing process data of CMAQ. CMAQ uses a non-local closure scheme for vertical diffusion, explicitly resolving turbulent mixing across layers. FastCTM's 2D framework parameterizes this via horizontal diffusivity and PBLH, which cannot capture vertical advection or entrainment. Besides, we considered a 2-D model in FastCTM, which means processes analysis could be different to that of CMAQ in its definition natures. We added further explanation in this section as follows,

*Besides, since the FastCTM is a 2-D model only considering atmospheric processes within the boundary layer, lower consistency with the CMAQ model during daytime could be due to more active vertical turbulence. Strong vertical mixing of air pollutants to the height above PBLH have been found (Li et al., 2017; Tang et al., 2016), which could not be not fully represented in FastCTM.*

"It is important to note that the relatively low $R^2$ values observed for NH4+ can be attributed to the fact that it is the sole cation included in the FastCTM model without a corresponding acid-base balance, which may affect the model's predictive accuracy." --> I don't see how this is the reason. WRF-CMAQ has many base pairings that can neutralize NH4+ that are not represented here. I don't recall conservation of mass as a constraint in your chemical module. Furthermore, how do you know that NH4+ does not precipitate out as it is very hydrophilic.

**Response**: We appreciate the reviewer's critical assessment of the $NH_4+$ prediction

performance and agree that our initial explanation oversimplified the issue. FastCTM's chemical module (Eq. 11–12) approximates $NH_4+$ dynamics using a data-driven approach trained on CMAQ outputs. While CMAQ explicitly resolves $NH_4+$ formation via reactions with HNO3 ($NH_3 + HNO_3 \rightarrow NH_4NO_3$) and $H_2SO_4$ ($2NH_3 + H_2SO_4 \rightarrow (NH_4)_2SO_4$), FastCTM does not explicitly encode these pathways. Instead, the neural network implicitly learns relationships between $NH_4+$ and precursor emissions ($NH_3$, NOx, $SO_2$) and meteorological variables (e.g., temperature, humidity). This simplification omits acid-base equilibria and aerosol thermodynamics, which are critical for partitioning NH4+ between gas and particle phases. The reviewer correctly notes that FastCTM's chemical module does not enforce mass conservation. While CMAQ rigorously tracks nitrogen and sulfur species across gas, aerosol, and aqueous phases, FastCTM's neural network predicts $NH_4+$ concentrations directly from emissions and meteorology without explicit mass-balance constraints. This can lead to unphysical predictions, especially when precursor emissions (e.g., $NH_3$) are over/underestimated or when thermodynamic conditions (e.g., high humidity) favor aerosol formation. The low R² for $NH_4+$ primarily reflects FastCTM's simplified chemical mechanism, which lacks explicit acid-base pairing and aerosol thermodynamics. We agree with the reviewer that attributing the error to NH4+ being a "sole cation" was incomplete and have revised the text accordingly in the Section 3.1, as follows,

*While CMAQ explicitly resolves $NH_4^+$ formation reactions, FastCTM does not explicitly encode these pathways. Instead, the neural network implicitly learns relationships between $NH_4^+$ and precursor emissions (NH3, NOx, SO2) and meteorological variables (e.g., temperature, humidity). This simplification omits acid-base equilibria and aerosol thermodynamics, which are critical for partitioning $NH_4^+$ between gas and particle phases. The low R² for $NH_4^+$ primarily reflects FastCTM's simplified chemical mechanism in this part, which could be improved by adding related species in the simulation.*

-I actually believe it is quite concerning that the RMSEs vary diurnally. You should also plot the WRF-CMAQ and FastCTM time series against each other. A diurnal error actually may suggest that you are not correctly learning the atmospheric dynamics of the system well. You may be predicting an average concentration across all time and that's why you see a diurnal error profile.

"FastCTM forecasts using zero values as input air quality data were almost the same as that using ordinary input in the long leading hours, indicating that FastCTM simulations in long leading hours are not affected by initial conditions, just like deterministic numeric CTMs (such as CMAQ)" --> This is hard to conclude, you need to plot actual concentration time series instead of RMSEs. It seems like the error is always the same, this could mean the FastCTM always predicts the same values given the time of day. More results need to be presented.

**Response**: As reviewer kindly pointed, FastCTM possibly has taken average pollutant concentration from five-year training data in 2018-2022. In order to confirm that FastCTM was able to predict air quality based on given meteorological conditions and emissions, daily average FastCTM simulation in the fifth leading day (leading hours 96-119) in the test year of 2023 is compared with daily average CMAQ simulations in 2023 and in the training years of 2018-2022. Results revealed that FastCTM forecasts generally in good correlation with CMAQ forecasts in 2023, rather than that in 2018-2023. It means FastCTM has learn the evolution rules of air pollutant concentrations, instead of just giving average air pollutant concentration according to time of the year. Related results have been added in the manuscript in the section 3.1.1, as follows.

*Annually, the daily air quality typically exhibits similar fluctuations to those in other years, which can be primarily attributed to the cyclical nature of meteorological conditions and pollutant emission patterns. The FastCTM model was trained using a comprehensive dataset spanning five years, from 2018 to 2022. In light of this, it was crucial to rule out the possibility that the model was merely reproducing historical averages during the test year of 2023. To this end, the daily national average concentrations of $PM_{2.5}$ and $O_3$ in 2023, as predicted by FastCTM, were meticulously compared with those simulated by CMAQ in the same test year, as well as with the CMAQ forecasts from the training years of 2018-2022. As illustrated in Figure 4, it becomes evident that the predictions made by FastCTM in 2023 align more closely with the actual CMAQ forecasts for that year, rather than with the forecasts generated from the training data of 2018-2022. This finding not only validates the adaptive learning capabilities of the FastCTM model but also indicates that the model is not resorting to a simplistic approach of taking the average concentration from the previous five years based on the time of day. Instead, it is likely incorporating real-time meteorological feedback, adjusting for any shifts in emission patterns, and leveraging its learned relationships to provide more accurate and contemporaneous predictions.*

[Figure]

*Figure 5: The evaluation performances of FastCTM forecasts against CMAQ forecasts in 2023.*

Figure 3 is unwieldy. There are 60 mulitplots and not well labeled on the figure. Here you should show spatial differences in terms of both absolute and relative error. Seems like FastCTM does not capture the highest concentration values, which is concerning given that is the largest impact on health and climate. Hard to have any substantive discussion of results without any quantitative measure regarding Figure 3.

**Response**: As reviewer suggested, we revised the manuscript in the section as follows, *The spatial distributions of the mean absolute error (MAE) and the normalized mean absolute error (NMAE) are presented in Figure 3. For all six pollutants under consideration, it is a notable finding that the MAE values tend to be higher in polluted areas. This can be attributed to the complex and dynamic nature of pollutant interactions in such regions. In polluted environments, there are often multiple sources of emissions, complex chemical reactions, and variable meteorological conditions that can lead to greater discrepancies between the model - predicted and actual pollutant concentrations. Conversely, the NMAE values exhibit an opposite trend, being lower in polluted areas. In these regions, the NMAE values typically hover around 0.2, in contrast to the relatively higher values of approximately 1 in cleaner areas. The NMAE is a normalized metric that takes into account the magnitude of the actual pollutant concentrations. A lower NMAE in areas with high pollution levels suggests that the FastCTM model is effectively capturing the overall magnitude and trends of pollutant concentrations relative to the reference CMAQ model.*

[Figure]

*Figure 12: Spatial distribution of mean absolute error (panel a, c, e, g, i, and k) and normalized mean absolute error for six pollutants (panel b, d, f, h, j and l) of FastCTM comparing to CMAQ in 2023.*

Section 3.1.2. Again, I don't see why this comparison makes sense. You do not incorporate any station data, so why would you make comparisons against it? The WRF-CMAQ model is the ground truth here.

**Response**: We agree with the reviewer's point that related comparisons between FastCTM to station observations are removed from the manuscript.

Sections 3.2: These don't have much meaning if we do not understand how the FastCTM model behaves in relation to the parent model

**Response**: We agree with reviewer's comment that more analysis needed to verify FastCTM's capabilities to project the impact of a change in emissions, especially comparing to the parent model of WRF-CMAQ. We added comparison between FastCTM and CMAQ in simulating 11 emission intervention scenarios in the winter month of January 2019 and in summer month of July 2019. The results signified that, on the whole, the FastCTM simulations manifested a high level of concordance with those of CMAQ, which was manifested in two principal aspects. Firstly, akin to CMAQ, the FastCTM model forecasted positive responses to augmented emissions in the no-control (NCtrl) scenario and negative responses in the other emission-controlled scenarios. This implies that when emissions were unrestricted and increased, as in the NCtrl scenario, both models detected a corresponding upward trend in pollutant levels. Conversely, in scenarios where emissions were curbed, they both predicted a decline. Secondly, in scenarios characterized by more substantial emission reductions, the FastCTM model simulated a more pronounced decrease in air pollutant concentrations. This is of particular significance as it showcases the model's sensitivity to the magnitude of emission interventions. It suggests that the FastCTM model is not only capable of discerning changes in emission scenarios but can also reflect the degree of impact on air quality, thereby reinforcing its reliability and utility in simulating air quality dynamics in tandem with CMAQ. Related results in the manuscript were shown as follows.

*The sensitivities of FastCTM simulations to emission interventions were contrasted with those of CMAQ. Specifically, CMAQ was employed to simulate 11 emission scenarios over the two-month periods of January and July 2019 in Southwest China. The alterations in emissions relative to the base case are presented in Table 1. Among these scenarios, 10 involved reduced emissions of major species, with only the no-control scenario exhibiting increased emissions. Utilizing the identical emissions and meteorological data, FastCTM also conducted simulations, which were then compared to those of CMAQ. For the 11 scenarios in question, the changes in air pollutant concentrations relative to the base case at the locations of 139 national air quality monitoring stations (Figure S14 in the SI) were extracted and compared in the winter month of January 2019 (Figure 9) and in summer month of July 2019 (Figure 10). The results indicated that, overall, the FastCTM simulations were in good agreement with those of CMAQ reflected in two aspects. First, FastCTM predicted positive responses to increased emissions in the nocontrol (NCtrl) scenario and negative responses to other emission-controlled scenarios just as CMAQ. Second, FastCTM simulated larger air pollutant concentration decrease in those scenarios with higher emission reductions. Specifically, in January 2019, with the exception of $NO_2$, FastCTM responded to emission changes with an interquartile range (IQR, 25% - 75% percentile) similar to that of CMAQ (Figure 9). For $NO_2$, in the same emission reduction scenarios, FastCTM*

simulated lower NO₂ values. In the summer month of July 2019, as depicted in Figure 10, all the criteria pollutants except CO demonstrated a comparable degree of response to emission reductions. The comparison suggests that the FastCTM model is not only capable of discerning changes in emission scenarios but can also reflect the degree of impact on air quality, thereby reinforcing its reliability and utility in simulating air quality dynamics in tandem with CMAQ. It should be noted that in both months, FastCTM exhibited slightly larger median values, suggesting its greater sensitivity to emission interventions.

**Table 2**. The emission change details of emission scenarios

| Scenario | abbreviation | Sector | $NO_x$ | VOCs | $SO_2$ | CO | $PM_{2.5}$ | PMC |
|---|---|---|---|---|---|---|---|---|
| nocontrol | NCtrl | Industrial | 30% | 30% | 30% | 30% | 30% | 30% |
| | | Traffic | 20% | 20% | 20% | 20% | 20% | 20% |
| medianX | MedX | Industrial | -36% | -35% | -48% | -23% | -9% | -9% |
| | | Traffic | -40% | -10% | 0 | -26% | -10% | -10% |
| medianY | MedY | Industrial | -26% | -20% | -38% | -13% | -4% | -4% |
| | | Traffic | -30% | 0% | 0 | -16% | -5% | -5% |
| medianZ | MedZ | Industrial | -36% | -10% | -48% | -23% | -9% | -9% |
| | | Traffic | -40% | 0% | 0 | -26% | -10% | -10% |
| median-3 | Med-3 | Industrial | -10% | -10% | -18% | 0 | 0 | 0 |
| | | Traffic | -10% | 0% | 0 | 0 | 0 | 0 |
| median-2 | Med-2 | Industrial | -16% | -20% | -28% | -3% | 0 | 0 |
| | | Traffic | -20% | 0% | 0 | -6% | 0 | 0 |
| median-1 | Med-1 | Industrial | -26% | -35% | -38% | -13% | -4% | -4% |
| | | Traffic | -30% | -10% | 0 | -16% | -5% | -5% |
| median0 | Med0 | Industrial | -36% | -50% | -48% | -23% | -9% | -9% |
| | | Traffic | -40% | -20% | 0 | -26% | -10% | -10% |
| median+1 | Med+1 | Industrial | -46% | -65% | -58% | -33% | -19% | -19% |
| | | Traffic | -50% | -30% | 0 | -36% | -20% | -20% |
| median2030 | Med30 | Industrial | -55% | -70% | -80% | -40% | -40% | -40% |
| | | Traffic | -60% | -40% | 0 | -40% | -40% | -40% |
| | Med35 | Industrial | -80% | -80% | -90% | -60% | -50% | -50% |

| median2035 | | Traffic | -80% | -60% | 0 | | -60% | -50% | -50% |
|---|---|---|---|---|---|---|---|---|---|

[Figure]

*Figure 9: Air pollutant concentration changes in terms of base case simulated by CMAQ (subplots of a, c, e, g, i and k in first column) and by FastCTM (subplots of b, d, f, h, j and l in second column) in January 2019.*

[Figure]

*Figure 10: Air pollutant concentration changes in terms of base case simulated by CMAQ (subplots of a, c, e, g, i and k in first column) and by FastCTM (subplots of b, d, f, h, j and l in second column) in July 2019.*

Figure 8: These color bars are difficult to discern changes in concentrations. Does adding d through h yield panels a or b? Again, individual contribution doesn't matter if we don't know how the model actually behaves.

**Response**: Adding panel d through h yield panel c. To validate the process analysis by FastCTM, its simulation results are compared to that by CMAQ. We added related results and discussion in Section 3.3 as follows,

In this study, we further selected the data recorded at 23:00 on October 13, 2024, to compare the impacts of the five major atmospheric physical and chemical processes, as simulated by FastCTM and CMAQ, on PM$_{2.5}$ concentration changes (Figure 12). Specifically, the simulation outcomes of atmospheric emissions, advection processes, and diffusion processes demonstrated a relatively high degree of consistency between the two models. Regarding the simulation of chemical reactions, while the spatial distribution of high-value areas in the FastCTM results was comparable to that of CMAQ, the simulated values in FastCTM were notably higher. Correspondingly, FastCTM overestimated the contribution of the deposition process. This overestimation counterbalanced the impact of the higher chemical reaction values. The difference in the simulated deposition contributions between the two models could be due to differences in how they represent these influencing factors. Overall, the simulation results of the process contributions by FastCTM and its parent model CMAQ were relatively consistent. This consistency indicates that, despite some differences in the magnitude of certain process simulations, FastCTM is capable of capturing the essential features of atmospheric processes related to PM$_{2.5}$ concentration changes, similar to CMAQ. Such consistency provides confidence in the reliability of FastCTM for simulating and understanding the complex interplay of atmospheric processes that govern PM$_{2.5}$ levels.

[Figure]

**Figure 13: An example of contributions from five major atmospheric processes to PM$_{2.5}$ changes (µg/m$^3$) by CMAQ (first row) and FastCTM (second row)  in 23:00 on October 13, 2024.**

---

## Author Response (AR2)

**Point-to-point Responses to Reviewers' Comments**

**Review #1**

As this is a second review, I have focused my efforts on the elements changed in response to my original review.

First, I appreciate the effort put in by the reviewers to address many of my previous concerns. The comparison to land use regression models and to the average CMAQ output is welcome, as is the additional clarity on how R2 and RMSE are calculated. I was particularly happy to see that the authors have conducted paired sensitivity tests with CMAQ and FastCTM.

However, I was disappointed to see that the comparison between FastCTM and the LUR methods appears to be entirely qualitative, with no numerical assessment of the relative performance. Furthermore, the extremely low R2 values achieved by land use regression are surprising. Studies such as Zhang et al (2021) have achieved R2 values of ~0.8 with simple generalized linear models. For this comparison to be useful, I recommend that the authors first establish that their point of comparison (the land use regression model) is a fair representation of recent efforts in this area, achieving reasonable levels of accuracy. It would then be more helpful to compare the accuracy of their LUR models to CMAQ quantitatively, perhaps using the suite of metrics already recommended for air quality comparison by Huang et al. (2021). Said metrics could also be used as a point of comparison for FastCTM against the literature.

**Response**: We have added statistical metrics of R2, NME and NMB for comparing FastCTM and LUR models as the reviewer suggested. The LUR model used in this study (random forest and XGBoost) represent widely-used approaches in recent literature, as demonstrated in the comprehensive review of popular LUR models by Ma et al. (2024).. The related revision in lines 313-316 are as follows,

*"These LUR models were developed using the same input meteorological data, emission, and geophysical variables as FastCTM to ensure fair comparison. When compared with the FastCTM model, the performance of the LUR models was found to be significantly inferior as demonstrated in the Table. 1 and Figure S10 – S12 in the SI. For example, $R^2$ values for FastCTM range from 0.68-0.90, whereas the LUR models only achieve 0.06-0.33."*

*Table 1. Performance metrics of LUR models and FastCTM compared against CMAQ*

| Variable | Model | RMSE | $R^2$ | NMB |
|----------|-------|------|-------|-----|
| $PM_{2.5}$ | FastCTM | 8.78 | 0.81 | -0.15 |
| | Liner Model | 35.05 | 0.09 | -0.24 |
| | Random Forest | 33.08 | 0.19 | -0.25 |
| | XGBoost | 33.02 | 0.14 | -0.12 |
| $PM_{10}$ | FastCTM | 11.58 | 0.80 | -0.17 |
| | Liner Model | 44.66 | 0.10 | -0.23 |
| | Random Forest | 45.07 | 0.19 | -0.33 |
| | XGBoost | 44.53 | 0.15 | -0.21 |
| $SO_2$ | FastCTM | 4.51 | 0.80 | 0.09 |
| | Liner Model | 39.42 | 0.14 | -1.18 |
| | Random Forest | 25.74 | 0.33 | -0.65 |
| | XGBoost | 25.57 | 0.26 | -0.60 |
| $NO_2$ | FastCTM | 4.24 | 0.83 | 0.04 |
| | Liner Model | 21.42 | 0.27 | -0.30 |
| | Random Forest | 25.13 | 0.16 | -0.58 |
| | XGBoost | 23.88 | 0.15 | -0.43 |
| CO | FastCTM | 51.84 | 0.90 | 0.01 |
| | Liner Model | 427.67 | 0.03 | 6.38 |
| | Random Forest | 83.25 | 0.08 | 1.32 |
| | XGBoost | 70.06 | 0.06 | 1.10 |
| $O_3$ | FastCTM | 11.46 | 0.68 | 0.02 |
| | Liner Model | 357.97 | 0.09 | -0.46 |
| | Random Forest | 285.16 | 0.19 | -0.21 |
| | XGBoost | 291.58 | 0.15 | -0.22 |

However, through in-depth literature search and review, we found that LUR models have been seldom used in the way that the CTM models have been used for air quality simulations and forecasts. CTM models are able to simulate air pollutant concentrations given initial conditions, meteorological conditions and emissions, etc. However, LUR models are typically used to estimate air pollutant concentrations across complete spatial fields by interpolating from discrete station observations using land use and other covariates as predictors. For example, Zhang et al. (2021) used a GLM-based LUR model to predict 10km resolution gridded daily

concentrations of six criteria pollutants in Beijing, supervised by observations from 35 stations. Similarly, Wong et al. (2021) predicted daily PM$_{2.5}$ concentration at 50m resolution in Taiwan using observations from 73 monitoring sites.

LUR models face significant challenges in accurately predicting air pollutant concentrations when relying solely on static relationships between pollutant concentrations and supporting variables, particularly without corresponding in-situ observations. This challenge becomes especially pronounced for hourly air pollutant predictions, as concentrations can change rapidly between consecutive hours even when meteorological conditions show minimal variation. Consequently, there are virtually no successful studies demonstrating LUR models' capability for this type of application, as confirmed by the comprehensive review by Ma et al. (2024)."

Ma, X., Zou, B., Deng, J., Gao, J., Longley, I., Xiao, S., Guo, B., Wu, Y., Xu, T., Xu, X., Yang, X., Wang, X., Tan, Z., Wang, Y., Morawska, L., and Salmond, J.: A comprehensive review of the development of land use regression approaches for modeling spatiotemporal variations of ambient air pollution: A perspective from 2011 to 2023, Environment International, 183, 108430, https://doi.org/10.1016/j.envint.2024.108430, 2024.

Similarly, the comparison of FastCTM versus the simple average of CMAW 2018-2022 does not contain any quantitative assessment. The authors state on the basis of line graphs in Figure 5 that "it becomes evident that the predictions made by FastCTM in 2023 align more closely with the actual CMAQ forecasts", but that is an assertion and not evidence. While I am inclined to agree in the case of PM2.5, the performance for ozone does not appear to be particularly improved – however without any statistical assessment it is hard to say. I am also concerned by the significant gaps in the data shown in Figure 5. Why are some days simply not shown for CMAQ 2023 or FastCTM? These gaps must be explained to ensure a fair and transparent comparison.

**Response**: We added quantitative assessment for the difference and correlations between FastCTM and CMAQ simulations. The related description was revised in this section (Lines 335-342) as follows.

"*As illustrated in Figure 5, the predictions made by FastCTM in 2023 align more closely with the actual CMAQ forecasts for that year with $R^2$ = 0.94 and 0.72 respectively for PM$_{2.5}$ and O$_3$, rather than with the forecasts generated from the training data of 2018-2022 with $R^2$=0.54 and 0.59. The NMB was also lower between FastCTM and CMAQ for the same year 2023. These*

*results not only validate the adaptive learning capabilities of the FastCTM model but also indicate that the model is not using a simplistic approach of averaging concentrations from the previous five years based on time of day. Hourly time series plots of air pollutant concentrations (Figure S6 in the SI) further demonstrate that FastCTM appears to incorporate real-time meteorological feedback, adjust for shifts in emission patterns, and leverage its learned relationships to provide more accurate and contemporaneous predictions.*"

Regarding the missing gaps, these occur because the CMAQ simulations are incomplete due to data unavailability. We have added an explanation for this in the caption of Figure 5 and the Model Evaluation section as follows.

*"FastCTM was assessed against CMAQ simulations using the same input emission data and meteorological fields. Starting from 0:00 local time on each day, the CMAQ model simulated 120-hour forecasts in one cycle. There are 139 cycles in the evaluation year of 2023 due to data unavailability in the remaining days."*

[Figure]

***Figure 1: The daily FastCTM forecasts compared with CMAQ forecasts, respectively in the training period of 2018-2022 and evaluation period of 2023 for (a) PM$_{2.5}$ and (b) O$_3$. The gaps for FastCTM and CMAQ in 2023 are due to data unavailability in these days.***

The lack of quantitative analysis further extends to the valuable paired sensitivity tests comparing the response of CMAQ and FastCTM to emissions perturbations (second half of Section 3.2). Showing 24 bar charts (each with 11 box-and-whisker elements) side by side is certainly transparent, but it is very difficult to assess the degree to which the model is better or worse at predicting certain changes. The analysis only states that changes are "comparable", have "similar" IQRs, or are in "good agreement". What is needed is a robust comparison which seeks to identify the conditions under which FastCTM performs well (as defined by some reasonable quantitative standard), and when it performs poorly. For example, it seems that NO2 benefits are routinely overestimated whereas changes in SO2 seem to be generally underestimated, but with some dependence on the specific scenario. In many ways the comparisons for ozone and PM2.5 are encouraging, suggesting that FastCTM may have more

skill in predicting the response to policy than it does in predicting baseline concentrations. These would be very useful features to understand quantitatively, but as it stands the manuscript falls short by providing almost no numerical analysis of the model's performance for this test.

**Response**: As the reviewer suggested, we added quantitative analysis of the FastCTM's responses to emission changes under different scenarios. Specifically, correlation coefficients (R) were calculated for each air pollutant and each scenario at 139 stations to evaluate consistency between FastCTM and CMAQ in simulating emission interventions. Meanwhile, the boxplots of responses in Figures 9 and 10 have been moved to the supplementary material. The comparative analysis between FastCTM and CMAQ has been revised accordingly to better reflect FastCTM's capabilities in replicating CMAQ model performance, as follows,

*"The results indicated that, overall, the FastCTM simulations due to emissions changes were in good agreement with those of CMAQ, as reflected in two aspects. The correlation coefficient R values are around 0.9 for $SO_2$, $NO_2$ and $O_3$ in both summer and winter months. For $PM_{2.5}$ and $PM_{10}$, FastCTM exhibited higher consistency with CMAQ in July than in January, with R values around 0.6 for most cases. For CO, FastCTM has much better performance in January than in July, with R values of approximately 0.8 and 0.2. Considering that CO concentration changes are mostly due to physical dispersion and transport, the decreased performance is probably due to increased vertical mixing in summer, which is not fully represented in the 2D scheme of FastCTM. Specifically, in January 2019, except $NO_2$, FastCTM responded to emission changes with an interquartile range (IQR, 25% - 75% percentile) similar to that of CMAQ (Figure S16). In July 2019, as depicted in Figure S17, all the criteria pollutants except CO demonstrated a comparable degree of response to emission reductions."*

[Figure]

*Figure 2: Coefficient of correlation (R) for responses of FastCTM and CMAQ to different emission scenarios and different air pollutants in January, 2023 (panel a) and July, 2023 (panel b).*

Regarding the training of the five operators, I was glad to see that there is more description but the new analysis of these operators raises some concerns. Figure 12 seems to show that CMAQ and FastCTM differ markedly in their estimation of the relative contribution of different processes to (what I assume is) the local rate of change of PM2.5, but again there is no quantitative assessment; just a qualitative assertion that the models are comparable but consistent.

**Response**: We added a description comparing process contributions of the two models with statistical analysis.

*"Simulated contributions of five major processes to hourly $PM_{2.5}$ concentration changes are compared between FastCTM and CMAQ at 139 stations (Figure S15) in the Sichuan-Chongqing region from October 12, 2024 to October 16, 2024, as shown in boxplots of Figure 11. Overall, the simulation results of the process contributions by FastCTM and its parent model CMAQ were relatively consistent. Higher degrees of consistency were found in simulations of emissions, advection processes, and diffusion processes between the two models. Contributions from chemical reactions of FastCTM exhibited overestimation compared to CMAQ, while contributions from deposition were underestimated. The differences in the simulated deposition and reaction contributions between the two models could be due to incomplete representation of influencing factors, given the complexity of the two processes. In general, the consistency between the two models provides confidence in the reliability of FastCTM for simulating and understanding the complex interplay of atmospheric processes that govern $PM_{2.5}$ levels."*

[Figure]

*Figure 3: Boxplots of contributions from five major atmospheric processes at 139 evaluation stations from October 13, 2024, to October 16, 2024, simulated by (a) CMAQ*

*and (b) FastCTM.*

The lack of quantitative analysis throughout the manuscript, but in particular in the new sections, means that I cannot recommend this article for publication and recommend a substantial restructuring to focus on reproducible, quantitative assessments of accuracy. I would strongly recommend that the authors seek to reduce the length of the manuscript by condensing the many qualitative comparisons and multi-panel plots into quantitative analyses with comparisons of key metrics of performance. Without these it is very difficult to evaluate the performance of the model against either the parent CMAQ model or existing reduced-order models, and therefore whether there is sufficient novelty.

**Response**: As the reviewer kindly suggested, we added more quantitative analysis regarding accuracy evaluation, emission sensitivity analysis and process contribution assessment. Qualitative analysis, such as Figures 9, 10, and 12 in the previous version, have been moved to the supplementary material and replaced by more-straightforward comparisons with statistical metrics.

Finally, the manuscript still needs some superficial improvement. The edits appear to have introduced numerous new grammatical errors which I would recommend the authors seek to correct. I would also ask that the authors incorporate higher-resolution images, as some (including Figure 12) are almost unreadable (at least in the PDFs I have access to).

**Response**: We conducted a thorough check on the manuscript to address grammatical accuracy, clarity, and consistency with scientific writing conventions. Images were originally generated in high resolution. We further improved their quality by revising the axis name, labels and legends.

Huang, L., Zhu, Y., Zhai, H., Xue, S., Zhu, T., Shao, Y., ... & Li, L. (2021). Recommendations on benchmarks for numerical air quality model applications in China–Part 1: PM 2.5 and chemical species. Atmospheric Chemistry and Physics, 21(4), 2725-2743.

Zhang, L., Tian, X., Zhao, Y., Liu, L., Li, Z., Tao, L., ... & Luo, Y. (2021). Application of nonlinear land use regression models for ambient air pollutants and air quality index. Atmospheric Pollution Research, 12(10), 101186.

**Review #2**

I thank the authors for addressing most of my comments. Here are additional comments:

From comments to reviewer #1:

For the scheme, masses are conserved. However, FastCTM as a whole is not mass conserved, because it also consists other neural network modules such as reaction and deposition.

--> This should be said explicitly; in other words, mass is not conserved by the model

**Response**: We added the description explicitly, as follows in Section 2.2,

*"With the scheme, this transport module itself is mass conserved, even though FastCTM is not mass conserved as a whole."*

From comments to reviewer #2:

"Even though these successful applications using deep learning methods to simulate individual atmospheric chemical and physical processes, there is an missing gap in coupling these NN operator replacements together as an complete deep learning based CTM. "

--> I don't think this point is properly motivated still. CTMs are slow but certain components (chemistry, transport) are much slower than others (deposition, emissions). If anything, the previous citations suggest that we do not want a complete deep learning based CTM due to accumulating errors and oversimplifying of the physical schemes. Just because you did the work in creating FastCTM does not mean the motivation and utility is intrinsic.

**Response**: We appreciate the reviewer's critical perspective on the motivation for developing a complete deep learning-based CTM by coupling individual neural network (NN) operators. This concern prompts us to clarify the unique value of such an integrated framework beyond merely combining existing components, and to address the potential challenges of error accumulation and physical oversimplification.

First, while prior work has successfully applied deep learning to simulate individual processes (e.g., chemical reactions, deposition; Kelp et al., 2022; Silva et al., 2019; Xia et al., 2024), these efforts remain fragmented. A "complete" deep learning CTM is necessary to capture these interdependencies—individual NN operators, in isolation, cannot replicate the holistic dynamics of air quality evolution that are critical for applications such as pollution episode

attribution, emission sensitivity analysis, or real-time forecasting. For example, understanding how a regional emission reduction affects PM$_{2.5}$ requires not only simulating emissions but also how those reduced species are transported, chemically transformed, and deposited—interactions that only a unified framework can model coherently.

Second, we acknowledge the risk of accumulating errors in full deep learning-based models, which the reviewer rightfully highlights. FastCTM's design explicitly addresses this by adopting a principle-informed structure (Section 2.2), where each module (transport, reaction, deposition, etc.) is constrained by the governing physical/chemical equations (e.g., Eq. 1). This distinguishes it from unconstrained "black-box" deep learning models, which are more prone to oversimplification and error propagation. As demonstrated by Sturm and Wexler (2020), physics-constrained machine learning frameworks can mitigate such risks by ensuring process interactions align with fundamental principles. Our results indicated that FastCTM maintains high agreement with CMAQ across long-term simulations (Section 3.1) and exhibits good consistent sensitivity to meteorology and emissions (Section 3.2).

Third, the utility of FastCTM stems from its ability to bridge the gap between the computational efficiency of deep learning and the functional completeness of traditional CTMs. Traditional CTMs, while accurate, are computationally prohibitive for many practical use cases—for example, high-resolution ensemble forecasting or rapid evaluation of hundreds of emission control scenarios (Efstathiou et al., 2024). Individual NN operators, while fast, cannot replace a full CTM in these contexts because they lack the integrated process chain needed to simulate end-to-end air quality. FastCTM addresses this by providing a unified tool that retains the multi-functionality of traditional CTMs (process attribution, sensitivity analysis) while achieving GPU-accelerated speeds (Section 3), making it feasible for applications where both speed and completeness are critical.

In summary, the motivation for FastCTM lies not in "intrinsic" value from coupling components, but in addressing a critical gap: the need for an efficient, integrated model that captures interconnected atmospheric processes—constrained by physical principles to avoid oversimplification—for real-world air quality management and research. This aligns with recent calls in the literature for physics-informed machine learning models that retain the interpretability and reliability of traditional models while leveraging the speed of deep learning (Irrgang et al., 2021; Reichstein et al., 2019).

**References**

Efstathiou, C. I., et al. (2024). Enabling high-performance cloud computing for the Community Multiscale Air Quality Model (CMAQ) version 5.3.3: performance evaluation and benefits for the user community. *Geoscientific Model Development*, 17, 7001–7027.

Irrgang, C., et al. (2021). Towards neural Earth system modelling by integrating artificial intelligence in Earth system science. *Nature Machine Intelligence*, 3, 667–674.

Kelp, M. M., et al. (2022). An online-learned neural network chemical solver for stable long-term global simulations of atmospheric chemistry. *Journal of Advances in Modeling Earth Systems*, 14, e2021MS002926.

Reichstein, M., et al. (2019). Deep learning and process understanding for data-driven Earth system science. *Nature*, 566, 195–204.

Silva, S. J., et al. (2019). A deep learning parameterization for ozone dry deposition velocities. *Geophysical Research Letters*, 46, 983–989.

Sturm, P. O., & Wexler, A. S. (2020). A mass- and energy-conserving framework for using machine learning to speed computations: a photochemistry example. *Geoscientific Model Development*, 13, 4435–4442.

Xia, Z., et al. (2024). Advancing Photochemistry Simulation in WRF-Chem V4.0: Artificial Intelligence PhotoChemistry (AIPC) Scheme with Multi-Head Self-Attention Algorithm.

"Interpretations of deep learning network are also widely vowed to improve their applications in earth system science and climate studies"

--> Again, perhaps a translation issue here, but I do not know what interpretations mean here. I also don't think this work will 'widely vowed to improve' climate studies, which are way ahead of their adoption of ML in their modeling approaches compared to air quality

**Response**: We appreciate the reviewer's insightful comment, which highlights the need for clearer terminology and contextualization. We apologize for the imprecision and confusion in our original statement. The term "interpretations" refers to model interpretability—i.e., the ability to trace predictions to underlying processes or mechanisms, rather than treating the model as an opaque "black box". This aligns with the core design of FastCTM: its principle-informed structure represents five modular processes, each with physically constrained formulations (Eqs. 3–14). As demonstrated in Section 3.3, this enables quantification of individual process contributions to pollutant concentration changes (e.g., Figure 10 shows transport dominating $PM_{2.5}$ changes in a pollution episode, while deposition offsets chemical production). Such interpretability distinguishes FastCTM from black-box deep learning models, enabling error attribution and physical insight.

We also acknowledge the reviewer's point about the relative maturity of machine learning adoption in climate studies compared to air quality modeling. Our intention was not to suggest that this work would revolutionize climate studies, but rather to position FastCTM within the broader context of interpretable machine learning in Earth system science. Interpretability is a shared priority across earth system science, including climate research. As noted by Irrgang et al. (2021), neural network applications in earth system modeling (whether for air quality, climate, or other domains) increasingly demand interpretability to ensure physical consistency and facilitate knowledge discovery. FastCTM contributes to this broader effort by demonstrating how principle-constrained architectures can retain machine learning efficiency while enabling process-level analysis.

We refined this sentence to: "*Enhancing the interpretability of deep-learning models is critical for advancing their application in Earth-system science, including climate and air-quality studies.*"

"FastCTM's design supports incremental integration of additional species (e.g., via user-defined modules) without overhauling the core framework. Future versions will explore adding VOCs and secondary organics to address broader research needs."

--> Can this be expanded on? How is this the case. If you wanted to add VOC species won't those species have to be added in multiple modules (e.g., emissions, chemistry)? Or can this be fine-tuned like in transformer-like models (also referred to as transfer learning in different settings). If so, how can you do this? It is unclear to me how this can be incrementally added without retraining the entire pipeline if they are trained together, even though each individual process is discretized differently.

**Response**: We appreciate the reviewer's critical question regarding the incremental integration of additional species (e.g., VOCs) into FastCTM. As noted, adding new species such as VOCs—which participate in multiple processes (emissions, chemical reactions, transport, etc.)—requires updates across multiple modules, and retraining of the modified model with the new variables is indeed necessary. However, if added species are chemical inactive, which did not participate in chemistry, we could freeze parameters in the chemical module to fine-tune FastCTM to adapt to new input. We revised the original description that clarify the need for retraining when adding new species, while still highlighting the model's modular design for targeted updates to avoid confusions, as follows,

"*FastCTM's modular, principle-informed architecture facilitates targeted updates to integrate additional species (e.g., VOCs or secondary organics) by focusing modifications on relevant*

*processes rather than overhauling the entire framework. However, adding new species, especially those participating in multiple atmospheric processes, requires updating associated modules and retraining the model with the expanded set of variables to ensure the model learns the new species' interactions with existing pollutants and processes. Future work will explore such expansions, leveraging the framework's modularity to streamline updates while retraining to incorporate the new species and their dynamics."*

"Response: As reviewer kindly pointed, FastCTM possibly "

--> You used this response for reviewer #1 which works well. But this is not the same thing I am asking. You can't see any diurnal errors here. Choose week long time series and show how we get to diurnally varying RMSEs. Furthermore, Figure 5 is a bit messy. There are many trajectories and it's hard to see the difference between CMAQ 2023 and FastCTM 2023.

**Response**: We apologize for not responding adequately to this specific concern in the previous revision. As suggested, we compared CMAQ forecasts and FastCTM forecasts by plotting time series of hourly pollutant concentrations averaged for all China at two winter dates, in Figure S6 of the supplementary material. It is clear that FastCTM did not predict the same values given the time of day, as the reviewer kindly asked in the previous revision. We can also conclude this from Figure 5. In this figure, daily average FastCTM simulations in 2023 correlate much better with CMAQ simulations in 2023 rather than in the training period of 2018-2022, indicating FastCTM is not just borrowing time-dependent rules from the training dataset. We also revised Figure 5 by adding statistical metrics for the time series.

Descriptions in the manuscript are revised accordingly, as follows in Lines 273-274 and 336-343:

*"Hourly RMSE values show clear diurnal variation with higher RMSE values in the nighttime than daytime, which could be due to higher hourly concentrations of air pollutants at night, except for $O_3$ (Figure S5 of SI)."*

*"As illustrated in Figure 5, the predictions made by FastCTM in 2023 align more closely with the actual CMAQ forecasts for that year with $R^2 = 0.94$ and 0.72 respectively for $PM_{2.5}$ and $O_3$, rather than with the forecasts generated from the training data of 2018-2022 with $R^2=0.54$ and 0.59. The NMB was also lower between FastCTM and CMAQ for the same year 2023. These results not only validate the adaptive learning capabilities of the FastCTM model but also indicate that the model is not using a simplistic approach of averaging concentrations from the previous five years based on time of day. Hourly time series plots of air pollutant concentrations (Figure S6 in the SI) further demonstrate that FastCTM appears to incorporate real-time*

*meteorological feedback, adjust for shifts in emission patterns, and leverage its learned relationships to provide more accurate and contemporaneous predictions."*

[Figure]

*Figure 4: The daily FastCTM forecasts comparing to CMAQ forecasts respectively in training period of 2018-2022 and evaluation period of 2023 for (a) PM$_{2.5}$ and (b) O$_3$. The gaps for FastCTM and CMAQ in 2023 are due to data unavailability in these days.*

"it is a notable finding that the MAE values tend to be higher in polluted areas. This can be attributed to the complex and dynamic nature of pollutant interactions in such regions. In polluted environments, there are often multiple sources of emissions, complex chemical reactions, and variable meteorological conditions that can lead to greater discrepancies between the model - predicted and actual pollutant concentrations. Conversely, the NMAE values exhibit an opposite trend, being lower in polluted areas. In these regions, the NMAE values typically hover around 0.2, in contrast to the relatively higher values of approximately 1 in cleaner areas."

--> I think this is a normal finding. If your FastCTM model is able to show the tendency effect of individual processes why can't you discuss errors in terms of this as well? This would actually be interesting and useful if you can definitively say that transport or diffusion is the problem in urban areas. Not many ML error analyses are able to do such comparisons

**Response**: We agree with the reviewer that higher MAE in regions with elevated pollutant concentrations is a normal and expected finding, rather than "notable" as initially described. This phenomenon arises because high-pollution areas are characterized by more intense atmospheric processes—including stronger emissions, more vigorous chemical reactions, and complex transport dynamics. Minor inaccuracies in modeling these intensified processes (e.g., small deviations in reaction rates) can be amplified, leading to larger absolute discrepancies in simulated concentrations, which directly contribute to higher MAE values. Conversely, the lower NMAE in these regions reflects the model's ability to capture the relative magnitude of pollution levels, as the large baseline concentrations normalize the impact of absolute errors.

We greatly appreciate the reviewer's insightful suggestion to decompose errors into contributions from specific processes (transport, diffusion, chemistry, etc.). This would indeed be a valuable and innovative contribution to ML-based air quality modeling, as few studies have been able to provide such process-specific error attribution. FastCTM's modular architecture theoretically enables this type of analysis by allowing us to isolate individual process contributions to prediction errors—for instance, determining whether transport errors dominate in urban areas or if chemical reaction uncertainties are the primary source of discrepancies in specific regions.

However, implementing such a comprehensive process-oriented error analysis would require extensive retrospective simulations with systematic perturbation of individual modules, along with detailed validation against process-specific observational data (e.g., tracer studies, chamber experiments). This would demand substantial computational resources and time that extend beyond the scope of the current study. We recognize this as a compelling direction for future research, where we can systematically conduct process-oriented error attribution by leveraging long-term historical datasets, enhanced computational capabilities, and potentially collaborating with observational campaigns designed to isolate individual process contributions.

For the current manuscript, we revised the relevant text to clarify that the observed MAE patterns in polluted areas are expected phenomena arising from the amplification of minor process-related uncertainties in regions with intense atmospheric activity, while acknowledging the potential for future process-specific error analysis enabled by FastCTM's modular design.

*"The spatial distributions of the mean absolute error (MAE) and the normalized mean absolute error (NMAE) are presented in Figure 3. For all six pollutants under consideration, MAE values tend to be higher in polluted areas. In polluted environments, there are often multiple sources of emissions, complex chemical reactions, and variable meteorological conditions that can lead to greater discrepancies between the predicted concentrations between the two models."*

*"It should also be noted that atmospheric physical and chemical processes are defined in principles-guided neural network modules in FastCTM. Their specific formulation was learned and optimized to minimize the sum of loss errors of all species concentrations, rather than being supervised by data of actual internal processes in CMAQ. The actual contributions of each process to pollutant concentration changes can be calculated using the integrated process rate (IPR) analysis and integrated reaction rate (IRR) analysis tools within CMAQ. Future studies could use these IPR and IRR results to supervise the simulated processes in FastCTM to further improve its simulation accuracy and robustness."*

I find most of the figures quite poorly made. The legends, texts, and labels are much too small:

Fig 2: legends too small. Are these all forecasts across all time in 2023?

3: color bar, axes labels too small

4: Not colorblind friendly, should be "R-squared"

5: see above comment in text, hard to see difference between 2023 CMAQ, FastCTM

9, 10: way too many subplots and its actually pretty hard to tell the difference between CMAQ and FastCTM in this format

12: too small, figure and labels are fuzzy

**Response**: Thanks. Figures are re-plot with larger axis, labels and legend texts, to improve quality and readability. Figures with more quantitative analysis are also added.

Fig.2: legend, axis tick name and label name are enlarged, evalution time are added. Colors of data points are also revised, as follows,

[Figure]

*Figure 5: The evaluation performances of FastCTM forecasts against CMAQ forecasts in 2023. Panel (a) and (b) respectively show RMSE values of criteria pollutants and the PM$_{2.5}$ components. Panel (c) and (d) respectively show R$^2$ values. It should be noted that RMSE value of CO corresponds to the right axis in panel (a).*

Fig.3 color bar and axes labels are enalarged as follows,

[Figure]

*Figure 6: Spatial distribution of mean absolute error (panels a, c, e, g, i, and k) and normalized mean absolute error for the six criteria pollutants (panels b, d, f, h, j, and l) of FastCTM compared with CMAQ in 2023.*

Fig.4: Colors are changed and label name are revised, as follows,

[Figure]

*Figure 7: The mean evaluation $R^2$ values for all 119 leading hours of FastCTM forecasts in warm/cold seasons, rural/urban areas, and coastal/inland areas.*

Fig.5: reivsed as shown above.

Correlation anlaysis are exhibited to better demonstrate the relation and difference between FastCTM and CMAQ, as follows. Fig.9 and 10 are moved to the supplementary material.

[Figure]

*Figure 8: Coefficient of correlation (R) for responses of FastCTM and CMAQ to different emission scenarios and different air pollutants in January, 2023 (panel a) and July, 2023 (panel b).*

Fig. 12 is replaced with quantative boxplots to better compare process contributions from different processes, as follows,

[Figure]

*Figure 9: Boxplots of hourly PM2.5 contribution changes from five major atmospheric processes at 139 evaluation stations from October 13, 2024, to October 16, 2024, simulated by (a) CMAQ and (b) FastCTM.*

---

## Author Response (AR3)

**Responses to Editor and Reviewers' Comments**

**Responses to Editor**

Many thanks for addressing the reviewers' comments and revising your manuscript.

The response of the reviewers is diverse. While reviewer #1 indicates technical corrections that I would like to ask you to consider carefully, reviewer #2 still has substantial comments concerning the approach (see reviewer comments).

I am inclined to accept this as an scientific debate that should be documented.

However, this requires a deeper discussion of these points. Please consider the reviewers comments carefully. I propose to include this in the discussion section, e.g. as individual subsections.

For example the reviewers concerns of

- Accuracy: In fact, literature precedent suggests that they are less accurate and when coupled together will have a greater, unexplained uncertainty.

- Choice of NN: if the deposition component of a CTM is not a computational bottleneck and you replace it with a NN solver that is less accurate, then what is the utility of that?

- Error identification: How do you move beyond a 'black box' model?

**Response**: Thank you for your guidance to deepen the discussion of key debates raised by reviewers. We have revised the Discussion section by adding three dedicated subsections (4.1-4.3) to explicitly address the core concerns: model accuracy/uncertainty, the rationale for neural network (NN) component selection, and interpretability beyond "black-box" limitations. These sections integrate perspectives from the reviewers and our responses, ensuring the scientific debate is thoroughly documented. The revised Discussion section are shown as follows,

"*4 Discussions*

*4.1 Model Accuracy and Uncertainty*

[revised manuscript text omitted]

**Responses to Reviewer #1**

I am grateful to the authors for their responses to my requests. The figures are much improved in particular, and I appreciate the additional statistical analysis. I believe that the manuscript is acceptable for publication pending copy-editing (there remain some typographical errors, e.g. line 406: "which could be caused by the reason that increased VOC"). I otherwise have no further comment.

**Response**: We appreciate your confirmation of the manuscript's readiness pending copy-editing. We have carefully revised the text to correct typographical errors. Thank you for your meticulous feedback.

**Responses to Reviewer #2**

The reviewers' replies to several of my comments were lackluster and did not address the main text

at all, instead opting to opine as a comment directly to me. Any reviewer comments I leave should be addressed in the text explicitly. The authors responded to my second comment about motivating this work in a poor manner. Each component or operator in a CTM has its own physics associated with it. Although the authors train individual NN operators with physics/chemistry constraints, this does not mean they are any more accurate than a traditional CTM. In fact, literature precedent suggests that they are less accurate and when coupled together will have a greater, unexplained uncertainty. For example, if the deposition component of a CTM is not a computational bottleneck and you replace it with a NN solver that is less accurate, then what is the utility of that? Having a "complete" deep learning CTM is ill-posed in a situation when specific CTM model components are accurate and fast. Further, as evidenced by the copious comments by Reviewer #1 concerning the presentation of results in a robust/quantitiatve manner, just because you incorporate physical constraints into the NN operators does not mean that they are more accurate/stable.

"However, implementing such a comprehensive process-oriented error analysis would require extensive retrospective simulations with systematic perturbation of individual modules, along with detailed validation against process-specific observational data (e.g., tracer studies, chamber experiments). This would demand substantial computational resources and time that extend beyond the scope of the current study."

--> This response is also lacking. The authors champion FastCTM as a tool that is able to circumvent the computational cost of a CTM, but then running a simple error analysis would "demand substantial computational resources". There is no need to run tracer studies or use chamber experiments but a simple error analysis visualized similarly to Figure 10 would be helpful. Otherwise, is this not the same as being a 'black box' model, which you claim to move beyond?

**Response**: We sincerely appreciate your detailed comments, which have helped strengthen the manuscript. We apologize for any previous oversights in addressing your concerns and have revised the main text extensively to incorporate our responses explicitly, particularly in the new subsections of the Discussion.

1. Accuracy and uncertainty of coupled NN modules

As highlighted in Section 4.1, we acknowledge that coupling NN modules may introduce uncertainties, as noted in the literature. To mitigate this, FastCTM adopts a principle-informed design where each module is constrained by physical/chemical equations (e.g., mass conservation in transport, kinetic theory in reactions). Our evaluation shows high consistency with CMAQ ($R^2 >$ 0.8 for most pollutants), but we also explicitly discuss limitations (e.g., lower $R^2$ for $NH_4^+$ due to simplified aerosol thermodynamics) and plans to reduce uncertainties using CMAQ's integrated process rate (IPR) data for supervised module training in the future.

2. Utility of NN for non-bottleneck components

Section 4.2 now addresses the utility of applying NNs to non-bottleneck components, clarifying our rationale on three fronts. First, we explain that even efficient traditional parameterizations have

known limitations (e.g., in deposition) where NNs can offer significant improvements. Second, we highlight that FastCTM is a flexible, hybrid framework that allows users to retain original CMAQ modules if they prioritize fidelity for a specific process. Finally, we emphasize that accelerating any component contributes to a crucial reduction in overall runtime, which is vital for computationally demanding applications like ensemble forecasting.

3.  Beyond "black-box" models: interpretability and error analysis

Section 4.3 demonstrates that FastCTM's modular design enables process-level attribution (e.g., Figure 10 shows transport dominating $PM_{2.5}$ changes in a pollution episode). While comprehensive process-oriented error analysis (e.g., isolating urban-rural discrepancies) is beyond the current scope, we outline plans to use observational datasets (e.g., tracer studies) for such analyses, enhancing transparency.

4.  Integration of responses into the main text

All key points from our responses are now incorporated into the revised manuscript, particularly in the Discussion subsections. We have ensured that all the concerns raised (e.g. uncertainty accumulation, "black-box" risks) are explicitly addressed in the text, not just in replies to reviewers. The revised texts in the Discussion section are shown as follows,

"*4 Discussions*

*4.1 Model Accuracy and Uncertainty*

[revised manuscript text omitted]

Thank you again for your insightful feedback, which has significantly improved the rigor and clarity of our work.